# CALCOCO1 acts with VAMP-associated proteins to mediate ER-phagy

Thaddaeus Mutugi Nthiga, Birendra Kumar Shrestha [iD], Eva Sjøttem, Jack-Ansgar Bruun [iD], Kenneth Bowitz Larsen [iD], Zambarlal Bhujabal, Trond Lamark* [iD] & Terje Johansen** [iD]

## Abstract

The endoplasmic reticulum (ER) plays important roles in protein synthesis and folding, and calcium storage. The volume of the ER and expression of its resident proteins are increased in response to nutrient stress. ER-phagy, a selective form of autophagy, is involved in the degradation of the excess components of the ER to restore homeostasis. Six ER-resident proteins have been identified as ER-phagy receptors so far. In this study, we have identified CALCOCO1 as a novel ER-phagy receptor for the degradation of the tubular ER in response to proteotoxic and nutrient stress. CALCOCO1 is a homomeric protein that binds directly to ATG8 proteins via LIR- and UDS-interacting region (UIR) motifs acting co-dependently. CALCOCO1-mediated ER-phagy requires interaction with VAMP-associated proteins VAPA and VAPB on the ER membranes via a conserved FFAT-like motif. Depletion of CALCOCO1 causes expansion of the ER and inefficient basal autophagy flux. Unlike the other ER-phagy receptors, CALCOCO1 is peripherally associated with the ER. Therefore, we define CALCOCO1 as a soluble ER-phagy receptor.

**Keywords** Autophagy; CALCOCO1; ER-phagy; FFAT; VAPA
**Subject Categories** Autophagy & Cell Death; Membranes & Trafficking; Organelles
**The EMBO Journal (2020) 39: e103649**

## Introduction

Organelles are intracellular membrane-confined structures that carry out specialized functions important for cell function and survival. Eukaryotic cells have different organelles such as the endoplasmic reticulum (ER), Golgi apparatus, mitochondria, lysosomes, and peroxisomes. The amount and vitality of each organelle is regulated depending on the energetic and functional needs of cells (Anding & Baehrecke, 2017). Surplus and damaged organelles are cleared through macro-autophagy (henceforth autophagy) (Okamoto, 2014; Anding & Baehrecke, 2017), an evolutionary conserved process that

delivers cytoplasmic materials for degradation in the lysosome (Mizushima & Komatsu, 2011; Ohsumi, 2014). Autophagy involves sequestration of cytoplasmic contents into double-membraned vesicles called autophagosomes, which then fuse with lysosomes to degrade their contents. At basal level, autophagy occurs in cells to maintain homeostasis by facilitating constitutive turnover of cytoplasmic contents. Autophagy also acts selectively in the degradation of excess components or toxic materials in the cell such as surplus or damaged organelles, protein aggregates, and invading pathogens (Johansen & Lamark, 2011, 2019; Stolz et al, 2014; Gatica et al, 2018; Kirkin, 2019). Autophagy is activated during stresses, such as starvation, to degrade cellular macromolecules in order to recycle nutrients and generate energy (Schroder, 2008; Ohsumi, 2014).

Autophagosome formation is mediated by evolutionary conserved core autophagy (ATG) proteins, which assemble into temporal hierarchical complexes to initiate the formation and expansion of the phagophores and their closure around the cargo to form autophagosomes. The co-ordinated actions of the first two complexes, ULK complex comprising FIP200, ATG13, ATG101, and ULK1/2, and PI3KC3 complex I comprising VPS34, BECN1, VPS15 and ATG14L, at the phagophore formation site, generate phosphatidylinositol-3-phosphate (PI3P). This recruits the PI3P-binding ATG2-WIPI complex, and the two ubiquitin-like conjugation systems mediating the formation of the ATG5-ATG12:ATG16L complex for the lipidation of ATG8 family proteins to the growing phagophore. The only integral membrane protein of the conserved core autophagy components, ATG9, is involved in the trafficking of vesicles adding some unknown components to the growing phagophore in a kiss and run fashion (Mizushima et al, 2011; Bento et al, 2016).

The selectivity in autophagy is mediated by selective autophagy receptors (SARs), which link the cargo material to the phagophore membranes (Johansen & Lamark, 2011, 2019; Stolz et al, 2014; Gatica et al, 2018; Kirkin, 2019). The linkage involves SAR binding to the cargo on one hand and to ATG8 family proteins on the phagophore membrane on the other (Pankiv et al, 2007; Birgisdottir et al, 2013; Rogov et al, 2014; Johansen & Lamark, 2019). The interaction with ATG8 family proteins is mediated by a LIR (LC3-interacting region) motif, which has a core sequence of [W/F/Y]xx[L/V/I], but also contains negatively charged residues inside or adjacent to the

Department of Medical Biology, Molecular Cancer Research Group, University of Tromsø—The Arctic University of Norway, Tromsø, Norway
*Corresponding author. Tel: +47 77644720; E-mail: trond.lamark@uit.no
**Corresponding author. Tel: +47 77644720; E-mail: terje.johansen@uit.no

core motif. This motif interacts with a LIR docking site (LDS) in the ATG8 family protein, consisting of two hydrophobic pockets mediating the interaction with the core motif and adjacent positively charged side chains forming electrostatic interactions (Johansen & Lamark, 2019; Wirth et al, 2019). Recently, it has emerged that ATG8 family proteins may also recognize ubiquitin-interacting motif (UIM)-like sequences present on some receptors, like RPN10, to recruit cargo-receptor complexes to the phagophore membranes (Marshall et al, 2019). The binding site for UIM-like motifs is called UIM-docking site (UDS) and is on the opposite side of the ATG8 molecule relative to the LDS. In mammals, there are six different ATG8 family proteins, i.e. the MAP1LC3 (microtubule associated protein 1 light chain 3) subfamily consisting of LC3A (two isoforms), LC3B and LC3C, and the GABARAP (GABA type A receptor-associated protein) subfamily consisting of GABARAP, GABARAPL1, and GABARAPL2. The lipidated ATG8 proteins act as adaptors for the recruitment of LDS- or UDS-interacting proteins to the phagophore. One essential function of ATG8 proteins in selective autophagy is to act as adaptors for the attachment of SARs and cargos to the inner surface of phagophore. However, ATG8 family proteins are also essential for autophagosome formation and maturation, mediated, at least in part, by recruiting core autophagy proteins and proteins involved in the transport or fusion of autophagosomes with lysosomes (Kriegenburg et al, 2018; Johansen & Lamark, 2019). Expanding phagophores and autophagosomes therefore are congregates of autophagy regulatory proteins, cargo materials, and receptors, all associating directly or indirectly.

Clearance of surplus or damaged organelles, such as endoplasmic reticulum (ER), is an important function of selective autophagy (Wilkinson, 2019b). Mammalian ER is a continuous membrane bound organelle consisting of the nuclear envelope (NE) and a cytoplasmic peripheral ER made up of sheets and reticulated tubular network. The ER plays important roles in processes such as protein synthesis and folding, mitochondrial division, calcium storage and signaling, lipid synthesis and transfer and detoxification (Chen et al, 2013; Nixon-Abell et al, 2016; Schwarz & Blower, 2016). In response to physiological or pathological conditions such as nutrient deprivation, accumulation of unfolded proteins, or exposure to chemicals, the ER engages the unfolded protein response pathways (UPR) to restore homeostasis.

The UPR is characterized by signaling events from ER integral membrane sensor proteins: protein kinase RNA-like ER kinase (PERK), activating transcription factor 6 (ATF6), and inositol-requiring enzyme 1α (IRE1α), which cumulatively trigger inhibition of global protein translation while transcriptionally upregulating ER chaperones, ER-associated degradation (ERAD) proteins, and apoptotic mediators, causing the ER to undergo spatiotemporal changes in morphology, molecular composition, and functional specification. More particularly, UPR increases the ER volume and the expression of ER-resident proteins to buffer ER functions. At the same time, there is a continuous remodeling and turnover of the ER to restore homeostasis. Selective autophagic degradation of ER fragments and components, called ER-phagy, contributes to this remodeling (Bernales et al, 2007; Fregno & Molinari, 2018; Wilkinson, 2019b). Autophagy-deficient cell lines contain expanded ER while inhibition of general autophagy by depleting ATG5 or ATG7 has been shown to cause ER stress and dilation, suggesting that ER-phagy is a critical process for ER homeostasis (Jia et al, 2011; Antonucci et al, 2015).

ER-phagy in yeast is mediated by two receptors, Atg39 and Atg40, which play critical roles in sequestering ER fragments into autophagosomes (Mochida et al, 2015). In mammals, six ER-phagy receptors, targeting different ER sub-domains for degradation, have so far been identified: FAM134B, RTN3L, SEC62, CCPG1, ATL3, and TEX264 (Khaminets et al, 2015; Fumagalli et al, 2016; Grumati et al, 2017; Smith et al, 2018; An et al, 2019; Chen et al, 2019; Chino et al, 2019). A recent study also found that a COPII subunit, SEC24C, was required for starvation-induced ER-phagy in concert with FAM134B and RTN3 ER-phagy receptors (Cui et al, 2019). FAM134B is a reticulon homology domain-containing protein, and it has been shown to mediate basal and starvation-induced degradation of ER sheets through interaction with atlastin2 (ATL2) (Khaminets et al, 2015; Liang et al, 2018). FAM134B also interacts with calnexin to mediate degradation of misfolded procollagen (PC) (Fregno et al, 2018; Forrester et al, 2019). RTN3L mediates starvation-induced degradation of tubular ER and also contain a reticulon homology domain which anchors it to the ER tubules (Grumati et al, 2017).

ATL3 is a GABARAP-interacting ER-phagy receptor for the degradation of tubular ER, while SEC62, a component of the ER translocon that promotes co-translational of proteins into ER, has been shown to function as ER-phagy receptor during recovery from ER stress (Fumagalli et al, 2016; Chen et al, 2019). CCPG1 is an ER transmembrane protein that mediates ER-phagy of the tubular ER during starvation and ER stress by interacting with GABARAP and FIP200 (Smith et al, 2018). TEX264 was recently identified as single-pass transmembrane ER-phagy receptor responsible for the turnover of a large number of ER proteins during nutrient starvation (An et al, 2019; Chino et al, 2019). Very recently, p62 and the ER transmembrane E3 ligase TRIM13 was implicated in ER-phagy induced by proteotoxic stress via the N-degron pathway (Ji et al, 2019). This ER-phagy pathway is important in ER protein quality control and is activated by the binding of p62 to N-terminally arginylated proteins. Binding of p62 to TRIM13 then activates the E3 ligase and this creates a platform for ER-phagy induction. The involvement of p62 in this autophagy pathway shows that resident ER proteins and soluble SARs may co-operate in ER-phagy processes. Despite the growing number of identified ER-phagy receptors, it is not known how and whether the receptors co-operate to promote degradation of the ER and how such co-operation could be regulated. In addition, loss of the known receptors does not block ER-phagy completely and the effects of their loss appears to be tissue-restricted (Wilkinson, 2019b), suggesting that the loss is compensated by yet unidentified receptors.

CALCOCO1 is an evolutionary conserved protein and a paralog to TAX1BP1 and NDP52, two well-known selective autophagy receptor proteins. The three proteins form a small protein family with substantial similarity and identity with a similar domain structure composed of an N-terminal SKIP carboxyl homology (SKICH) domain, middle coil–coil regions (CC) and varying carboxy terminal (CT) domains that contain one or two zinc finger domains. In addition, they contain an atypical LIR (CLIR) motif (LVV) in the linker region between the SKICH domain and the coiled-coil domain (von Muhlinen et al, 2012; Tumbarello et al, 2015; Fig 1A). Despite this similarity, no role for CALCOCO1 in autophagy has been defined so far. However, in a quantitative proteomics study aimed at identifying novel and known

autophagosome-enriched proteins in human cells, CALCOCO1 was found to be enriched in autophagosomes from pancreatic cancer cell lines (Mancias *et al*, 2014). CALCOCO1 was also one of the top hits in another quantitative proteomics study of proteins that were stabilized in ATG16L1 KO murine bone marrow-derived macrophages relative to WT controls (Samie *et al*, 2018). Here, we show that CALCOCO1 is continuously degraded by autophagy. Detailed studies revealed that CALCOCO1 is homomeric and has both LIR and UIR motifs for co-dependent binding to GABARAP subfamily proteins. CALCOCO1 acts as a soluble selective autophagy receptor for ER-phagy. It accomplishes this by interacting with ER tethering proteins VAPA and VAPB via a FFAT motif.

# Results

## CALCOCO1 is homomeric

TAX1BP1 and NDP52 self-associate through their coiled-coil domains (Sternsdorf *et al*, 1997; Ling & Goeddel, 2000) and heterodimerizes with each other (Morriswood *et al*, 2007). To explore whether CALCOCO1 is homomeric, full-length EGFP-CALCOCO1 and Myc-CALCOCO1 were co-expressed in HEK293 cells and EGFP-CALCOCO1 was pulled down from cell extracts using GFP-TRAP. An efficient co-precipitation of Myc-CALCOCO1 indicated that CALCOCO1 is homomeric (Fig EV1A). To clarify which domain in CALCOCO1 is mediating the self-association, Myc-tagged deletion mutants of CALCOCO1 were tested in the same immuno-precipitation experiment for interaction with full-length GFP-CALCOCO1. Only the deletion mutant containing the CC domain (145–513) was immunoprecipitated by full-length EGFP-CALCOCO1, implying that the observed self-association is mediated by the CC domain (Fig EV1A). Supporting such a conclusion, a deletion construct lacking the CC domain (Δ145–513) did not interact with full-length CALCOCO1 (Fig EV1A). Further, we tested whether the self-oligomerization of CALCOCO1 occurred by direct interaction. The same combinations of full-length GFP-CALCOCO1 and Myc-tagged deletion mutants were now co-translated *in vitro* in the presence of $^{35}$S-methionine. Immunoprecipitations were then performed followed by autoradiography analysis. As in the HEK293 cell extracts, the only deletion construct that co-precipitated with GFP-CALCOCO1 was Myc-CALCOCO1 (145–513) encompassing the CC domain (Fig EV1B). The CC domain of CALCOCO1 is separated into three coiled-coil regions (CC1–3) (Fig 1A). To determine which of the CCs contributes to the homomerization, Myc-CALCOCO1 constructs containing a specific deletion of each of the coiled-coil regions were also tested. When precipitated from cell extracts, none of the individual CC deletions affected the self-interaction (Fig EV1A). However, in the *in vitro* assay, a specific deletion of CC3 (Δ413–513) prevented the interaction and clearly had a much more pronounced effect than a deletion of any of the other internal coiled-coil regions (Fig EV1B).

Next, we tested whether CALCOCO1 heterodimerizes with TAX1BP1 and NDP52. GFP-CALCOCO1 was *in vitro* co-translated with either Myc-CALCOCO1, Myc-NDP52, or Myc-TAX1BP1 followed by immunoprecipitation using GFP-Trap. Autoradiography analysis showed that GFP-CALCOCO1 co-precipitated with Myc-CALCOCO1, but neither with Myc-TAX1BP1 nor Myc-NDP52,

indicating that CALCOCO1 does not heterodimerize with these paralogs (Fig EV1C).

An important difference between CALCOCO1 and its paralogs is the presence of ubiquitin-binding zinc fingers in NDP52 and TAX1BP1. However, although CALCOCO1 contains a C-terminal zinc finger domain too, it does not bind to ubiquitin (Thurston *et al*, 2009). The C-terminus of NDP52 also interacts with galectins to mediate xenophagy (Thurston *et al*, 2012). To determine whether CALCOCO1 interacts with galectins, Myc-CALCOCO1 was *in vitro*-translated and tested for interaction with GST-tagged galectin-3 and galectin-8 in *in vitro* pull-down assay, whereupon no interaction was found. In contrast, galectin-8 interacted with both TAX1BP1 and NDP52 (Fig EV1D).

## CALCOCO1 is degraded by macro-autophagy

To investigate the possible role of CALCOCO1 in autophagy, we first tested whether CALCOCO1 is degraded in the lysosome or in the proteasome by monitoring levels in the presence of either the lysosomal and autophagy inhibitor bafilomycin A1 (Baf A1), or the proteasome inhibitor, MG132. In normally growing HeLa (Fig 1B and C) and MEF (Fig 1D and E) wild-type (WT) cells, treatment with Baf A1 resulted in an accumulation of endogenous CALCOCO1, similar to the accumulation observed for autophagy receptor p62, suggesting basal turnover of CALCOCO1 by autophagy. Upon induction of autophagy by nutrient starvation, the amount of CALCOCO1 in the starved cells reduced significantly after 6 h. The reduction was blocked by treating the cells with Baf A1 during the starvation period (Fig 1B–E), suggesting that CALCOCO1 is an autophagy substrate during starvation. The lysosomal degradation of endogenous CALCOCO1 was confirmed by Western blots of extracts from human BJ-1 diploid fibroblasts treated for different times with Baf A1 or MG132 (Fig EV2A). To clarify whether macro-autophagy was involved in the degradation, we investigated the turnover of CALCOCO1 in autophagy-deficient cells. In ATG8 knock out (KO) HeLa cells (Fig 1B and C), ULK1 KO MEF cells (Fig EV2B), and Atg5 KO MEF cells (Fig 1D and E), both basal and starvation-induced degradation of CALCOCO1 were impaired, suggesting that the degradation of CALCOCO1 is dependent on macro-autophagy.

Next, we tried to look at the intracellular localization of endogenous CALCOCO1, but the endogenous protein was poorly detected by immunostaining. Therefore, we stably expressed EGFP-CALCOCO1 from a tetracycline-inducible promoter in CALCOCO1 KO Flp-In T-REx HeLa cells (Appendix Fig S1). Imaging of these cells revealed that a large proportion of the stably expressed EGFP-CALCOCO1 formed a perinuclear pattern characteristic of Golgi and endoplasmic reticulum (ER) localization. Co-imaging with the cis-Golgi marker protein GM130 displayed extensive co-localization, strongly indicating that a significant fraction of CALCOCO1 is localized in cis-Golgi structures (Fig 1F). We also observed extensive co-localization of EGFP-CALCOCO1 with endogenous p62 and LC3 in cytoplasmic puncta (Figs 1G–I and EV2C). Addition of Baf A1 strongly increased the number of EGFP-CALCOCO1 puncta. About 50% of these puncta co-localized with p62 and LC3 (Fig 1I), suggesting that CALCOCO1 is degraded by autophagy together with p62 and LC3 (Bjørkøy *et al*, 2005). LAMP1 staining of cells treated with Baf A1 demonstrated localization of EGFP-CALCOCO1 dots inside LAMP1-labeled structures (Fig 1J). Baf A1 treatment strongly

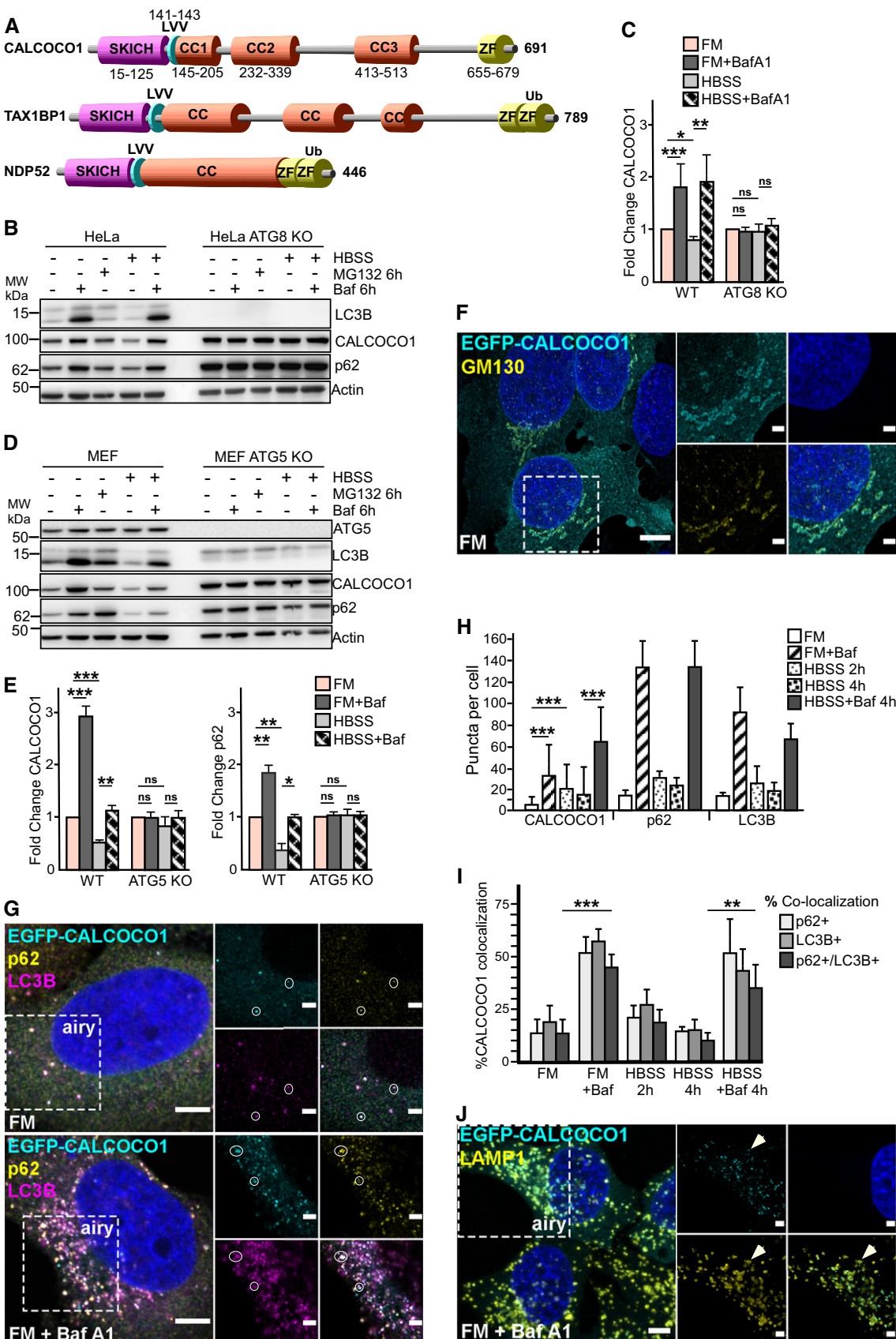

Figure 1.

**Figure 1. CALCOCO1 is degraded by macro-autophagy.**

A    Domain architecture of CALCOCO paralogs showing the SKICH domain, a conserved LIR motif (LVV), coiled-coli regions (CC), and zinc finger domains (ZF).

B–E  Immunoblot analysis of indicated cell lines, starved for 6 h (HBSS) as indicated, and treated with 25 μM MG132 or 200 ng/ml of bafilomycin A1 (Baf A1) for the indicated times. In (C, E), endogenous CALCOCO1 is analyzed and the bars represent the mean ± SD of band intensities relative to the actin loading control, as quantified using ImageJ of three independent experiments. Statistical comparison was analyzed by one-way ANOVA followed by Tukey multiple comparison test and significance displayed as ***$P < 0.001$, **$P < 0.005$, *$P < 0.01$; ns is not significant.

F    A representative micrograph using widefield and deconvolution microscopy of HeLa CALCOCO1 KO cells stably expressing EGFP-CALCOCO1 and immunostained for endogenous GM130. Scale bars are 5 and 2 μm (zoomed inset).

G    Same cells as in (E) were left untreated or treated with Baf A1 for 6 h and then immunostained for endogenous p62 and LC3B. Scale bars are 5 μm for the confocal microscopy images and 2 μm for the airyscans.

H    CALCOCO1, p62, and LC3B puncta in the indicated conditions, counted using an automated system. The error bars represent mean ± SEM of puncta per cell of three independent experiments per condition and 250 cells per experiment. Statistical comparison was analyzed by one-way ANOVA followed by Tukey multiple comparison test. Significance is displayed as ***$P < 0.001$.

I    Percentage of co-localization of CALCOCO1 puncta with p62 and/or LC3B in cells treated as indicated. The error bars represent mean ± SEM of three independent experiments per condition and over 250 cells per experiment. Statistical comparison was analyzed by one-way ANOVA followed by Tukey multiple comparison test. Significance is displayed as ***$P < 0.001$.

J    HeLa CALCOCO1 KO cells stably expressing EGFP-CALCOCO1 were treated with Baf A1 for 6 h and immunostained for endogenous LAMP1. Scale bars are 5 μm for the confocal microscopy images and 2 μm for the airyscans.

increased the association of EGFP-CALCOCO1 with LAMP1 rings (Fig EV2D), further supporting that CALCOCO1 is degraded by autophagy. In response to starvation, the localization pattern of EGFP-CALCOCO1 became more punctated and dispersed (Figs 1H and EV2C). EGFP-CALCOCO1 also co-localized with GABARAP in puncta (Fig EV2E).

### CALCOCO1 binds directly to ATG8 family proteins with preference for the GABARAP subfamily

In *in vitro* GST pull-down binding assays, CALCOCO1 interacted with several of the human ATG8 family proteins (Fig 2A). The strongest interaction was seen with GST-tagged GABARAP, but a strong interaction was also seen with GABARAPL1 and GABARAPL2 and a weaker interaction with LC3B and LC3C. We also performed GST pull-down assays using HeLa cell extracts from cells transfected with Myc-CALCOCO1. This assay similarly revealed a binding preference for the GABARAP subfamily (Fig 2B).

Previous studies have reported that an atypical LIR core motif (LVV), engaging only one of the hydrophobic pockets used by canonical LIRs, mediate the interactions of TAX1BP1 and NDP52 with ATG8 protein family (von Muhlinen *et al*, 2012; Whang *et al*, 2017). To define the role of LIR motif in CALCOCO1, we mutated the core motif (LVV to AAA) which resulted in a substantially reduced interaction with GABARAPL2. However, the interactions with the other ATG8s were only partially reduced (Fig 2C and D), suggesting the existence of an additional binding motif. Hence, we generated deletion mutants lacking or containing SKICH + LIR (1–144), CC (145–513), and CT (514–691), and some of the constructs also carrying the LVV to AAA LIR mutation (mLIR) (Fig 2C). A simultaneous mutation of LIR and deletion of CT (mLIR ΔCT) completely abrogated the interaction with all the tested ATG8 family proteins (Fig 2D). Thus, our data support an important role for the LIR motif, but the interaction also depends on an additional ATG8 family interaction motif in the C-terminal region of CALCOCO1. We also noted that CALCOCO1 ΔCC containing both binding motifs interacted strongly with ATG8 proteins, while constructs containing only one of the binding motifs appeared to depend on the CC domain for efficient interaction (Fig 2D).

To identify the C-terminal motif, we deleted 11, 21, 31, 41, 51, or 61 residues from the C-terminal end of CALCOCO1 ΔLIR (Δ126–144) and tested the interaction with GABARAP subfamily proteins (Figs 2E and EV3A). While a deletion of 41 amino acids (Δ651–691) had no apparent effect on the interaction, a deletion of the C-terminal 51 amino acids (Δ641–691) abolished the interaction (Fig 2E). A deletion of the C-terminal 68 amino acids (Δ623–691) similarly abolished the interaction (Fig 2F). We also compared the effect of deleting residues 651–679 or 654–679 and found that the extended deletion of residues 651–653 had a small but detectable effect on the interaction (Fig EV3B). We therefore consider these residues to form part of the ATG8-interacting motif. However, the zinc finger domain (residues 655–679) does not seem to be important for the interaction. To identify the N-terminal extension of the interaction, we made several internal deletions within the predicted ATG8-interacting region of CALCOCO1, and two of these (Δ615–634 and Δ619–646) strongly reduced the interaction (Fig 2G and H). Thus, we propose that the ATG8 interaction is mediated by the LIR (sdilLVVpkatvl) and the region encompassing amino acids 615–653 (EEANLLLPELGSAFYDMASGFTVGTLSETSTGGPATPTW; Fig 2C). Finally, we demonstrated that WT CALCOCO1, but not the ΔLIR + Δ623–691 mutant, could pull down endogenous LC3B and GABARAP (both lipidated and unlipidated forms) from cell extracts (Fig 2I).

### CALCOCO1 binds both to LDS and UDS of ATG8 family proteins

Recently, a novel docking site on ATG8 family proteins binding to ubiquitin-interacting motif (UIM)-like sequences was reported (Marshall *et al*, 2019). This UIM-docking site (UDS) is located on the opposite side of the ATG8 proteins relative to the LDS. This makes it possible for ATG8 proteins to simultaneously recruit both LIR and UIM-containing proteins (Marshall *et al*, 2019). To test if CALCOCO1 interacts with the two sites, we made GST-tagged GABARAP subfamily constructs with LDS (mLDS) or UDS (mUDS) point mutations and tested their binding to *in vitro*-translated CALCOCO1. All the tested UDS and LDS + UDS mutants completely lost the interaction with full-length CALCOCO1 (Fig 3A). These results suggest that GABARAP subfamily proteins require UDS contacts to stabilize their interactions with CALCOCO1, suggesting

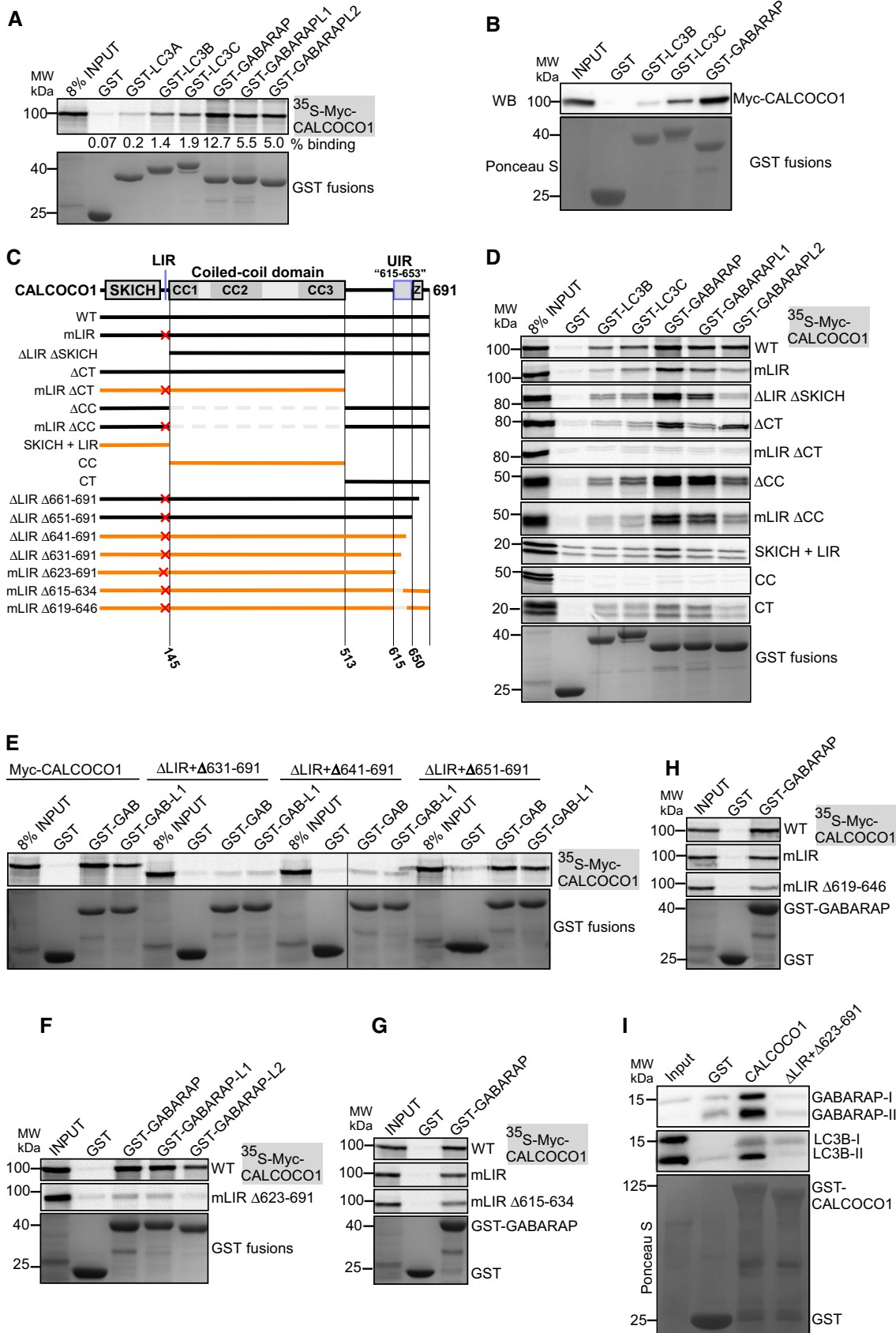

**Figure 2.**

**Figure 2. CALCOCO1 binds directly to ATG8 family proteins with preference for the GABARAP subfamily.**

A  GST pull-down binding assay of *in vitro* transcribed/translated [35]S-Myc-CALCOCO1 with recombinant GST-tagged ATG8 family proteins. GST and GST fusion proteins were visualized by Coomassie Brilliant Blue staining (bottom panel), and the co-precipitated Myc-CALCOCO1 was detected by autoradiography (upper panel). The numbers below the AR represent % binding in the shown AR.

B  GST pull-down assay of transiently transfected Myc-CALCOCO1 from HEK293 cell extracts with recombinant GST-tagged ATG8 family proteins. GST and GST fusions were visualized by Ponceau S staining (bottom panel), and co-precipitated Myc-CALCOCO1 detected by immunoblotting with anti-Myc antibody (upper panel).

C  CALCOCO1 deletion constructs used to map the ATG8 interactions. The red X indicate a point mutation or deletion of the LIR motif. Constructs with no or very weak interaction are indicated in orange.

D  D-H GST pull-down assays of indicated *in vitro* transcribed/translated [35]S-Myc-CALCOCO1 constructs with indicated recombinant GST-tagged ATG8 family proteins. Precipitated GST and GST fusions and co-precipitated Myc-CALCOCO1 constructs were analyzed as in (A).

E  GST pull-down assay of endogenous GABARAP from HEK293 cell extracts with recombinant GST-tagged CALCOCO1 constructs. GST and GST fusions were visualized as in A, and co-precipitated GABARAP with anti-GABARAP antibody (upper panel).

that the C-terminal motif of CALCOCO1 interacts with the UDS. The mutation of LDS in GABARAP and GABARAPL1 strongly inhibited the interaction with CALCOCO1 (Fig 3A), but a similar LDS mutation in GABARAPL2 had no effect on the interaction (Fig 3A).

The tolerance of the GABARAPL2-CALCOCO1 interaction toward a loss of the LDS was unexpected, since this interaction was strongly affected by a LIR mutation (Fig 2D). To specifically look at the LIR-LDS interaction, we performed GST pull-down assays with LDS-mutated ATG8 family proteins and CALCOCO1 ΔCT (Fig 3B). As expected, since binding to the ΔCT construct depends on the LIR-LDS interaction, binding of all ATG8s, including GABARAPL2, was strongly impaired by an LDS mutation (Fig 3B). Similarly, we tested CALCOCO1 constructs lacking the LIR motif, i.e. mLIR or ΔSKICH. It appeared that the LDS mutation in GABARAPL2 had a strong and positive effect on its interaction with LIR deleted CALCOCO1 constructs (Fig 3B). This probably explains why the LDS mutation in GABARAPL2 did not inhibit the full-length CALCOCO1 interaction, but it also seems to indicate that a mutation of LDS in GABARAPL2 has an unexpected positive effect on the UDS interaction.

Because of the possibility that mutations in the LDS or UDS site of ATG8 family could interfere with binding of proteins to the UDS or LDS site, respectively, we tested the binding of the LDS and UDS mutants to p62/SQSTM1, a protein known to bind ATG8 family proteins via LIR-LDS contact only (Pankiv *et al*, 2007). As expected, the LDS mutants lost interactions with p62 but UDS mutants had no effect on the binding (Fig 3C), suggesting that mutation of UDS sites did not interfere with interactions at the LDS sites.

## TAX1BP1, but not NDP52, binds to GABARAP via a region interacting with the UDS

NDP52 is reported to bind preferentially to LC3C via its LIR motif while TAX1BP1 interacts with LC3C, GABARAP, and GABARAPL1 (von Muhlinen *et al*, 2012; Whang *et al*, 2017). In *in vitro* GST pull-down assays, we confirmed that NDP52 binds preferentially to LC3C as reported (von Muhlinen *et al*, 2012), but also observed a potent interaction with GABARAP (Fig EV3C). TAX1BP1 interacted most strongly with LC3C and GABARAP (Fig EV3D). Mutation of the LIR motif (LVV to AAA) in NDP52 abolished the interaction with both LC3C and GABARAP (Fig EV3C). Similar to CALCOCO1, mutation of the LIR (mLIR) sequence in TAX1BP1 only reduced the interaction with GABARAP but did not eliminate binding (Fig 3D and E). Given that CALCOCO1 and TAX1BP1 are paralogous proteins, we reasoned that TAX1BP1 could also have

UIM-like motif in the C-terminal half that binds to the UDS site of GABARAP. We therefore tested the binding of GABARAP LDS, UDS, and LDS + UDS mutants to TAX1BP1. Both GABARAP UDS and GABARAP LDS + UDS mutants completely abolished the interaction with TAX1BP1 (Fig 3D), suggesting involvement of UIM-UDS interface in the binding of TAX1BP1 to GABARAP, additively to the LIR-LDS interface.

To identify the UIM motif in TAX1BP1, we made a series of deletions in the C-terminal half of TAX1BP1 mLIR and examined their binding to WT, LDS, and UDS mutants of GABARAP. TAX1BP1 mLIR bearing a deletion of amino acids 701–789 abolished interaction with both the WT and mLDS versions of GABARAP (Fig 3D), suggesting amino acids 701–789 contain the region for UDS contact during TAX1BP1 interaction with GABARAP. A strong inhibition was also caused by a deletion of residues 725–789 (Fig EV3D). These results indicate that TAX1BP1, like CALCOCO1, bears both LIR and UIM-like motifs that interact with ATG8 family proteins co-dependently. However, inspection of the sequences of the regions of CALCOCO1 (amino acids 615–653) and TAX1BP1 (amino acids 725–786) required for binding to UDS reveal no homology and there is also no homology to the UIM sequences reported to bind to ATG8s by Marshall *et al* (2019). Therefore, we suggest to call these regions UDS-interacting region (UIR).

## Degradation of CALCOCO1 is dependent on binding to ATG8 family proteins

To test whether the degradation of CALCOCO1 is dependent on its binding to ATG8 family proteins, we stably expressed EGFP-CALCOCO1 mLIR + Δ623–691 in Flp-In T-REx CALCOCO1 KO HeLa cells. In these cells, reconstituted EGFP-CALCOCO1 strongly accumulated in response to the treatment with Baf A1 both in FM and starvation conditions (Fig 3E and F), indicating efficient degradation of the WT protein by autophagy. However, the amount of the reconstituted EGFP-CALCOCO1 mLIR + Δ623–691 was neither affected by addition of Baf A1 nor by starvation (Fig 3E and F). This strongly suggests that the degradation of CALCOCO1 is dependent on binding to ATG8 family proteins via LIR and UIR-binding motifs. HeLa CALCOCO1 KO cells reconstituted with EGFP-CALCOCO1 mLIR + Δ623–691 also revealed a complete loss of co-localization with p62/LC3-positive puncta in cells treated with Baf A1 (Fig 3G). Furthermore, the mutant construct did not respond to starvation, and the starvation-induced redistribution into puncta seen for WT CALCOCO1 was not seen with the mutant (Appendix Fig S2).

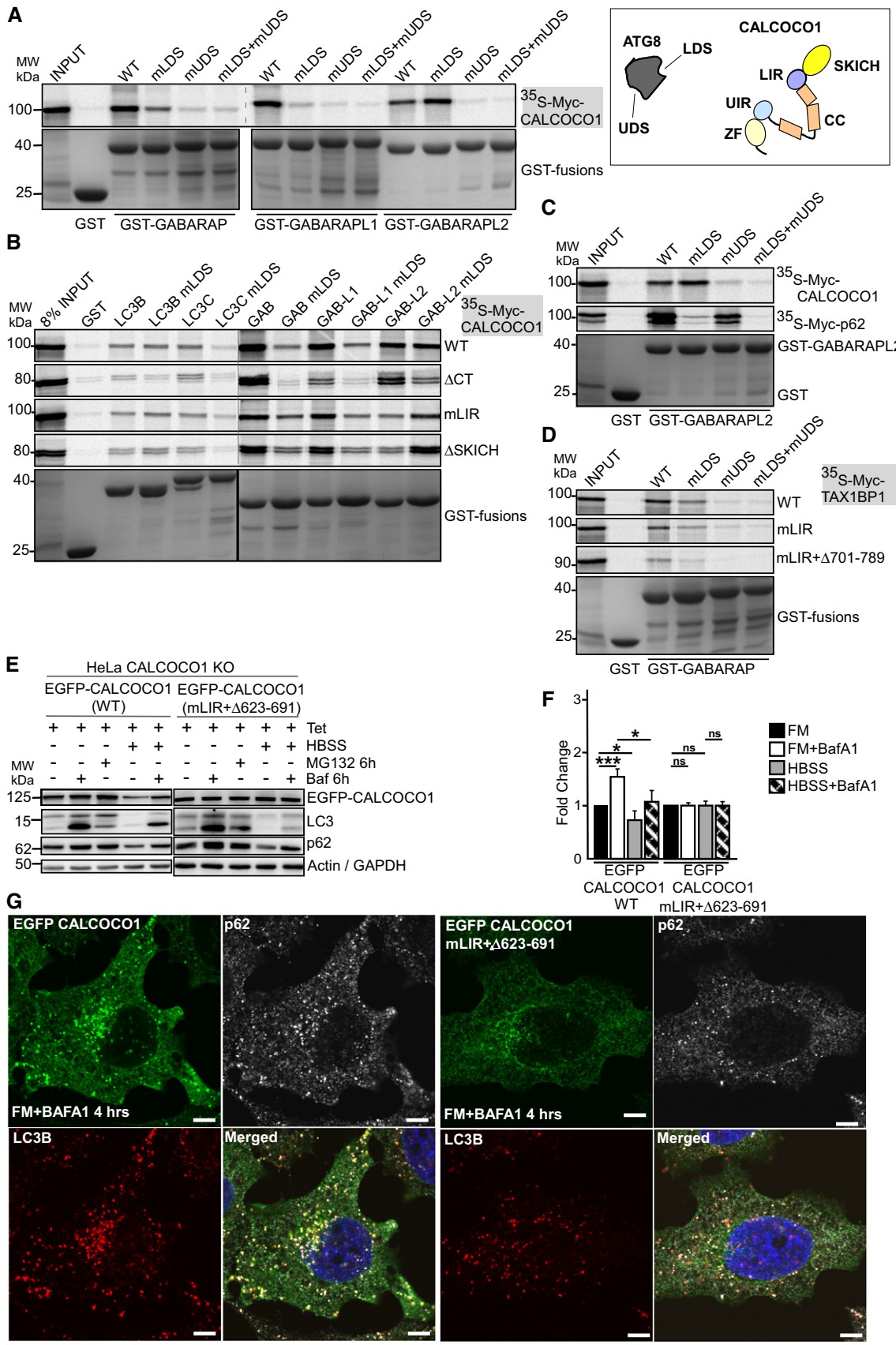

Figure 3.

**Figure 3.  CALCOCO1 binds both to LDS and UDS of ATG8 family proteins.**

A, B   GST pull-down testing binding of indicated *in vitro* transcribed/translated ³⁵S-Myc-CALCOCO1 constructs with indicated recombinant GST-tagged ATG8 family
       proteins (left). Cartoon of CALCOCO1 with domain organization indicated and the location of LIR and UIR motifs. The presence of two well separated binding
       surfaces on ATG8 proteins binding to LIR (LDS) and UIR (UDS) is indicated (right).
C      GST pull-down assays of *in vitro* transcribed/translated ³⁵S-Myc-CALCOCO1 and ³⁵S-Myc-p62 with recombinant GST-GABARAPL2 (WT and indicated mutants).
D      GST pull-down assays of *in vitro* transcribed/translated ³⁵S-Myc-TAX1BP1 (WT and indicated mutants) with recombinant GST-GABARAP (WT and indicated
       mutants).
E, F   Immunoblot analysis of HeLa CALCOCO1 KO cell lines stably transfected with WT EGFP-CALCOCO1 or EGFP-CALCOCO1 mLIR + Δ623–691. Cells were induced with
       tetracycline for 24 h and then starved or treated with MG132 or Baf A1 as indicated. The blot panels are from more than one Western blot experiment but for
       clarity, only a single actin/GAPDH loading control is shown. In (F), the bars represent the mean ± SD of band intensities relative to the actin or GAPDH loading
       controls of three independent experiments quantified using ImageJ. Statistical comparison was analyzed by one-way ANOVA followed by Tukey multiple
       comparison test and significance displayed as ***P < 0.001, *P < 0.01; ns is not significant.
G      HeLa CALCOCO1 KO cells stably expressing EGFP-CALCOCO1 or EGFP-CALCOCO1 mLIR + Δ623–691 grown in full medium and treated with Baf A1 as indicated were
       immunostained for endogenous p62 and LC3B. Scale bars, 5 μm.

Source data are available online for this figure.

## CALCOCO1 promotes basal autophagic flux but not bulk autophagy

In response to starvation, the number of cytoplasmic puncta formed by EGFP-CALCOCO1 increased (Fig 1H). Co-staining of the HBSS-treated cells with WIPI2 and ATG13 antibodies demonstrated co-localization of EGFP-CALCOCO1, WIPI2, and ATG13 in the cytoplasmic puncta in both FM and starvation conditions (Fig 4A and B, and Appendix Fig S2C), suggesting that CALCOCO1 is recruited to early autophagic structures.

Because of its degradation by autophagy, we speculated that CALCOCO1, like its paralogues NDP52 and TAX1BP1, could play a role in autophagy. Hence, we generated CALCOCO1 knockout (KO) HeLa and HEK293 cells by CRISPR/Cas9 (Appendix Fig S1) and investigated how the absence of CALCOCO1 affected the autophagy process. Lipidated LC3B (LC3B-II) and GABARAP are components of mature autophagosomes and are degraded together with the cargo. Hence, their abundance can be used to measure autophagy flux. Compared to WT cells, CALCOCO1 KO HeLa and HEK293 cells retained higher amounts of LC3B-II and GABARAP-II under basal conditions (Fig 4C–F). Treatment of the cells with Baf A1 led to an accumulation of comparatively equal amounts in both WT and KO cells (Fig 4C–F), suggesting that the increased basal amounts of LC3B-II and GABARAP-II in the KO cells were caused by a less efficient autophagy process.

Completion of the autophagy process can also be measured by monitoring the abundance of substrates such as selective autophagy receptors (SARs). To further test whether the absence of CALCOCO1 impaired degradation by autophagy, we monitored the turnover of some of the known SARs. Compared to WT cells, the basal levels of p62, NBR1, and NDP52 were higher in the CALCOCO1 KO HeLa and HEK293 cells (Fig 4C–F). The levels of these SARs were comparatively equal when the cells were treated with Baf A1 (Fig 4C–F), suggesting that the increased basal amounts in CALCOCO1 KO cells were caused by inefficient degradation. To clarify whether the absence of CALCOCO1 was causing the inefficiency, we reconstituted KO HeLa cells with inducible EGFP-CALCOCO1 and monitored the effect on degradation. Induced expression of EGFP-CALCOCO1 rescued the turnover of LC3B-II, GABARAP-II, p62, NBR1, and NDP52, similarly to the turnover observed in WT cells (Fig 4G–I). Taken together, these results suggest that CALCOCO1 promotes basal autophagy flux.

The observed effect on basal autophagy prompted us to test whether CALCOCO1 had a similar effect on starvation-induced bulk autophagy. Therefore, we monitored the degradation of LC3B and p62 in starved CALCOCO1 KO HeLa cells reconstituted with inducible EGFP-CALCOCO1. Under starvation conditions, p62 and LC3B degradation were similar in both induced and non-induced cells, suggesting that CALCOCO1 is not required for starvation-induced bulk autophagy (Fig 4J). To test this further, we applied mCherry-EYFP tandem tagging of LC3B to quantify and compare autophagy flux (Pankiv *et al*, 2007). EYFP is unstable while mCherry is stable in the acidic environment of the lysosomes. Hence, mCherry-EYFP-LC3B-containing autolysosomes appear as red-only puncta. Cells with more autophagy flux therefore have a higher mCherry puncta/total puncta ratio due to fusion of autophagosomes with lysosomes. When transiently transfected mCherry-EYFP-LC3B was monitored by confocal microscopy in full medium, the ratio of mCherry puncta as a percentage of the total puncta was higher in WT cells than in CALCOCO1 KO cells. Under starved conditions, however, the ratio was similar in both cells (Fig EV4A and B), solidifying the conclusion that CALCOCO1 promotes basal autophagy but not starvation-induced bulk autophagy.

Next, we monitored the degradation of LC3B-II in WT, CALCOCO1 KO, and EGFP-CALCOCO1-reconstituted CALCOCO1 KO HeLa cells after pharmacological inhibition of mTORC1 with Torin 1 (Fig 4K and L). In the WT cells and EGFP-CALCOCO1-reconstituted cells, there was a transient increase in the level of LC3B-II relative to the untreated cells. In both cell lines, co-treatment with Baf A1 resulted in an accumulation of LCB-II, indicating efficient turnover of LC3B-II by autophagy. Endogenous CALCOCO1 (left panel in Fig 4K) and EGFP-CALCOCO1 (right panel in Fig 4K) were also similarly degraded. In the CALCOCO1 KO cells, the amount of LC3B-II in the untreated cells was much higher than in the CALCOCO1 expressing cell lines, and Torin 1 treatment caused a reduction of LC3B-II. The reduction was however blocked by co-treatment with Baf A1, indicating efficient autophagic degradation of LC3B-II after Torin 1 treatment of the CALCOCO1 KO cells (Fig 4K and L). Also, p62 and NDP52 were efficiently degraded in Torin 1-treated cells (Fig 4K). Taken together, these results suggest that, while CALCOCO1 is required for basal autophagy, it is dispensable for bulk autophagy induced by either starvation or pharmacological inhibition of mTORC1.

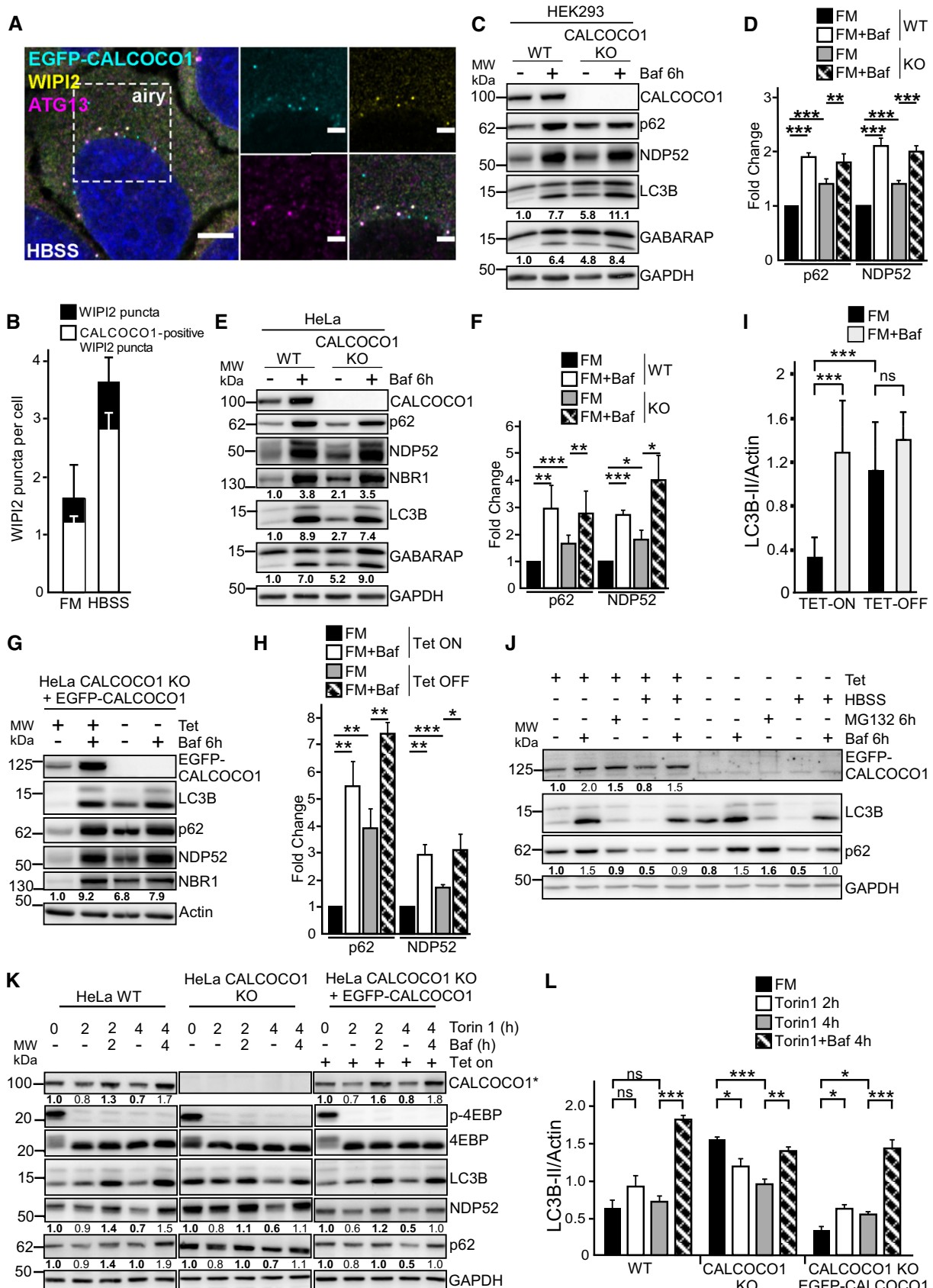

**Figure 4.**

◄

**Figure 4. CALCOCO1 promotes basal autophagic flux but not bulk autophagy.**

A, B  HeLa CALCOCO1 KO cells stably expressing EGFP-CALCOCO1 were starved for 4 h and then immunostained with anti-ATG13 and anti-WIPI2 antibodies. Scale bars in (A) are 5 μm for the confocal microscopy images and 2 μm for the airyscans. In (B), the error bars represent mean ± SD of puncta per cell from three independent experiments and 100–200 cells per each experiment.

C–L  Immunoblot analysis of indicated cell lines treated as indicated. Numbers below the blots represent relative intensity of the bands in the shown blots normalized against the loading control (GAPDH or actin). The asterisk in (K) indicates that endogenous CALCOCO1 is detected in WT and KO cell extracts and EGFP-CALCOCO1 in cells extracts from the rescued cells. In (D, F, H, I, and L), the bars represent the mean ± SD of band intensities relative to the actin or GAPDH loading control as quantified using ImageJ, $n = 5$ in (I), $n = 3$ in others. Statistical comparison was analyzed by one-way ANOVA and significance displayed as ***$P < 0.001$, **$P < 0.005$, *$P < 0.01$; ns is not significant.

## CALCOCO1 interacts with VAPA/B via a FFAT-like motif

The ER and Golgi localization of CALCOCO1 raised the possibility that it interacts with ER and Golgi-associated proteins. Hence, we performed IP experiments using EGFP-CALCOCO1 expressed stably in HEK293 cells as bait and identified the bound proteins by mass spectrometry. After stringent filtering against GFP control, about 30% of the identified proteins were either ER- or Golgi-associated (Fig 5A). Among the proteins in the CALCOCO,1 interactome was the ER tethering protein VAPA.

VAPA and VAPB are integral ER membrane proteins involved mainly in forming contacts between the ER and other membranes via interaction with proteins bearing the VAP-interacting motif called FFAT (two phenylalanines (FF) in an acidic tract (AT) using their N-terminal major sperm domain (MSP) (Murphy & Levine, 2016). The interaction is initiated by the acidic tract binding to the electro-positive surface of MSP domain and then cemented by specific interactions with the core FFAT motif. Given the perinuclear localization of CALCOCO1, VAP proteins were prime candidates for recruiting CALCOCO1 to the ER, and therefore, we focused on them in our study. To validate the interactome and test whether CALCOCO1 and VAP proteins interacted, we co-expressed EGFP-CALCOCO1 with either Myc-VAPA or Myc-VAPB in HEK293 cells and investigated their interaction by immunoprecipitation. Both Myc-VAPA and Myc-VAPB were co-precipitated by EGFP-CALCOCO1, suggesting interaction of CALCOCO1 with VAPA/VAPB in cells (Figs 5B and EV4C).

The critical residues in the MSP domain involved in the binding to FFAT motifs are K94/M96 and K87/M89 for VAPA and VAPB, respectively. The interaction can be blocked by double charge substitutions; K94D/M96D and K87D/M89D (KD/MD mutants) (Murphy & Levine, 2016). To determine whether the interaction of CALCOCO1 with VAPA and VAPB was via the FFAT motif, we made KD/MD mutants of VAPA and VAPB and tested their binding to CALCOCO1 in *in vitro* GST pull-down assays. WT GST-tagged VAPA and VAPB interacted with *in vitro*-translated Myc-CALCOCO1, suggesting direct interaction between CALCOCO1 with VAP proteins. The interaction was stronger with VAPA than with VAPB. KD/MD mutants of VAPA and VAPB abolished the interactions (Fig 5C), suggesting existence of a FFAT or a FFAT-like motif in CALCOCO1.

To test whether CALCOCO1 actually has a FFAT motif, we made deletion mutants lacking either the SKICH domain (Δ1–144), coil–coil domain (Δ145–513) or C-terminal region (Δ514–691) and investigated which region of CALCOCO1 was binding to VAP proteins. Only deletion of the C-terminal region (Δ514–691) abolished the interaction of CALCOCO1 with VAPA and VAPB (Fig 5D), suggesting involvement of the C-terminal region in the interaction. Analysis

of the CALCOCO1 primary structure in this region identified the sequence 680-FFFSTQD-686 as a potential FFAT-like motif. To test if this motif is responsible for the interaction, we made further mutations in CALCOCO1 ΔSKICH + ΔLIR, a construct strongly interacting with the VAPs (Fig 5D). Mutations of the first three residues of the predicted core FFAT-like motif (FFF/AAA) (Murphy & Levine, 2016) significantly reduced the interaction with VAPs (Fig 5D). A simultaneous deletion of the core FFAT-like motif and the flanking upstream acidic tract region (Δ671–691) strongly reduced the interactions (Fig 5D), suggesting that the FFAT-like motif was specifically mediating the interactions. Co-expression of EGFP-CALCOCO1 with either Myc-VAPA or Myc-VAPB showed perinuclear co-localization (Fig 5E), suggesting association of CALCOCO1 and VAPs in cells. Taken together, these results show that CALCOCO1 binds directly to ER integral membrane tethering proteins VAPA and VAPB via a FFAT-like motif. Transient expression of EGFP-CALCOCO1(Δ671–691) that lacks both the FFAT-like motif and the upstream acidic tract region did not alter the localization pattern of CALCOCO1 in cells (Fig EV4D). This may be a reflection of the fact that the deletion does not abolish the VAP interaction completely (Fig 5D).

## VAP proteins promote autophagy and starvation-induced degradation of tubular ER

Two recent studies showed that VAP proteins promote autophagy flux by positively augmenting the endosomal pathway and autophagosome biogenesis (Zhao *et al*, 2018; Mao *et al*, 2019). However, despite their localization in the ER membrane, whether VAP proteins play a role in ER-phagy has not been clarified. ER-phagy degrades specific sub-domains of ER in response to physiological or pathological conditions such as proteotoxic stress and nutrient starvation. Given our discovery of the interaction of VAP proteins with CALCOCO1 and its degradation by autophagy, we asked whether VAPs are degraded by autophagy and what could be their effect on ER-phagy. Consequently, we investigated how inhibition of autophagy influences turnover of ER proteins, including VAP proteins, in cultured mammalian cells. In WT MEF cells, the levels of tubular ER proteins VAPA, VAPB, and TEX264, and ER sheet protein FAM134 were reduced after 6 h of nutrient starvation. The starvation-induced decrease was however blocked when the cells were co-treated with Baf A1 (Fig 6A and B), suggesting that the reduction was due to autophagy-mediated degradation. In contrast, in autophagy-deficient Atg5 knockout (KO) MEF cells, the starvation-induced degradation of VAPA, VAPB, TEX264, and FAM134B was impaired when compared to the degradation in the WT cells (Fig 6A and B). Another protein, CLIMP63, an ER sheet protein, though not robustly degraded in the WT cells, also accumulated

slightly in the ATG5 KO cells after 6 h of nutrients starvation. These results suggest involvement of autophagy in starvation-induced degradation of the ER proteome, including VAPs, consistent with recent reports (Grumati *et al*, 2017; Smith *et al*, 2018).

The degradation of VAP proteins by autophagy and their integral ER membrane localization suggested they could play a role in ER-phagy. Depletion of both VAPA and VAPB with siRNAs in HeLa cells for 50 h increased the levels of LC3B-II, GABARAP, and

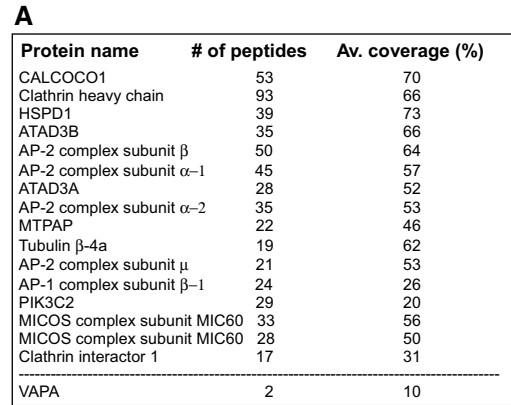

**A**

| Protein name | # of peptides | Av. coverage (%) |
|---|---|---|
| CALCOCO1 | 53 | 70 |
| Clathrin heavy chain | 93 | 66 |
| HSPD1 | 39 | 73 |
| ATAD3B | 35 | 66 |
| AP-2 complex subunit β | 50 | 64 |
| AP-2 complex subunit α−1 | 45 | 57 |
| ATAD3A | 28 | 52 |
| AP-2 complex subunit α−2 | 35 | 53 |
| MTPAP | 22 | 46 |
| Tubulin β-4a | 19 | 62 |
| AP-2 complex subunit μ | 21 | 53 |
| AP-1 complex subunit β−1 | 24 | 26 |
| PIK3C2 | 29 | 20 |
| MICOS complex subunit MIC60 | 33 | 56 |
| MICOS complex subunit MIC60 | 28 | 50 |
| Clathrin interactor 1 | 17 | 31 |
| VAPA | 2 | 10 |

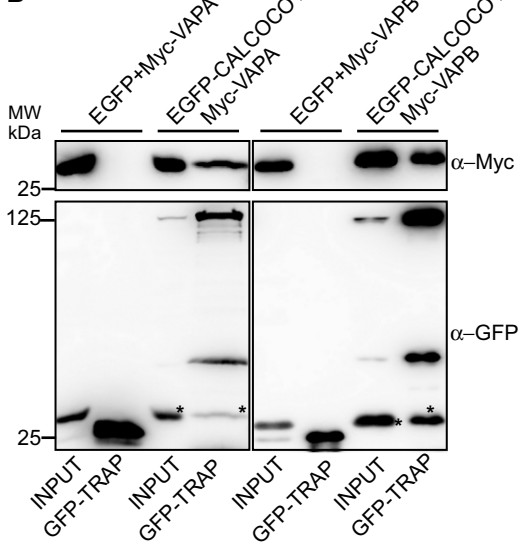

**B**

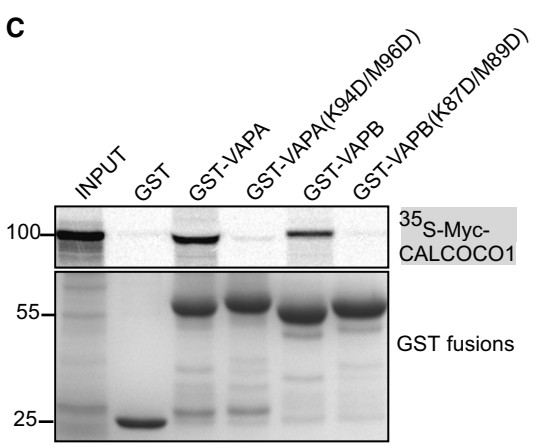

**C**

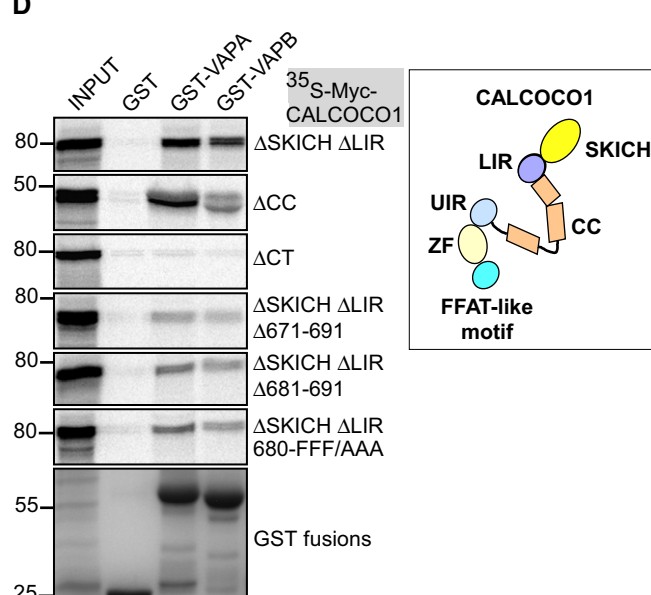

**D**

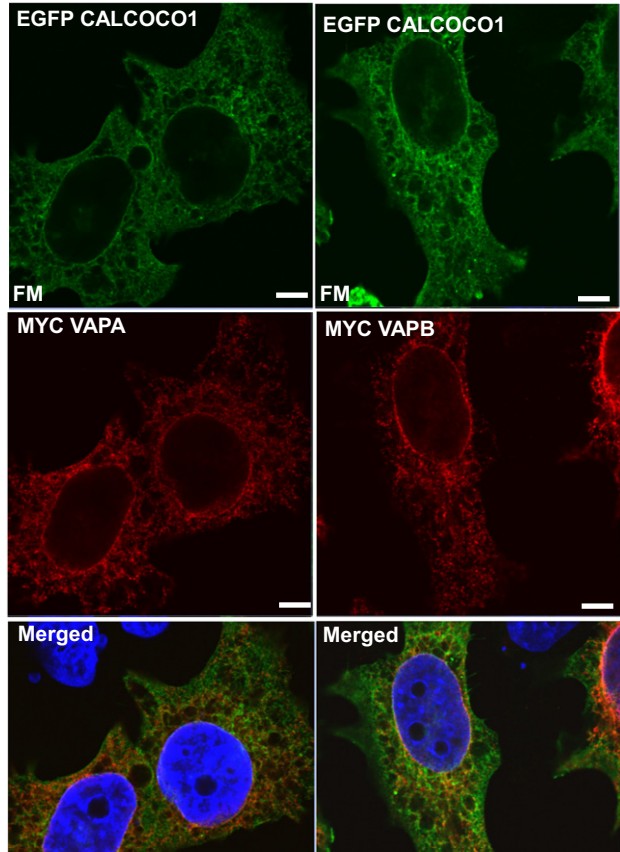

**E**

**Figure 5.**

**Figure 5. CALCOCO1 interacts with VAPA/B via a FFAT-like motif.**

A Stably expressed EGFP or EGFP-CALCOCO1 in HEK293 cells were immunoprecipitated from the cell lysates followed by mass spectrometry identification of interacting proteins. Only some of the identified proteins are shown.

B Co-IP of Myc-VAPA or Myc-VAPB with EGFP-CALCOCO1 from transiently transfected HEK293 cells. The asterisks (*) indicate Myc-VAP previously detected on the same membrane.

C, D GST pull-down assays of *in vitro* transcribed/translated $^{35}$S-Myc-CALCOCO1 constructs with indicated recombinant GST-VAPA or GST-VAPB constructs. The scheme is a representation of CALCOCO1 domains and motifs.

E HeLa cells transiently co-transfected with EGFP-CALCOCO1 and Myc-VAPA or—VAPB were immunostained with anti-Myc antibody. Scale bars, 5 μm.

CALCOCO1 relative to cells transfected with control siRNA (Fig 6C). A further accumulation of CALCOCO1 was observed when the cells were treated with Baf A1 (Fig 6D and E). From this data, we conclude that an efficient autophagic degradation of CALCOCO1 is clearly affected by the loss of VAP proteins but there is also a degradation of CALCOCO1 that may be VAP-independent, suggesting that CALCOCO1 is likely involved in more than one selective autophagy pathway. We also tested the effect of VAP depletion on RTN3 (ER tubular marker), FAM134B, and CLIMP63 (ER sheet markers) and p62. Similar to CALCOCO1, RTN3 accumulated in VAP-depleted cells while the levels of FAM134B, CLIMP63, and p62 were not affected (Fig 6D and E). Taken together, these results not only confirms recent reports (Zhao *et al*, 2018; Mao *et al*, 2019) that VAPs may promote autophagy flux, but also suggest that VAPs selectively mediate degradation of the tubular ER.

To clarify whether CALCOCO1 and VAPA/B-positive ER fragments traffic together to the autophagosomes, we assessed their co-localization with endogenous LC3B under starvation conditions in the presence of Baf A1. Co-expressed EGFP-CALCOCO1 and either Myc-VAPA or Myc-VAPB formed perinuclear and cytoplasmic puncta that co-localized with endogenous puncta of LC3B (Fig 6F), suggesting that CALCOCO1 and VAPs-positive fragments traffic together in the autophagosomes.

**CALCOCO1 is a soluble ER-phagy receptor**

In response to ER stress-causing conditions such as nutrients deprivation, the ER increases in size through increased synthesis of ER-resident proteins to augment its functions (Bernales *et al*, 2006; Wilkinson, 2019b). During recovery, ER-phagy is involved in remodeling the ER back to physiological size by sequestering ER sub-domains and excess membrane proteins for degradation in the lysosome. Given that CALCOCO1 interacts directly with transmembrane ER tethering proteins VAPA and VAPB and accumulated following their depletion (Fig 6C–E), we surmised that CALCOCO1 mediates ER-phagy in response to proteotoxic stress and nutrient starvation in conjunction with VAPA/B. CALCOCO1 fits the criteria for an ER-phagy receptor (Wilkinson, 2019b). Firstly, the interaction with VAP proteins gives it specificity for tubular ER membranes. Secondly, the strong interaction with ATG8 family proteins suggests that CALCOCO1 could not only form the platform for the recognition of the ER membranes by phagophores, but could also promote feedforward recruitment of autophagy machinery required for autophagosome biogenesis and clustering-mediated ER fragmentation.

To clarify the role of CALCOCO1 in ER turnover, we investigated how lack of CALCOCO1 influenced starvation-induced ER-phagy using CALCOCO1 KO HeLa cells. In WT cells during starvation, there was an autophagic degradation of VAPA, FAM13B, p62, and

NDP52, as indicated by their higher levels in cells treated with HBSS + BafA1 than in cells treated with HBSS only (Fig 7A and B). When compared to the WT cells, CALCOCO1 KO impaired the autophagic degradation of the tubular ER protein VAPA during starvation (Fig 7A and B), but had no effect on the starvation-induced autophagic degradation of the ER sheet marker FAM134B or autophagy receptors p62 and NDP52 (Fig 7A). Starvation-induced autophagic degradation of VAPB and CLIMP63 was not observed in this experiment (Fig 7A and B). To analyze this further, we applied tandemly tagged mCherry-EYFP-VAPA to monitor and compare the ER turnover in HeLa WT and CALCOCO1 KO cells under starvation conditions. When transiently transfected mCherry-EYFP-VAPA was monitored by confocal microscope under starved conditions, the ratio of mCherry puncta as a percentage of the total puncta was higher in WT cells than in CALCOCO1 KO cells (Fig EV5A and B), indicating that the tubular ER turnover was decreased by the absence of CALCOCO1. When a similar experiment was done with tandemly tagged mCherry-EGFP-FAM134B (82–238), a transmembrane fragment of FAM134B, the ratio was comparatively equal in both cells (Fig EV5A and B), indicating that the turnover of the sheet ER was not affected by the absence of CALCOCO1.

To enable a more controlled investigation of the role of CALCOCO1 in ER-phagy, we used the reconstituted CALCOCO1 KO cells. In non-induced cells, HBSS strongly increased the amount of tubular ER proteins RTN3, TEX264, VAPA and VAPB (Fig 7C and D). In cells expressing EGFP-CALCOCO1, HBSS treatment either reduced the amounts of these proteins (TEX264) or kept them at constant levels (Fig 7C and D). Furthermore, in EGFP-CALCOCO1 expressing cells, HBSS + BafA1-treated cells had higher levels of the tubular ER proteins than HBSS-only-treated cells (Fig 7C and D), suggesting a CALCOCO1-dependent degradation of the tubular ER proteins by autophagy during starvation. When the cells were treated with MG132 to induce proteotoxic stress, the amounts of RTN3, TEX264, VAPA, and VAPB were consistently lower in the induced cells than in the non-induced cells (Fig 7C), suggesting involvement of CALCOCO1 also in the degradation of the tubular ER during proteotoxic stress.

The behavior of the ER sheet proteins FAM134B and CLIMP63 in response to starvation was not uniform. The degradation of FAM134B in response to starvation was similar in both induced and non-induced cells, and its levels in response to proteotoxic stress were more or less equal in both induced and non-induced cells (Fig 7C), suggesting that the starvation-induced degradation of FAM134B zone of ER sheets is not mediated by CALCOCO1. On the other hand, the levels of CLIMP63 in the non-induced cells were reproducibly higher than in the induced cells in response to both starvation and proteotoxic stress (Fig 7C and D). There was however no significant accumulation of CLIMP63 after Baf A1 treatment in the induced starved cells after 6 h (Fig 7C and D).

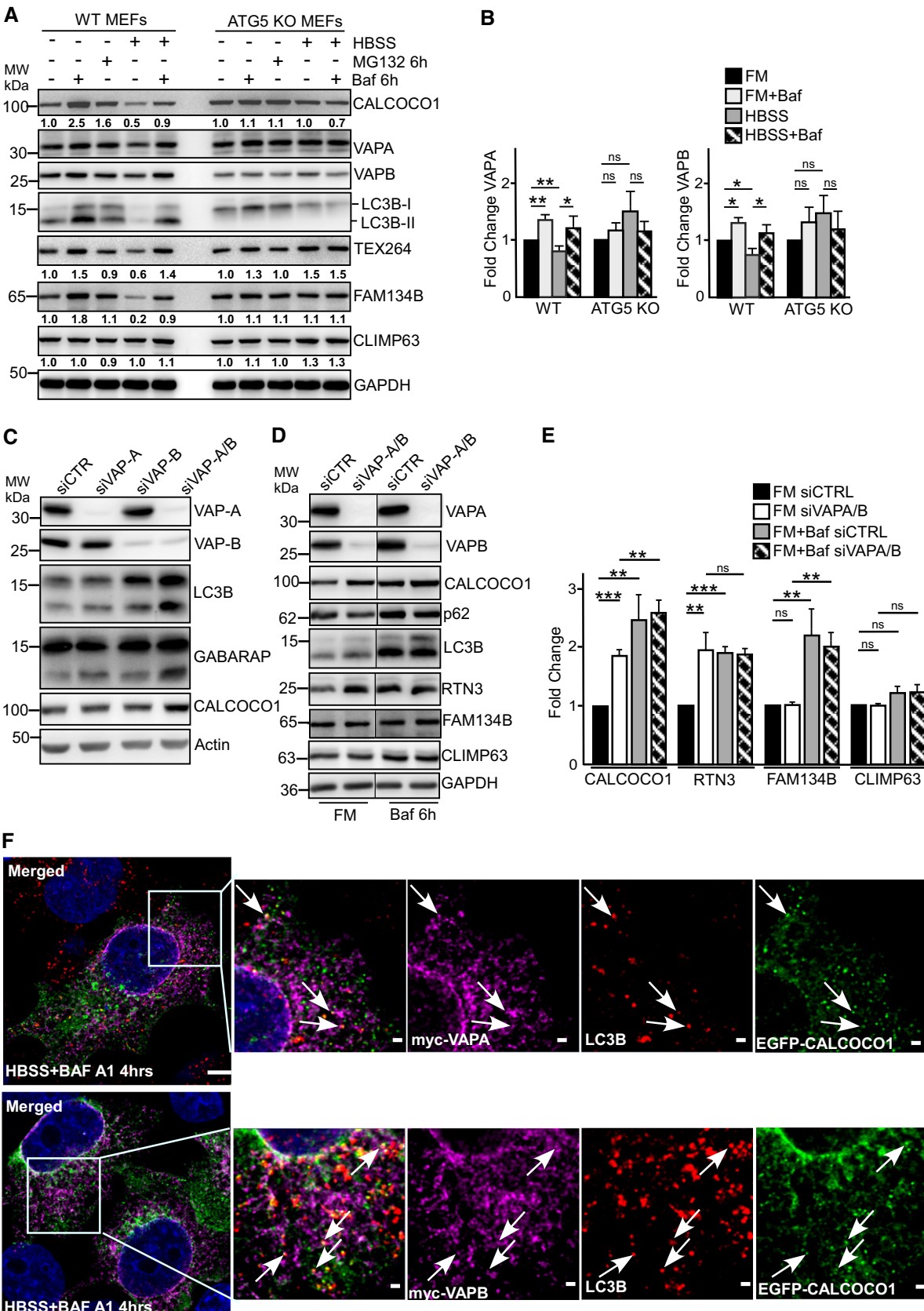

**Figure 6.**

◄

**Figure 6. VAP proteins promote autophagy and starvation-induced degradation of tubular ER.**

A, B Immunoblot analysis of indicated cell lines, starved for 6 h (HBSS) as indicated, and treated with MG132 or Baf A1 as indicated. Numbers below the blots in (A) represent relative intensity of the bands in the shown blots normalized against GAPDH loading control. In (B), the bars represent the mean ± SD of band intensities of three independent experiments as quantified using ImageJ. Statistical comparison was analyzed by one-way ANOVA and significance displayed as **$P < 0.005$, *$P < 0.01$; ns is not significant.

C Immunoblot analysis of HeLa cells transfected with the indicated siRNAs.

D, E Immunoblot analysis of HeLa cells transfected with the indicated siRNAs and treated with Baf A1 as indicated. In (D), the panels are from more than one Western blot experiment but only a single GAPDH loading control is shown. In (E), the bars represent the mean ± SD of band intensities of three independent experiments as quantified using ImageJ. Statistical comparison was analyzed by one-way ANOVA followed by Tukey multiple comparison test and significance displayed as ***$P < 0.001$, **$P < 0.005$; ns is not significant.

F HeLa KO CALCOCO1 cells were transiently co-transfected with EGFP-CALCOCO1 and Myc-VAPA or Myc-VAPB, and then immunostained with anti-Myc and anti-LC3B antibodies. Arrows indicate dots of co-localization of all three proteins. Scale bars, 5 μm for large merged images and 1 μm for zoomed images.

Source data are available online for this figure.

Because CALCOCO1 could mediate selective degradation of tubular ER, it was postulated that it could be involved in ER-phagy-mediated remodeling of ER in response to physiological stress conditions. We used immunofluorescence analysis of endogenous RTN3 to monitor the effect of CALCOCO1 on ER morphology and distribution during normal and starvation conditions. Induced expression of EGFP-CALCOCO1 for 24 h at basal conditions in HeLa cells showed reduced ER to cytoplasm ratio when compared to the non-induced cells (Fig 7E), suggesting that CALCOCO1 promoted reduction of peripheral ER. The ER-to-cytoplasm ratio when the cells were stimulated by nutrients starvation for 6 h was higher in the non-induced cells than in the EGFP-CALCOCO1-expressing cells. The ratios increased slightly when cells were treated with Baf A1, suggesting that CALCOCO1 was involved in the degradation of the peripheral ER, similarly as observed with the knockouts of other ER-phagy receptors FAM134B, CCPG1, and TEX264 (Khaminets *et al*, 2015; Smith *et al*, 2018; Chino *et al*, 2019). Immunoblotting analysis of the same cells revealed that the non-induced cells retained greater amounts of tubular ER proteins RTN3, VAPA, and VAPB at all the time points tested during starvation but not the ER sheet protein marker FAM134B (Fig 7F and G). We interpreted these results to mean that CALCOCO1 facilitates selective degradation of the tubular ER.

To define whether VAPs are required for CALCOCO1-mediated tubular ER degradation, we investigated how depletion of VAPs in the induced reconstituted cells affected ER-phagy and autophagy. Double depletion of VAPA and VAPB with siRNA impaired the EGFP-CALCOCO1-induced turnover of RTN3 under both normal and starvation conditions (Fig 7H and I). We interpreted this to mean that CALCOCO1 interacts with VAP proteins at the ER membrane to facilitate degradation of the tubular ER.

### Interaction with ATG8s is required for CALCOCO1-mediated ER-phagy

To clarify whether the CALCOCO1-mediated ER turnover was mediated by macro-autophagy, we investigated how inhibition of autophagy in the induced reconstituted cells affected ER-phagy. The cells were treated with SAR405, an inhibitor of PIK3C3, the catalytic subunit of the PI3K class III complex (Ronan *et al*, 2014). PI3KC3 is critical for the generation of PI3P that is required for autophagosome biogenesis. SAR405 treatment impaired the CALCOCO1-mediated starvation-induced degradation of RTN3, VAPA, and VAPB (Fig 8A), suggesting dependency of

the CALCOCO1-mediated degradation of the ER tubular proteins on macro-autophagy.

Selective degradation by autophagy involves a receptor bridging the cargo to the autophagosome via interaction with ATG8 family proteins (Rogov *et al*, 2014; Stolz *et al*, 2014; Johansen & Lamark, 2019). To clarify whether ATG8 protein family interaction is required for CALCOCO1-mediated degradation of tubular ER, the CALCOCO1 KO cells were reconstituted with EGFP-CALCOCO1 mLIR + Δ623–691, a deletion mutant incapable of both ATG8 protein family interaction and VAP interaction. Compared to the WT EGFP-CALCOCO1 reconstituted cells (Fig 7C), reconstitution of the CALCOCO1 KO cells with EGFP-CALCOCO1 mLIR + Δ623–691 impaired the starvation-induced degradation of RTN3 and VAPA, but not autophagy receptor p62 (Fig 8B). We interpreted this to mean that CALCOCO1 is a specific autophagy receptor for the degradation of ER tubules. The monomeric CALCOCO1 Δ145–513 construct (ΔCC) interacts with ATG8 family proteins (Fig 2D), but it failed to induce ER-phagy when reconstituted in CALCOCO1 KO cells (Fig EV5C). This indicates that the self-interaction of CALCOCO1 plays an important role in the ER-phagy process.

Autophagosomes deliver their cargo to lysosomes for degradation. We used immunofluorescence to assess delivery of CALCOCO1- and VAPA-positive autophagosomes to the lysosomes for degradation in HeLa cells. To adequately capture autophagosome–lysosome co-localization, cells were treated with Baf A1. In both basal and starved conditions, co-expressed mCherry-CALCOCO1 and Myc-VAPA formed cytoplasmic puncta that co-localized with EGFP-LAMP1, a lysosomal membrane marker (Fig 8C and D), suggesting delivery of CALCOCO1-bound VAPA-positive fragments to the lysosomes. A model of how we envision CALCOCO1 acting as a soluble ER-phagy receptor via binding to ATG8s and VAPA/B is shown in Fig 8E.

## Discussion

ER-phagy remodels ER by sequestering specific sub-domains for degradation in the lysosomes (Dikic, 2018; Wilkinson, 2019b). Recent studies have identified six mammalian ER-resident membrane proteins as ER-phagy receptors that recruit autophagy machinery to initiate autophagosome formation around the portions of the ER to be degraded (Khaminets *et al*, 2015; Fumagalli *et al*, 2016; Grumati *et al*, 2017; Smith *et al*, 2018; An *et al*, 2019; Chen *et al*, 2019; Chino *et al*, 2019). In addition, p62 acts as a soluble ER-

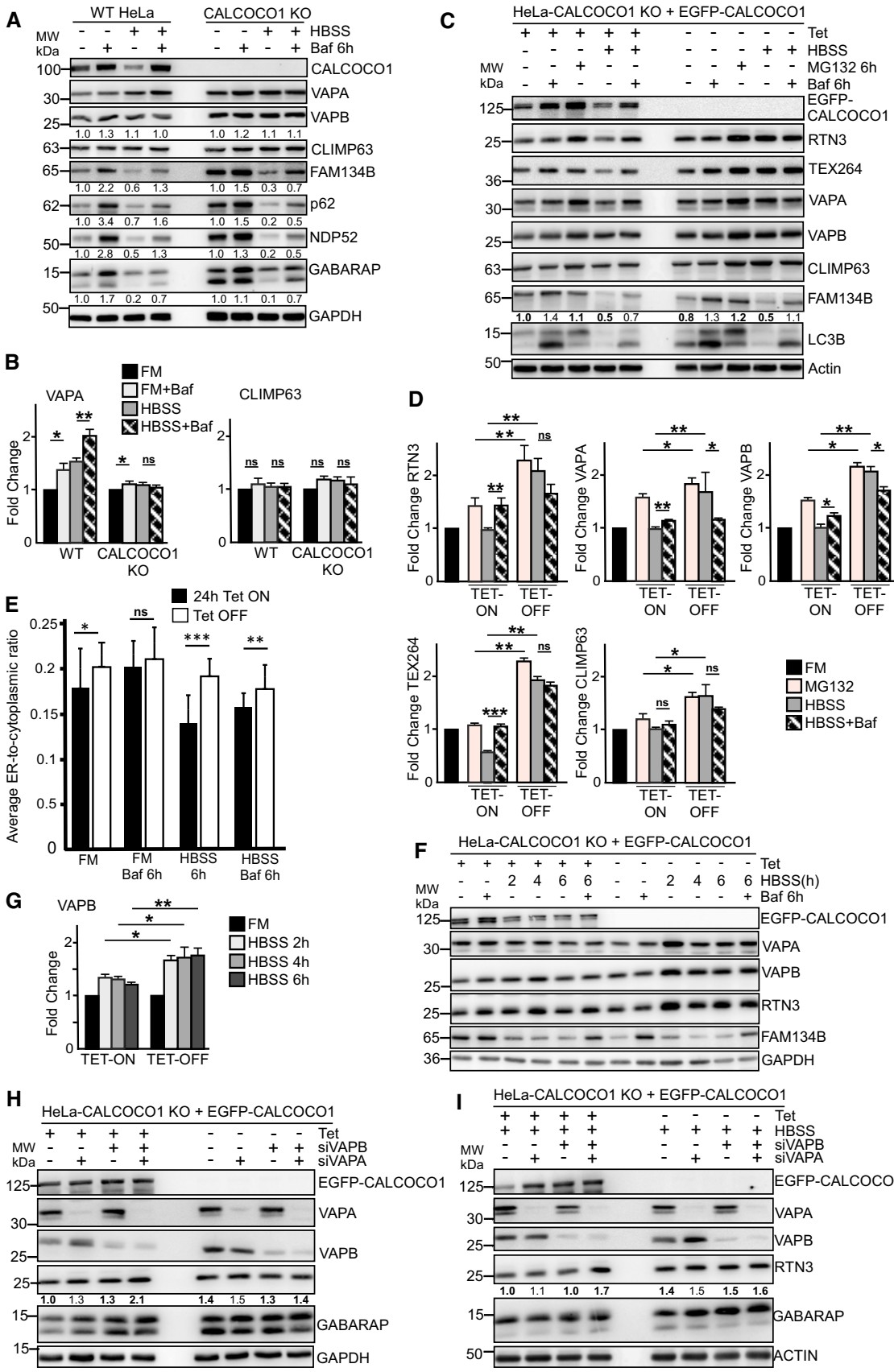

**Figure 7.**

**Figure 7. CALCOCO1 is a soluble ER-phagy receptor.**

A, B  Immunoblot analysis of HeLa WT and HeLa CALCOCO1 KO cells. The cells were starved for 6 h (HBSS) as indicated and treated with Baf A1 as indicated. Numbers below the blots in (A) represent relative intensity of the bands in the shown blots normalized against GAPDH loading control. In (A), the panels are collected from more than one Western blot experiment but for clarity, only a single GAPDH loading control is shown. In (B), the bars represent the mean ± SD of band intensities of three independent experiments as quantified using ImageJ. Statistical comparison was analyzed by one-way ANOVA followed by Tukey multiple comparison test and significance displayed as **$P < 0.005$, *$P < 0.01$; ns is not significant.

C, D  Immunoblot analysis of HeLa CALCOCO1 KO cell lines reconstituted with EGFP-CALCOCO1. Expression of EGFP-CALCOCO1 was induced or not with tetracycline, and the cells were treated with MG132 or Baf A1 as indicated. Numbers below the blots in (C) represent relative intensity of the bands in the shown blots normalized against actin loading control. In (C), the panels are collected from more than one Western blot experiment but only a single actin loading control is shown. In (D), the bars represent the mean ± SD of band intensities of three independent experiments as quantified using ImageJ. Statistical comparison was analyzed as in B and significance displayed as **$P < 0.005$, *$P < 0.01$; ns is not significant.

E  HeLa CALCOCO1 KO cell lines reconstituted with EGFP-CALCOCO1 were treated with tetracycline or not to induce expression of EGFP-CALCOCO1. Abundance of the ER was quantified from widefield fluorescence images of endogenous RTN3 staining (see Materials and Methods). Data are presented as mean ± SD of three independent experiments. Statistical comparison was analyzed as in (B) and significance displayed as ***$P < 0.001$, **$P < 0.005$, *$P < 0.01$; ns is not significant.

F, G  Immunoblot analysis of cells analyzed in (C). The bars in (G) represent the mean ± SD of band intensities relative to the loading control from three independent experiments quantified using ImageJ. Statistical comparison was analyzed by one-way ANOVA followed by Tukey multiple comparison test. Significance is displayed as **$P < 0.005$, *$P < 0.01$; ns is not significant.

H, I  Immunoblot analysis of HeLa CALCOCO1 KO cells stably expressing EGFP-CALCOCO1 in fed or starved conditions and transfected with the indicated VAPA/B siRNAs.

Source data are available online for this figure.

phagy receptor by binding to TRIM13 anchored in the ER membrane to mediate ER-phagy via the N-degron pathway (Ji *et al*, 2019). In this study, we have identified CALCOCO1 as a specific soluble ER-phagy receptor for the degradation of tubular ER in response to starvation and proteotoxic stress. The ER sub-domains are characterized by the presence of key proteins performing different functions. FAM134B, a known ER-phagy receptor, and CLIMP63 are localized in the ER sheets while RTN family proteins, TEX264 and VAPA/B preferentially localize to the ER tubules (Khaminets *et al*, 2015; Grumati *et al*, 2017; Wang *et al*, 2017). We have shown that CALCOCO1 regulates the turnover of RTN3, TEX264, VAPA, and VAPB during starvation but not ER sheet protein FAM134B. Surprisingly, there was an accumulation of CLIMP63, an ER sheet protein, in cells not expressing CALCOCO1. However, because there was no clear degradation of CLIMP63 in the EGFP-CALCOCO1-expressing cells in response to starvation and its degradation was not affected by knockdown of VAPs, the increased levels in the non-induced cells could be due to a general expansion of the peripheral ER due to ER stress. The ER stress may be brought by the accumulation of other proteins due to the absence of CALCOCO1, suggesting an important role of CALCOCO1 in overall ER homeostasis.

CALCOCO1 is unique relative to already identified receptors for having both a LIR motif and a second ATG8 family interaction motif that binds to the UDS. We named the second interacting motif UIR, since there was no UIM-like motif within the interacting region (residues 615–653) of CALCOCO1 or TAX1BP1. Also, ATG4B similarly interacts both with the LDS and the UDS (Satoo *et al*, 2009; Skytte Rasmussen *et al*, 2017), but no UIM-like motif can be identified in the region of ATG4B interacting with the UDS. Our results suggest that simultaneous interaction of the two motifs with ATG8 family proteins is required for strong and stable binding. The preferred interaction partner of CALCOCO1 is either GABARAP, GABARAPL1, or GABARAPL2. The ability to interact with ATG8 family proteins was required for its own degradation by autophagy and for CALCOCO1-mediated degradation of the ER, suggesting, in our opinion, that CALCOCO1 is an ER-phagy receptor. Consistently, CALCOCO1-mediated ER degradation was lost when autophagy was impaired by the inhibition of the PI3KC3

complex. In contrast to other ER-phagy receptors in which depletion does not affect basal autophagy flux (Wilkinson, 2019a), loss of CALCOCO1 impaired the degradation of both ATG8 family proteins and p62 in full medium, implying that CALCOCO1 promotes basal autophagy flux.

CALCOCO1 is not an ER transmembrane protein, setting it apart from most of the already identified ER-phagy receptors. Instead, our data show that CALCOCO1 is targeted to the ER by interacting with ER transmembrane proteins VAPA and/or VAPB via a FFAT-like motif. Therefore, we suggest to classify CALCOCO1 as a soluble ER-phagy receptor, perhaps akin to p62 acting with TRIM13 (Ji *et al*, 2019), to distinguish it from other ER-resident ER-phagy receptors. Loss of VAP proteins impaired CALCOCO1-mediated degradation of tubular ER, implying that the CALCOCO1-VAP interaction is required for CALCOCO1-mediated ER-phagy. These results suggest a model where CALCOCO1 bound to VAPs recruit autophagy machinery via ATG8 family to specific ER sub-domains to initiate autophagosome biogenesis and capture of the degradable cargo (Fig 8D). Emerging evidence suggests that the ER membrane fragmentation required for ER-phagy depend on ATG8 family-mediated clustering (Wilkinson, 2019a). We therefore postulate that CALCOCO1-VAP coupling mediates recognition of specific tubular ER regions by autophagy machinery via ATG8 family proteins. This in turn initiates autophagosome biogenesis in the vicinity of those regions for engulfment and degradation, consistent with the emerging notion from recent studies that receptors act upstream of the autophagy machinery (Turco *et al*, 2019). The co-localization of early phagophore markers WIPI2 and ATG13 with CALCOCO1 puncta in this study and with VAPA/B in a previous study (Zhao *et al*, 2018) supports this proposition.

Overexpression of VAPA/B causes ER punctation, and previous studies have determined that FFAT-VAP interaction may regulate ER morphology (Kaiser *et al*, 2005). Moreover, we have shown in this study that nutrient starvation upregulated ER tubular proteins RTN3, TEX264, VAPA, and VAPB. We envisage that the targeted regions of the ER are VAP-rich tubular extrusions generated by the remodeling activities of the ER in response to fluctuating conditions. The extrusions could be generated by the action of VAPs or by ER

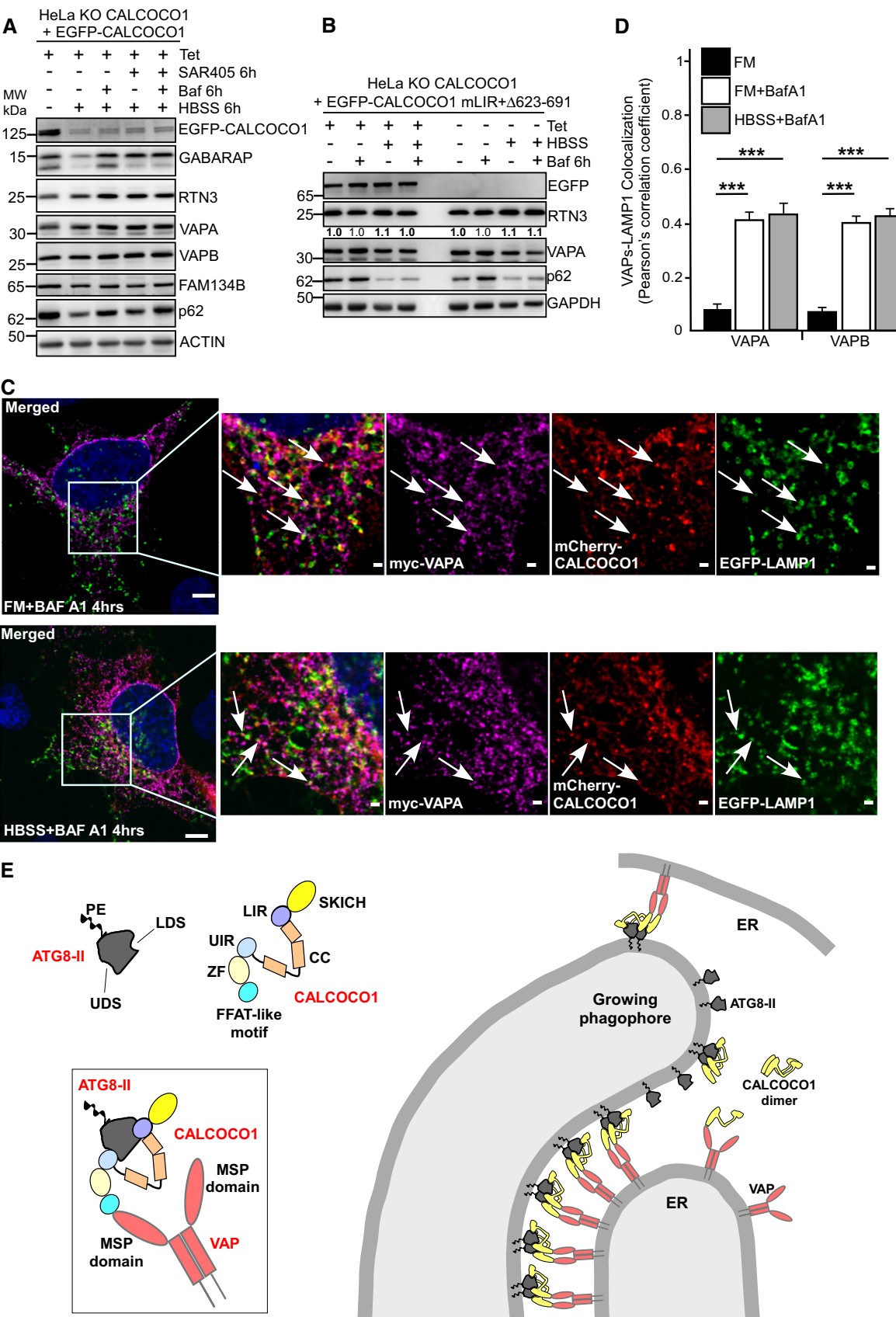

Figure 8.

**Figure 8. Interaction with ATG8 proteins is required for CALCOCO1-mediated ER-phagy.**

A Immunoblot analysis of HeLa CALCOCO1 KO cells stably expressing EGFP-CALCOCO1 in fed or starved conditions and treated as indicated with Baf A1 or PI3KC3 inhibitor SAR405.

B Immunoblot analysis of HeLa CALCOCO1 KO cells stably expressing EGFP-CALCOCO1 mLIR + Δ623–691 for 24 h or not, and then treated as indicated.

C, D Representative confocal images of HeLa CALCOCO1 KO cells transiently co-transfected with mCherry-CALCOCO1, Myc-VAPA, and EGFP-LAMP1, and then treated as indicated before immunostaining with anti-Myc antibody. Arrows indicate co-localization. Scale bars in (C), 5 μm for large merged images and 1 μm for zoomed images. In (D), data are presented as mean ± SD. Statistical comparison was analyzed by one-way ANOVA followed by Tukey multiple comparison test and significance displayed as ***$P < 0.001$; ns is not significant, $n = 3$.

E Model of ER-phagy mediated by CALCOCO1 dimers illustrating the dual LIR-LDS and UIR-UDS interaction with lipidated GABARAP subfamily proteins on the phagophore, and the FFAT-like motif-mediated interaction with MSP domains of VAP proteins on tubular ER.

Source data are available online for this figure.

---

membrane reshaping proteins. In support of this proposition, a recent study determined that VAPA and VAPB are present in interactomes of ER membrane reshaping proteins (Grumati *et al*, 2017).

Protein–protein interactions and self-interaction of receptors play critical roles in driving morphological rearrangements of the ER that is required for packaging into autophagosomes (Wilkinson, 2019a). Consistent with this notion, VAP interacts with itself via its coiled-coil domain to form homodimers and with other ER proteins via either the coiled-coil domain or the transmembrane domain to form oligomeric chains. In addition, dimerization occurs between two FFAT motifs already bound to the VAP (Kaiser *et al*, 2005; Kim *et al*, 2010). Because CALCOCO1 is dimeric, it could use its bivalent interaction with VAP dimers or oligomeric chains to target the ER membrane more tightly and recruit increased amounts of autophagy machinery proteins, resulting in increased ER morphological changes and fragmentation. Indeed we show in this study that homo-oligomerization of CALCOCO1 is required for its ER-phagy function.

The C-terminal regions of both TAX1BP1 and NDP52 contain two zinc finger domains, which mediate autophagy-critical interactions with ubiquitin, myosin VI, and galectins (Thurston *et al*, 2009; von Muhlinen *et al*, 2012; Tumbarello *et al*, 2012). In contrast, the C-terminal region of CALCOCO1, including its zinc finger domain, has not been well characterized and CALCOCO1 binds neither ubiquitin nor galectins. In this study, we have shown that the C-terminal region of CALCOCO1 contains a UIR motif that interacts with ATG8 family proteins and a FFAT-like motif that interacts with VAPA and VAPB. The FFAT (two phenylalanines in an acidic tract) motif targets proteins to the cytoplasmic face of the ER by binding to the VAP protein family. The consensus amino acid sequence of the core FFAT motif is EFFDAxE with an upstream acidic tract region (Loewen *et al*, 2003). The FFAT motif interaction with VAP involves both the acidic tract and the core FFAT motif (Furuita *et al*, 2010). The upstream acidic tract of the identified FFAT-like motif in CALCOCO1 (680-FFFSTQD-686) overlaps with the putative zinc finger domain. The core motif differs from the canonical FFAT motif at positions 1, 3, 4, and 7, consistent with other FFAT-like motifs (Mikitova & Levine, 2012). Analysis of CALCOCO1 showed that the motif is evolutionary conserved across species, implying the importance of the motif to its function.

Deletion of the FFAT-like motif in CALCOCO1 did not prevent its localization on the ER, but this may be because the deletion mutant show some interaction with the VAPs and therefore can be recruited via these weak interactions. Our model (Fig 8D) shows the interaction of CALCOCO1 with the VAPs and the ATG8 family proteins. However, it is also possible that CALCOCO1 co-operates with other proteins. Further studies are needed to address this possibility.

Besides their role in targeting peripheral proteins to the ER, FFAT-VAP interactions have been associated with cytoskeletal organization, membrane trafficking, calcium signaling, ER-associated degradation (ERAD), and autophagosome biogenesis (Murphy & Levine, 2016; Zhao *et al*, 2018). We show in this study that FFAT-mediated interaction of CALCOCO1 with VAP targets tubular ER for degradation by autophagy. An important question is how CALCOCO1-mediated ER-phagy is regulated considering there are many FFAT-containing proteins conceivably competing for VAP interaction (Murphy & Levine, 2016). It is also plausible that CALCOCO1-VAP coupling could play other roles besides ER-phagy. For instance, because VAP is part of different bridges between the ER and other organelles and CALCOCO1 is localized in the Golgi, CALCOCO1 could constitute part of ER-Golgi contact sites.

## Materials and Methods

### Antibodies

Mouse monoclonal anti-CALCOCO1 (A-10) (Santa Cruz Biotech Cat#sc-515670), rabbit polyclonal anti-CALCOCO1 (Sigma-Aldrich Cat#HPA038314), mouse polyclonal anti-CALCOCO1 (Abcam Cat# ab167237), rabbit polyclonal anti-VAPA (Proteintech Cat#15275-1-AP), rabbit polyclonal anti-VAPB (clone 4F6A6) (Proteintech Cat#66191-1-IG), rabbit polyclonal anti-GFP (Abcam Cat #ab290), mouse monoclonal anti-p62 (BD Biosciences Cat #610833), guinea pig polyclonal anti-p62 (Progen Cat #GP62-C), rabbit polyclonal anti-CALCOCO2 (Sigma-Aldrich Cat #HPA023195), rabbit anti-CALCOCO2 (Abcam Cat #ab68588), rabbit monoclonal anti-ATG7 (Cell Signaling Cat #D12B11), rabbit polyclonal anti-LC3B (Novus Bio Cat #NB100-2220), rabbit polyclonal anti-LC3B (Sigma-Aldrich Cat # L7543), mouse monoclonal anti-GABARAP (MBL Cat # M135-3), mouse monoclonal anti-Myc tag (9B11) cell signaling #2276), mouse monoclonal anti-RTN3 (F-6) (Santa Cruz Biotechnology Cat #sc-374599), rabbit polyclonal FAM134B (Proteintech Cat #21537-1-AP), mouse polyclonal anti-NBR1 (Santa cruz biotechnology #sc-130380), rabbit polyclonal anti-CKAP4 (CLIMP63) (Proteintech Cat#16686-1-AP), rabbit polyclonal anti-TEX264 (Novus Bio Cat #NBP1-89866), rabbit polyclonal anti-GAPDH (Sigma-Aldrich Cat#G9545), rabbit polyclonal anti-actin (Sigma-Aldrich Cat #A2066), HRP-conjugated goat polyclonal anti-rabbit (BD Biosciences Cat #554021), HRP-conjugated goat polyclonal anti-mouse (BD Biosciences Cat #554002).

## Reagents/Chemicals

Bafilomycin A1 (Santa Cruz Biotech sc-201550), MG132 (Sigma-Aldrich #C2211), [35S] methionine (PerkinElmer NEG709A500UC), T7-coupled reticulocyte lysate system (Promega #14610), Ponceau S (sigma #P3504), Dulbeco's modified Eagle's medium (DMEM) (Sigma-Aldrich #D6046), HBSS (Sigma-Aldrich #H9269), Hygromycin (Thermo Fisher #10687-010), Tetracycline (Sigma-Aldrich #87128), Doxycycline (Sigma-Aldrich #D9891), Pen/Strep (Sigma-Aldrich #P4333), Metafectene Pro (Biontex #T040), fetal bovine serum (FBS) (Biochrom #S0615), Lipofectamine RNAiMAX (Thermo Fisher #13778), Complete EDTA-free Protease Inhibitor (Roche #11836170001), Chemiluminescent HRP substrate (Sigma-Aldrich), GFP-TRAP (Chromotek #GA-20), Glutathione Sepharose beads (GE Healthcare #17-5132-01). The following siRNA oligonucleotides were used: CALCOCO1; 5′-GAAGCUGAGUGCAGAGAUA-3′ (Sigma), VAPA; 5′-CCUGAGAGAUGAAGGUUUA-3′ (Sigma), VAPB; 5′-GGAA GACAGUGCAGAGCAA-3′ (Sigma), Smartpool siGENOME VAPA siRNA (Dharmacon, M-021382-01-0005) and Smartpool siGENOME VAPB siRNA (Dharmacon, M-017795-00-0005).

## Plasmid constructs

All the plasmid constructs used in this study are listed in Appendix Table S1. The constructs were made using conventional cloning techniques and the Gateway recombination system (Invitrogen). Mutagenesis was performed using the QuikChange site-directed mutagenesis kit (Stratagene). Oligonucleotides for mutagenesis and sequencing were from Invitrogen. All constructs were verified by sequencing (BigDye, Applied Biosystems). pDONR201-CALCOCO1 was obtained from Harvard plasmid collection (HSCD00081507), pEGFPC1-hVAPA and pEGFPC1-hVAPB were obtained from Addgene (#104447 and #104448, respectively).

## Mammalian cell culture and cell treatments

We used human HeLa cells (ATCC CCL-2), MEFs and Atg5 KO MEFs (Kuma *et al*, 2004), and HeLa KO for all six ATG8 family genes (Y.P. Abudu, B.K. Shrestha, W. Zhang, A. Palara, H.B. Brenne, K. Larsen, D.L. Wolfson, G. Dumitriu, C.I. Øie, B.S. Ahluwalia, G. Levy, C. Behrends, S. Tooze, S. Mouilleron, T. Lamark, T. Johansen, in preparation). All cells were cultured in DMEM (Sigma-Aldrich, D6046) supplemented with 10% fetal bovine serum (Biochrom, S 0615) and 1% streptomycin–penicillin (Sigma-Aldrich, P4333) and kept in a humidified incubator at 37°C and 5% $CO_2$. Starvation experiments were conducted by incubating cells in Hanks' Balanced Salt solution (Sigma-Aldrich, H9269). Cells were treated with 1 μg/ml of tetracycline (Sigma-Aldrich), 200 ng/ml bafilomycin A1 (Santa Cruz Biotechnology, sc-201550), 25 μM MG132, for the indicated time periods. DNA transfection were done with metafectene Pro (Biontex #T040) according to manufacturers' protocol. SiRNA transfections were done with RNAiMAX according to manufacturer's protocol.

## Generation and propagation of inducible stable cell lines

Flp-In T-REx HeLa cells and Flp-In T-REx HEK293 cells were used to create inducible stable cell lines and to produce CALCOCO1 knockout cells. Tagged constructs were cloned into pcDNA5/FRT/TO

vector using the Gateway technology and then co-transfected with recombinase pOG44 into the Flp-In T-REx cells. After 48 h, colonies of cells with the gene of interest integrated into the FRT site were selected with 200 μg/ml of hygromycin (Calbiochem, 400051) and 7.5 μg/ml blasticidin. Polyclonal hygromycin-resistant cells were then expanded in the selection media and later tested for expression by immunoblotting and immunofluorescence. The expression of the gene was induced with 1 μg/ml tetracycline for 24 h. EGFP-p62 MEF cells were generated by lentiviral delivery of EGFP-p62 coding sequence in pCDNA5/LTR vector into MEF knockout cells followed by selection for stable integration with blasticidin. Viruses for infection were produced in Platinum E cells. Induction was performed with 1 μg/ml doxycycline.

## Generation of knockout cell lines using CRISPR-Cas9

Specific RNA guides were designed using the CHOPCHOP web tool (found at https://chopchop.cbu.uib.no; Labun *et al*, 2019). The sequences of the sgRNA used are 5′-CACCGGAAGAATCACCAC TAAGCC 3′, 5′-CACCGAGAAAGTTGACTCCACCAC-3′ and 5′-CACC GTTCCGATATGTGAACCGCC-3′ (see Appendix Fig S1). The sense and antisense oligonucleotides were annealed and phosphorylated and then ligated into a BbsI linearized pSpCas9(BB)-2A-Puro (PX459) vector (Addgene #62988). To generate CALCOCO1 knockout Flp-In T-REx HeLa cells, cells were transfected with the PX459 vector containing sgRNA targeting exon 2 while to generate CALCOCO1 knockout Flp-In T-REx HEK293 cells, cells were transfected with PX459 vector containing sgRNA targeting exon 2 and 4 using Metafectene Pro (Biontex #T040). Twenty-four hours post-transfection, the cells were selected by treatment with puromycin at 1 μg/ml for 72 h. Puromycin-resistant cells were then singly sorted into 96-well plates. The clones were then expanded and screened by immunoblotting. Once knockout was confirmed by immunoblotting, genomic DNA was extracted and the area of interest amplified by PCR. The amplified region was ligated into the PGEM vector (Promega #A3600) and sequenced to identify the indels.

## Western blotting

Cells were directly lysed in 2× Laemli buffer (50 mM Tris pH 7.4, 2% SDS, 10% Glycerol, 200 mM dithiothreitol (DTT, Sigma, #D0632)) and heated for 10 min. Protein concentrations were measured by Pierce BCA Protein Assay Kit (Thermo Fischer Scientific, #23227), and 30–40 μg protein of the sample was resolved by SDS–PAGE and transferred to nitrocellulose membrane. Membranes were blocked in PBS or TBS containing 0.1% Tween and 5% low fat milk and then incubated overnight at 4°C with the indicated primary antibodies in the blocking solution. Immunoblot bands were quantified using ImageJ program.

## Immunoprecipitation

Lysates from cells stably or transiently expressing EGFP or Myc-tagged proteins were immunoprecipitated by GFP/Myc-TRAP (Chromotek, # gta-20) or GST fusion immobilized on GST beads. Briefly, cells were lysed in modified RIPA buffer (50 mM Tris–HCl pH 7.4, 120 mM NaCl, 1 mM EDTA pH 8.0, 1% NP-40, 0.25% Triton X-100) supplemented with cOmplete Mini EDTA-free protease inhibitor cocktail tablets (Roche Applied Science, #11836170001) on ice for

30 min and then pelleted by centrifugation at 10,000 × *g* for 10 min. Supernatants were incubated with either GFP-TRAP, Myc-TRAP, or GST fusion protein loaded beads for 1 h or overnight at 4°C and then washed five times with RIPA buffer. Proteins were then eluted by boiling in 2× SDS sample buffer and resolved in SDS–PAGE. GFP-tagged proteins were also immunoprecipitated using the µMACS GFP Isolation Kit (Miltenyi Biotec) according to the instruction manual.

## Mass spectrometry

Gel pieces were subjected to in gel reduction, alkylation, and tryptic digestion using 6 ng/µl trypsin (V511A, Promega, Wisconsin, USA). OMIX C18 tips (Varian, Inc., Palo Alto, CA, USA) were used for sample cleanup and concentration. Peptide mixtures containing 0.1% formic acid were loaded onto a Thermo Fisher Scientific EASY-nLC1000 system and EASY-Spray column (C18, 2 µm, 100 Å, 50 µm, 50 cm). Peptides were fractionated using a 2–100% acetonitrile gradient in 0.1% formic acid over 50 min at a flow rate of 200 nl/min. The separated peptides were analyzed using a Thermo Scientific Q Exactive mass spectrometer. Data were collected in data dependent mode using a Top10 method. The raw data were processes using the MaxQuant software v1.6.0.16 using label-free quantification (LFQ) method. MS/MS data were searched against a UniProt human database. A FDR ratio of 0.01 were needed to give a protein identification. Perseus v1.6.0.7 was used for statistical analysis. The mass spectrometry proteomics data have been deposited to the ProteomeXchange Consortium via the PRIDE (Perez-Riverol *et al*, 2019) partner repository with the dataset identifier PXD018894.

## GST pull-down assays

GST fusion proteins (LC3s, GABARAPs, CALCOCO1, VAPs) were expressed in Escherichia coli SoluBL21 (DE3) (Genlantis, # C700200) in LB medium. Protein expression was induced by addition of 0.5 mM IPTG, and cells were incubated with shaking at 37°C for 4 h. Harvested cells were sonicated in the lysis buffer (20 mM Tris–HCl pH 7.5, 10 mM EDTA, 5 mM EGTA, 150 mM NaCl) and the GST-fused proteins then immobilized on Glutathione Sepharose 4 Fast Flow beads (GE Healthcare, #17-5132-01) by incubating in a rotator at 4°C for 1 h. Fusion protein-bound beads were then used directly in GST pull-down assays with *in vitro*-translated proteins. *In vitro* translation was done in the presence of radioactive $^{35}$S-methionine using the TNT T7 Reticulocyte Lysate System (Promega, #l4610). 12 µl of the *in vitro*-translated protein were then precleared by incubation with 10 µl of empty Glutathione Sepharose beads in 100 µl of NETN buffer (50 mM Tris pH 8.0, 150 mM NaCl, 1 mM EDTA, 0.5% NP-40) supplemented with cOmplete Mini EDTA-free protease inhibitor for 30 min at 4°C to remove non-specific binding. The precleared lysates were then incubated with the GST fusion protein loaded beads for 1 h at 4°C. The beads were then washed five times with NETN buffer followed by resuspension in sample loading buffer (100 mM Tris pH 7.4, 4% SDS, 20% Glycerol, 0.2% bromophenol blue and 200 mM dithiothreitol DTT (Sigma, # D0632) and boiled for 10 min and then resolved in SDS–PAGE. Gels were stained with Coomassie Brilliant Blue R-250 Dye (Thermo Fisher Scientific, #20278) for 30 min to visualize the fusion proteins, washed, and then vacuum-dried (in Saskia HochVakuum combined

with BIO-RAD Gel dryer model 583, #1651746) for 30 min. Radioactive signals were analyzed by Fujifilm bioimaging analyzer BAS-5000 (Fujifilm).

## Immunofluorescence

Cells were plated on glass coverslips (VWR, #631-0150) or in Lab-Tek chambered coverglass (Thermo Scientific, #155411) and fixed in 4% (wt/vol) formaldehyde for 10 min at room temperature and then permeabilized with 0.1% Triton X-100 in PBS at room temperature for 5 min and blocked in PBS containing 3% goat serum for 1 h at room temperature. Cells were then incubated overnight at 4°C with primary antibody diluted in PBS containing 2% goat serum. After five washes in PBS, they were incubated with Alexa Fluor secondary antibodies in PBS containing 2% goat serum for 1 h at room temperature followed by five washes in PBS. Nuclei were stained with 1 µg/ml DAPI in PBS for 10 min, followed by one final wash in PBS. Coverslips were mounted in 10 µl of Mowiol and placed on a glass microscope slide.

## Light microscopy

Cells were imaged on an Observer Z.1 inverted microscope, equipped either with an LSM880 scanner for confocal microscopy or an Axiocam 506 monochromatic camera for widefield microscopy followed by deconvolution (both systems Carl Zeiss Microscopy). Images were collected in ZEN software using a 63× NA1.4 oil immersion lens for coverslips, or a 40× NA1.2 water immersion lens for chambered coverglass. Optimal excitation and emission settings were determined using the Smart Setup function. Selected regions of interest in confocal images were further imaged with Airyscan super-resolution using optimal pixel size and *z* spacing as suggested by ZEN and processed with a strength setting of 6.0. For deconvolution microscopy, *z*-stacks were obtained with 0.1 µm step size and without camera binning, resulting in a lateral pixel spacing of 114 nm. Images were deconvolved in Huygens (Scientific Volume Imaging) ver. 19.04 using the Classic Maximum Likelihood Estimation (CMLE) algorithm with built-in theoretical point spread functions for each fluorophore. All fluorescence channels were recorded at non-saturating levels, and settings were kept identical between all samples used for comparisons or quantifications. For quantitative microscopy, images were acquired on a CellDiscoverer 7 widefield system running ZEN ver. 3.0 (Carl Zeiss Microscopy) using a 50× NA1.2 water immersion lens with a 2× optovar, and an ORCA-Flash 4.0 V3 sCMOS camera (Hamamatsu), resulting in images with lateral pixel spacing of 65 nm. For each treatment, 50–60 *z*-stacks ($z = 5$, $\Delta Z = 0.31$ µm) were recorded at randomly positioned *xy* coordinates. All camera and illumination settings were kept at identical (non-saturating) levels throughout the experiment.

## Image analysis

The abundance of ER was quantified from widefield fluorescence images of endogenous RTN3, acquired in random fashion using the tiles and position module of ZEN. For each condition analyzed, 25 regions of interest (typically containing 1,200–1,800 cells in total) were randomly distributed across each well. Cells were auto-focussed in the DAPI channel and images acquired with identical

illumination and camera settings between wells. Images were analyzed in Volocity (PerkinElmer) ver. 6.3 using a custom-made measurement protocol to segment images into populations of objects representing nuclei, total cell area, and ER. To quantify changes in ER abundance, the ratio (*ER area/total cell area*) was calculated for all images, and the average ratio reported for each treatment group.

EGFP-CALCOCO1, p62, and LC3B puncta were quantified per image for > 250 cells using a custom-made macro in Fiji (Schindelin *et al*, 2012). Briefly, EGFP-CALCOCO1-positive cell outlines were detected by automatically thresholding the EGFP channel and used to create a binary mask. Within this mask, puncta were detected and counted using find maxima in each channel. Counts were then divided by total EGFP-CALCOCO1 cell area to normalize for cell number and cell size. Finally, normalized counts were averaged for each treatment and normalized to the FM control.

To quantify co-localization between p62, LC3B, and EGFP-CALCOCO1, populations of objects representing fluorescent puncta in each channel were segmented using a custom-made protocol in Volocity ver. 6.3 (PerkinElmer). Briefly, automated puncta detection was performed by intensity thresholding, size exclusion, and noise filtering, to identify both high intensity and low intensity puncta. The overlap between EGFP-CALCOCO1 and p62 ($p62^+$ CALCOCO1 puncta), and between EGFP-CALCOCO1 and LC3B ($LC3B^+$ CALCOCO1 puncta) was identified by excluding EGFP-CALCOCO1 objects not touching the p62 or LC3B objects, respectively, from the total EGFP-CALCOCO1 population. The overlap between EGFP-CALCOCO1 and double-positive structures ($LC3B^+/p62^+$ CALCOCO1 puncta) was identified by first intersecting the LC3B and p62 populations, and then excluding EGFP-CALCOCO1 objects not touching the intersect from the total EGFP-CALCOCO1 population. The summed volume of each of the resulting $LC3B^+$, $p62^+$, and $LC3B^+/p62^+$ EGFP-CALCOCO1 populations was then divided by the summed volume of the total EGFP-CALCOCO1 population to determine the average degree of co-localization, in EGFP-CALCOCO1 puncta, for each treatment condition.

## Data availability

The mass spectrometry data from this publication have been deposited to the ProteomeXchange partner PRIDE database (https://www.ebl.ac.uk/pride/) and assigned the identifier PXD018894 (http://www.ebi.ac.uk/pride/archive/projects/PXD018894).

**Expanded View** for this article is available online.

## Acknowledgements

The technical assistance of Gry Evjen is greatly appreciated. We thank the Bioimaging core facility (KAM) at the Institute of Medical Biology (UiT—The Arctic University of Norway) for the use of instrumentation and expert assistance. This work was funded by grants from the FRIBIOMED (grant number 214448) and the TOPPFORSK (grant number 249884) programs of the Research Council of Norway to T.J.

## Author contributions

TMN designed the experiments, performed most of the experiments and analyzed the data. BKS, ES, and KBL performed imaging experiments and analyzed the data. J-AB performed mass spectrometry analyses and ZB made FAM134B reporter construct. TL supervised the project and analyzed the data. TJ supervised and co-ordinated the overall research, analyzed the data, and approved the final manuscript. TMN, TL, and TJ wrote the manuscript.

## Conflict of interest

The authors declare that they have no conflict of interest.

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
