## [Review Process File · The EMBO Journal]

CALCOCO1 acts with VAMP-Associated Proteins to mediate ER-phagy

Thaddaeus Nthiga, Birendra Kumar Shrestha, Eva Sjøttem, Jack-Ansgar Bruun, Kenneth Bowitz Larsen, Zambarlal Bhujabal, Trond Lamark, and Terje Johansen

DOI: [10.15252/embj.2019103649](https://doi.org/10.15252/embj.2019103649)

Review Timeline:

Submission Date:	7th Oct 19
Editorial Decision:	5th Nov 19
Revision Received:	4th Feb 20
Editorial Decision:	1st Mar 20
Revision Received:	1st Apr 20
Editorial Decision:	24th Apr 20
Revision Received:	1st May 20
Accepted:	6th May 20

Editor: Elisabetta Argenzio

Transaction Report:

Thank you for submitting your manuscript entitled "CALCOCO1 acts with VAMP-Associated Proteins to mediate ER-phagy" to The EMBO Journal. Your study has been sent to three referees for evaluation, whose reports are enclosed below.

As you can see, while the referees find the work potentially interesting, they also raise several key points that need to be addressed before they can support publication in The EMBO Journal. We find that the three reports are fair and balanced and require you to add new experimental data and controls, as well as appropriate quantifications and statistical analyses, in order to satisfy every request. In addition, referee #3 stresses that the role of CALCOCO1 in ER-phagy and bulk autophagy needs to be further investigated.

Given the overall interest of your study, I would like to invite you to revise the manuscript in response to the referee requests. I should also note that conclusively addressing all major and minor issues raised by the referees would be essential for publication in The EMBO Journal, as well as a strong support from the reviewers. Please feel free to contact me in case you would like to further discuss the experiments required for the revision.

When preparing your letter of response to the referees' comments, bear in mind that this will form part of the Review Process File and will be available online to the community. For more details on our Transparent Editorial Process, please visit our website:
http://emboj.embopress.org/about#Transparent_Process

Referee #1:

In this article the authors demonstrate the role CALCOCO1 in basal macroautophagy flux, more specifically involved in ER-phagy.

To reach this conclusion, they showed the homodimerization of CALCOCO1, identified the domains of interaction with AT8 proteins, particularly GABARAPs, and found an interaction with VAPA and VAPB, via conserved FFAT-like motif. They took advantage of multiple approaches including KO cells, and Tet expression of WT or fragments of proteins.

The results shown here are novel and of high interest in the field of autophagy and VAP/FFAT proteins. The methods used were rigorous and quantitative. The article is very clear and concise, with only a few typos to be corrected.

This reviewer certainly considers this article of high enough originality and quality as it is. Nevertheless, one wonders what would be the effect of a strong autophagy activation such as mediated by rapamycin or Torin-1 in cells defective for CALCOCO1.

The authors have added an important new brick to the molecular and cellular mechanisms of autophagy. Their results further suggest a link with VAPs thus possibly membrane contact sites between the ER and other membrane compartments.

Referee #2:

In this paper, the authors report on a role for the cytosolic protein CALCOCO1 as ER-phagy receptor that controls degradation of tubular ER during starvation and proteotoxic stresses. This occurs through the tethering of CALCOCO1 to the ER membrane by binding to VAPA and VAPB and engagement of the macro-autophagic machinery.

These results would be of outstanding interest for the EMBO J readership because they would highlight the first example of soluble ER-phagy receptor. However, a number of crucial issues including lack of important controls, lack of statistical analyses, data misinterpretation, or the consideration of minor differences as relevant to support the model presented in this work, led me to conclude that this manuscript is not ready for publication. I propose a rather long list of suggestions that should be addressed before taking a decision on suitability of this manuscript for publication.

- Figure 1B (MW are missing in this panel)

The finding that deletion of >50% of the polypeptide sequence abolishes formation of CALCOCO1 homomeric complexes (Fig 1B) is not surprising/uninteresting (or certainly less important than the finding shown in Fig. EV1, where CC3 is identified as the "interacting region"). The authors should consider to swap 1B with (some of) the data shown in EV1. Is homomeric complex formation required for CALCOCO1 function in ER-phagy?

- Figure 1C, D

Please write in text and legend that endogenous CALCOCO1 is examined. Here and elsewhere, quantification of the biological replicates should be shown with appropriate statistical analyses.

- Figure 1E

The authors decide to show EGFP-CALCOCO1 localization in the cis-Golgi. Is this relevant for the study? And why the Golgi distribution of CALCOCO1 is not visible in 1F and 5E, where the recombinant protein seems to be only localized in the ER?

The authors should define FM as full medium in the legend and should use FM rather than MEM in 1G.

- Figure 1G, H

As expected, the number of CALCOCO1 puncta significantly increases on nutrient deprivation + BafA1 vs. MEM + BafA1 (panel 1G). The authors should explain why the CALCOCO1 co-localization with autophagy markers substantially decreases in HBSS + BafA1 compared to MEM + BafA1 (panel 1H, where error bars are missing).

- Figure 1I

Quantification of CALCOCO1-LAMP1 positive puncta should be added (as done in 1H for co-localization with autophagy markers).

- Figure 2 (MW are missing in these panels)

Crucially, as shown in all other reports describing new ER-phagy receptors, the authors must show the association of CALCOCO1 (endogenous and recombinant) with endogenous, lipidated LC3s (and/or GABARAPs). These interactions should be abolished on deletion of the CALCOCO1's domain that mediates association with LC3/GABARAPs.

- Figure 3 (MW are missing in these panels)

Figure 3E reveals a crucial difference in behavior if one compares ectopic CALCOCO1 (not stabilized by BafA1 in cells exposed to HBSS (lane 5) with the endogenous one (Figure 1C, lane 5, stabilized by BafA1 in cells exposed to HBSS). This difference may question the use of ectopic CALCOCO1 and the extent to which the recombinant protein recapitulates the behavior of the endogenous one.

- Figure 4 (MW are missing in panel C and are wrong in panel D (EGFP-CALCOCO1 should be 125kD)).

Figure 4A and first paragraph, page 13. This is unclear. Are the authors writing that in response to starvation there is INCREASED co-localization of ectopically expressed CALCOCO1 and WIPI2 and ATG13? This should be shown by comparing FM vs. HBSS (+/-BafA1 as done for other autophagy markers in 1H) and should be quantified. The authors should then compare and comment these data with data shown in panel 1H, where the co-localization of CALCOCO1 and autophagy markers DECREASES on starvation (see comments above).

- Figure 6

The WB shown in this figure do not support the conclusion that VAPA and VAPB levels are regulated by autophagy or basal autophagy. Notably, WB is per se a semi-quantitative approach. Moreover, all quantifications shown in the manuscript (Figure 1, 2, 4, 6, 7, ...) lack indication of the statistic relevance of the data. Despite these facts, in some panels differences in protein level are visible and quite convincing. Not so in figure 6. Here, in some cases, a +10% difference in WB is considered relevant by the authors and highlighted as "supporting the model" (Figure 6A, VAPB, lanes 1 vs 2). In other cases, differences up to 60% are not commented or considered.

The experiments performed in cells depleted of both VAPA and VAPB and the conclusions reported at page 16 would imply that in these cells CALCOCO1 does not associate with the ER membrane, does not traffic in the autophagosomes and is not delivered in LAMP1-organelles. This should be checked, shown and quantified.

In Figure 6D, wild type HeLa cells with inducible VAPA expression are analyzed. Why is p62 not accumulating after 6 h treatment with BafA1 in non-induced cells? Lack of p62 accumulation is evident in the WB, and is also confirmed by the unchanged intensity value given for the polypeptide bands (average values for "more than 3 biological replicates") in cells exposed or not exposed to BafA1. Note that in Figure 1C the value of p62 was 2.5x higher in cells treated with BafA1 than in

untreated cells. BafA1 treatment also does not affect the levels of other proteins tested in this assay. This seemingly contradicts results shown in other figures.

Figure 6E is not convincing. One puncta of co-localization is shown. Moreover, by looking at the lower panel, I think that the arrow in LC3B and the arrow in EGFP-CALCOCO1 are showing two different puncta. To be more convincing, the authors should show the co-localization of these markers within LAMP1 compartments (as in Figure 1I) and quantify.

- Figure 7

The conclusion/comment at page 17, end of the first paragraph "More specifically, CALCOCO1 KO impaired starvation-induced degradation of tubular ER proteins VAPA and VAPB but not ER sheets marker FAM134B or autophagy receptor p62 (Fig 7A)." is wrong. In fact, CALCOCO1 KO also impairs starvation-induced p62 degradation (the quantification of more than 3 biological replicates gives an unchanged value of 1.2 in HBSS with and without BafA1).

The conclusion/comment at page 17, second paragraph "Compared to the non-induced cells, induced expression of EGFP-CALCOCO1 restored starvation and proteotoxic stress-induced degradation of tubular ER proteins RTN3, VAPA and VAPB (Fig 7B)." also seems wrong (or badly formulated). On expression of CALCOCO1 (as in its absence) proteotoxic stress (i.e., MG132) does not induce degradation of RTN3 (the RTN3 level actually increases, +1.4 times), VAPA (+1.3) or VAPB (+1.4). Also, starvation-induced degradation of RTN3 is very modestly affected as judged by the 20 and 10% level reduction, respectively, whereas VAPB increases upon starvation (1.3). Similar comments are valid for Figure 7D (where quantifications are missing).

The authors should quantify the values for the biological replicates and should comment on data shown in Figure 7E, namely on the reduction of EGFP-CALCOCO1 in cells depleted of VAPA and/or VAPB and on the behavior of RTN3 in these experiments.

- Figure 8

In Figure 8A, B, the control of non-induced cells, as well as the quantification of the biological replicates are missing.

Minor:

First sentence of the abstract: The ER plays ...

Phagophore is incorrectly written, page 3 and 4.

Page 5: "FAM134B also interacts with calnexin to mediate degradation of misfolded procollagen (PC) (Forrester et al., 2019)". This interaction has first been shown in Fregno et al EMBO J 2018 for degradation of misfolded ATZ.

Page 6: Please spell-out SKICH.

Page 13, line 5: "It's" in "Its".

Page 15, second paragraph, lines 10 and 12, "abolished" should be replaced with "reduced".

Referee #3:

In this manuscript, Nthiga et al. investigate the function of CALCOCO1, a paralog of the autophagy receptors, TAX1BP1 and NDP52. Localized at the ER and cis-Golgi, CALCOCO1 is degraded by autophagy. The coiled-coil domain of CALCOCO1 enables it to form homo-oligomers. CALCOCO1 and TAX1BP1 bind to both the LIR docking site (LDS) and UIM docking site (UDS) of GABARAP family proteins via a domain the authors call the "UDS interacting regions (UIR)" since both proteins do not have typical UIM-like motifs. The interaction with GABARAP family proteins is required for autophagosome targeting and autophagic degradation of CALCOCO1. The authors further found that CALCOCO1 interacts with VAPA/B, and CALCOCO1 serves as a soluble ER-phagy receptor for tubular ER rather than sheet ER.

This study can be divided into two parts. In the first part, the authors found that CALCOCO1 is degraded by autophagy and important for basal autophagy. This is a novel finding that is supported by the data. However, the second conclusion of CALCOCO1 also being able to specifically target tubular ER is not convincing, with many pieces of data lacking appropriate controls. Overall, this reviewer thinks that this study is rather preliminary.

Major concern

1. There are two major logical flaws in this study.

1-1. As long as CALCOCO1 is important for basal (general) autophagy, how can the authors conclude that CALCOCO1 is also important for selective autophagy of the ER? For this to be true, the authors should show that the magnitude of the defect in ER-phagy is greater than that in general autophagy. This also applies to the role of VAPs in ER-phagy. The authors did not perform such quantitative comparisons. The localization of CALCOCO1 on the ER does not necessarily indicate its role in ER-phagy. For example, VMP1, which also interacts with VAPs, is essential for general autophagy, not only for ER-phagy. The defect in ER-phagy observed in CALCOCO1 KO cells could be explained by the defect in general autophagy. For example, FAM134B accumulates in CALCOCO1 knockout cells even without ER-phagy stimulation (Fig. 7A). Thus, the hypothetical role of CALCOCO1 and VAPs in ER-phagy should be more vigorously investigated especially in comparison with its role in general autophagy. The difference in these two pathways could be distinguished (for example by the use of RFP-GFP-LC3 and RFP-GFP-ER protein).

1-2. The role of CALCOCO1 in tubular ER is also not clearly demonstrated. In many cases, the authors use VAPA and VAPB as tubular ER markers, but this is inappropriate. Because these proteins interact with CALCOCO1, it is natural that these proteins are degraded by autophagy together with CALCOCO1. In some cases, the authors use RTN3 as another marker of tubular ER. However, the degradation of RTN3 is not clear even in wild-type cells. The expression level of RTN3 (and also that of VAPs) actually increases during starvation in some cases (Fig. 6D, 7A, 7B). Thus, the authors' conclusion that CALCOCO1 mediates degradation of tubular ER is not valid. It seems that these results simply indicate that FAM134B is a more sensitive ER-phagy marker than RTN3 and VAPs, and demonstrate nothing about the selectivity of CALCOCO1. The authors should more directly determine whether CALCOCO1 is enriched in tubular ER and whether ER-phagy of tubular ER preferentially depends on CALCOCO1 in a more quantitative manner. The authors should use more established markers for sheet ER such as CLIMP-63.

2. The interaction between CALCOCO1 and ATG8 family proteins is determined with only in vitro experiments. As phosphorylation and other factors are often important for the recognition of selective substrates, the interaction CALCOCO1 and ATG8 family proteins and the requirement of

LIR and UIR should be tested in vivo (ideally at the endogenous level).

3. Although the requirement of CALCOCO1 in basal autophagy is demonstrated, its mechanism is unknown. It may be independent of VAPs. Some mechanistic data are required. Does CALCOCO1 interact with FIP200 to initiate autophagy or affect the mTORC1 activity?

4. The requirement of the FFAT-like motif of CALCOCO1 for ER targeting should be determined in Fig. 5E using the FFAT mutant.

5. It is important to determine whether CALCOCO1 is required for bulk autophagy during starvation, not only under basal conditions.

6. Many of the key experiments lack statistical analysis (e.g., Fig. 6 and Fig. 7).

Minor concerns

1. Fig. 1E: Localization of CALCOCO1 on the ER should be demonstrated using ER markers. Ideally, the localization of endogenous CALCOCO1 should be determined.

2. The amino acid sequences of LIR and UIR should be shown together with surrounding residues.

3. Fig. 4B and 4C need rescue experiments. Fig. 4D needs a WT cell control.

4. Fig. 5B should include negative controls (unrelated ER proteins).

5. Fig. 6A, It is unclear what the authors refer to "proteotoxic stress-induced degradation of ER proteins" in Fig. 6A. Please specify.

6. The basal level of FAM134B in CALCOCO1 KO cells increased in Fig. 7A but decreased in Fig. 7B. The authors should provide some explanation.

7. Fig. 7E should include HBSS(-) controls.

Thank you for the reviews of our manuscript EMBOJ-2019-103649 entitled "CALCOCO1 acts with VAMP-Associated Proteins to mediate ER-phagy". We are grateful for the opportunity to submit a revised version of our manuscript and thank the reviewers for their constructive criticism and helpful comments that we have used to improve our paper.

In the revised manuscript we have added new data in the form of new figure items and also revised original figure items. Hence, the revised MS contains 12 new main figure items (1D, 1F, 2I, 4B, 4G, 4H, 4I, 6B, 7B, 7D, 7H, 8D) and 2 new EV figure items (EV2D, and EV5C). We have revised 10 figure items (1I, 1J, 3E, 6A, 6D, 6E, 7A, 7C, 7I, 8B, 8C). In addition to all the new experimental data and controls we have also added appropriate quantifications and statistical analyses as requested by the reviewers. In order to better describe the experiments and results particularly related to Figures 6 and –7 we have rewritten text in the Results section making it more clear and easier to read to avoid confusion. Below we have answered all the comments made by the reviewers.

Referee #1:

In this article the authors demonstrate the role CALCOCO1 in basal macroautophagy flux, more specifically involved in ER-phagy.

To reach this conclusion, they showed the homodimerization of CALCOCO1, identified the domains of interaction with AT8 proteins, particularly GABARAPs, and found an interaction with VAPA and VAPB, via conserved FFAT-like motif. They took advantage of multiple approaches including KO cells, and Tet expression of WT or fragments of proteins.

The results shown here are novel and of high interest in the field of autophagy and VAP/FFAT proteins. The methods used were rigorous and quantitative. The article is very clear and concise, with only a few typos to be corrected.

This reviewer certainly considers this article of high enough originality and quality as it is. Nevertheless, one wonders what would be the effect of a strong autophagy activation such as mediated by rapamycin or Torin-1 in cells defective for CALCOCO1.

The authors have added an important new brick to the molecular and cellular mechanisms of autophagy. Their results further suggest a link with VAPs thus possibly membrane contact sites between the ER and other membrane compartments.

Answer: We are very happy that the referee #1 states: "The results shown here are novel and of high interest in the field of autophagy and VAP/FFAT proteins. The methods used were rigorous and quantitative. The article is very clear and concise, with only a few typos to be corrected. This reviewer certainly considers this article of high enough originality and quality as it is."

The referee wonders about the effect of rapamycin or Torin-1 in CALCOCO1 KO cells. We have performed several sets of experiments to address this and have added the new data as Fig. 4H and I. The turnover of LC3B-II by autophagy is indicated by the difference in LC3B-II in cells treated or not with bafilomycin A1 (level in Baf A1 treated cells minus level in untreated cells). In all cell lines tested (Fig 4H and I), Torin-1 inhibited mTORC1 (see dephosphorylation of 4EBP), and the turnover of LC3B-II by autophagy was observed in all the cell lines (see Fig 4I). We also tested the effect of CALCOCO1 knock out (KO) on starvation-induced bulk autophagy by treating the cells with HBSS for six hours (new Fig 4G). HBSS-induced turnover of LC3B-II was efficient in both the presence and absence of CALCOCO1. We also observed a normal degradation of autophagy receptors p62 and NDP52 in CALCOCO1 KO cells treated with Torin-1 or HBSS.

Our data show that CALCOCO1 primarily affects the turnover of LC3B-II under basal conditions, as is demonstrated by the consistent increase in the basal levels of lipidated ATG8s (e. g. LC3B-II) and soluble autophagy receptors (e.g. p62 and NDP52) in cells lacking CALCOCO1. A transient increase in LC3B-II was seen for the two cell lines expressing CALCOCO1, but these cell lines had a much lower basal level of LC3B-II. The elevated basal level of LC3B-II in CALCOCO1 KO cells may explain, at least in part, why we observed no transient increase in LC3B-II in response to Torin-1 addition (Fig 4H and I).

Referee #2:

In this paper, the authors report on a role for the cytosolic protein CALCOCO1 as ER-phagy receptor that controls degradation of tubular ER during starvation and proteotoxic stresses. This occurs through the tethering of CALCOCO1 to the ER membrane by binding to VAPA and VAPB and engagement of the macro-autophagic machinery.

These results would be of outstanding interest for the EMBO J readership because they would highlight the first example of soluble ER-phagy receptor. However, a number of crucial issues including lack of important controls, lack of statistical analyses, data misinterpretation, or the consideration of minor differences as relevant to support the model presented in this work, led me to conclude that this manuscript is not ready for publication. I propose a rather long list of

suggestions that should be addressed before taking a decision on suitability of this manuscript for publication.

Answer: We are happy that referee #2 says that: "These results would be of outstanding interest for the EMBO J readership because they would highlight the first example of soluble ER-phagy receptor." Below we addressed all comments and issues the referee has listed.

- Figure 1B (MW are missing in this panel)

Answer: We have added MW notations to Fig 1B.

The finding that deletion of >50% of the polypeptide sequence abolishes formation of CALCOCO1 homomeric complexes (Fig 1B) is not surprising/uninteresting (or certainly less important than the finding shown in Fig. EV1, where CC3 is identified as the "interacting region"). The authors should consider to swap 1B with (some of) the data shown in EV1. Is homomeric complex formation required for CALCOCO1 function in ER-phagy?

Answer: We tried to swap Fig 1B with Fig EV1A, but because of the larger size of Fig EV1A we found no satisfying way of displaying this. Therefore, the figures are not swapped in the revised MS.

We made a cell line expressing a EGFP-CALCOCO1 Δ CC construct lacking the coiled-coil region (see attached image below with full-length mCherry-CALCOCO1 transiently transfected). Since this construct was highly mislocalized and accumulated in the nucleus when stably expressed in CALCOCO1 KO cells, we therefore chose to not use this cell line in our revised MS.

- Figure 1C, D

Please write in text and legend that endogenous CALCOCO1 is examined. Here and elsewhere, quantification of the biological replicates should be shown with appropriate statistical analyses.

Answer: We have written in the revised text and figure legend that endogenous CALCOCO1 is examined, and we have added quantifications with standard deviations (new Fig 1D and 1F).

- Figure 1E

The authors decide to show EGFP-CALCOCO1 localization in the cis-Golgi. Is this relevant for

the study? And why the Golgi distribution of CALCOCO1 is not visible in 1F and 5E, where the recombinant protein seems to be only localized in the ER?

Answer: When staining for Golgi, we consistently see EGFP-CALCOCO1 on Golgi, and when this is not evident in images, this is either because Golgi is not in the focal plane, or is outside of the part of the cell included in the image. In Fig 1F (1H in revised MS), the Golgi region is not displayed in the image shown. In Fig 5E there is perinuclear accumulation of EGFP-CALCOCO1 in structures (more green than other areas) that resemble those co-localized structures we see when staining for Golgi proteins. In figure EV2E, Golgi localization is clearly seen, and in figure EV5 both WT and LIR mutated CALCOCO1 accumulate in Golgi-like structures.

The subcellular localization pattern of CALOCO1 is not known, and the co-localization of EGFP-CALCOCO1 with Golgi was very consistently observed. Several other autophagy related proteins (e. g. ATG9 and GABARAP) are in Golgi and therefore we believe that the localization of CALCOCO1 in the Golgi is an important new information relevant for the study. For these reasons, we wish to show this image.

The authors should define FM as full medium in the legend and should use FM rather than MEM in 1G.

Answer: This has been corrected.

- Figure 1G, H

As expected, the number of CALCOCO1 puncta significantly increases on nutrient deprivation + BafA1 vs. MEM + BafA1 (panel 1G). The authors should explain why the CALCOCO1 co-localization with autophagy markers substantially decreases in HBSS + BafA1 compared to MEM + BafA1 (panel 1H, where error bars are missing).

Answer: In the revised MS, we have changed the y-axis of original Fig 1G (Fig 1I in revised MS) so that it shows the total number of puncta (instead of fold increase in puncta shown in our initial figure). This does not affect any of our conclusions, but the bar graphs become more informative this way. In particular, the use of fold increase was problematic since the number of CALCOCO1 puncta in FM is very low, making relative fold increases higher for CALCOCO1 as compared to LC3B and p62.

More importantly, we have also revised Fig 1H and Fig EV2D (Fig 1J in revised MS). We analyzed more images and analyzed them in what we consider to be a more accurate way. Instead of manually selecting CALCOCO1 positive puncta, we detected the puncta automatically, and this way we strongly increased the number of puncta analyzed for each cell. We thank the reviewer for the criticism of our initial analysis. We have now clearly improved our analysis of the CALCOCO1 puncta. We have also included error bars in Fig 1J.

The reviewer asks about the substantial decrease in co-localization in HBSS + Baf compared to FM + Baf. In our new analysis (new Fig 1J), there was no difference in the co-localization with

p62 and the difference in LC3B co-localization was rather small. Hence, our new data show that there is no substantial decrease in co-localization in HBSS-treated cells.

- Figure 1I

Quantification of CALCOCO1-LAMP1 positive puncta should be added (as done in 1H for co-localization with autophagy markers).

Answer: Quantification of co-localization is added in the revised MS (new Fig EV2D).

- Figure 2 (MW are missing in these panels)

Answer: We have added MW in Fig 2A, 2B and 2I.

Crucially, as shown in all other reports describing new ER-phagy receptors, the authors must show the association of CALCOCO1 (endogenous and recombinant) with endogenous, lipidated LC3s (and/or GABARAPs). These interactions should be abolished on deletion of the CALCOCO1's domain that mediates association with LC3/GABARAPs.

Answer: We show in a new Fig 2I that recombinant GST-CALCOCO1 interacts strongly with endogenous, lipidated GABARAP from cell extracts. A construct deleted for the LIR and UIR motifs did not interact, verifying that the interaction depends on the identified motifs.

- Figure 3 (MW are missing in these panels)

Answer: We have added MW in Fig 3A.

Figure 3E reveals a crucial difference in behavior if one compares ectopic CALCOCO1 (not stabilized by BafA1 in cells exposed to HBSS (lane 5) with the endogenous one (Figure 1C, lane 5, stabilized by BafA1 in cells exposed to HBSS). This difference may question the use of ectopic CALCOCO1 and the extent to which the recombinant protein recapitulates the behavior of the endogenous one.

Answer: We cannot exclude that there may be differences between ectopically, stably expressed EGFP-CALCOCO1 and endogenous CALCOCO1 that can affect our results. When investigating autophagy in HBSS treated cells (HBSS + Baf A1 versus HBSS alone), the accumulation of the endogenous protein seems to be more easily detected in western blots than the corresponding accumulation of ectopic EGFP-CALCOCO1. We do not consider it to be a major problem since we do also consistently observe an accumulation of the ectopically expressed protein. In the revised MS we have replaced the old blot of EGFP-CALCOCO1 (WT) with a new blot that more clearly show that the ectopic protein is stabilized by Baf A1. For the new blot, the same lysates used for the old blot were used.

- Figure 4 (MW are missing in panel C and are wrong in panel D (EGFP-CALCOCO1 should be 125kD)).

Figure 4A and first paragraph, page 13. This is unclear. Are the authors writing that in response

to starvation there is INCREASED co-localization of ectopically expressed CALCOCO1 and WIPI2 and ATG13? This should be shown by comparing FM vs. HBSS (+/-BafA1 as done for other autophagy markers in 1H) and should be quantified.

Answer: We agree that the text on CALCOCO1 co-localization with WIPI and ATG13 was unclear. We have revised the text and added quantification in a new Fig 4B for the EGFP-CALCOCO1 co-localization with WIPI dots. There is a starvation-induced increase in both the number of WIPI and EGFP-CALCOCO1 puncta and a corresponding linear increase in the number of co-localized puncta. BafA1 did not affect the number of WIPI dots or the co-localization with EGFP-CALCOCO1, and therefore we did not include this data in the revised MS.

The authors should then compare and comment these data with data shown in panel 1H, where the co-localization of CALCOCO1 and autophagy markers DECREASES on starvation (see comments above).

Answer: As commented above, in our revised Figs 1I and 1J there is no evidence for a reduced co-localization with LC3B and p62 in cells treated with HBSS + Baf A1 (compared to FM + Baf A1). In our initial MS we focused on showing co-localization, and our quantitative analysis of this co-localization was not ideal. As outlined above we have performed a more accurate image analysis involving much more puncta, more time points and added statistics.

- Figure 6

The WB shown in this figure do not support the conclusion that VAPA and VAPB levels are regulated by autophagy or basal autophagy. Notably, WB is per se a semi-quantitative approach. Moreover, all quantifications shown in the manuscript (F figure 1, 2, 4, 6, 7, ...) lack indication of the statistic relevance of the data.

Answer: We apologize that in our initial text the aim of the experiment depicted in Fig 6A did not come out clearly. Since we report a role for CALCOCO1 in starvation induced ER-phagy, we wanted to test if VAPA and -B are degraded by autophagy in HBSS treated cells. Our conclusion is that the VAP proteins are degraded by autophagy in response to HBSS treatment. In our revised MS, we have added bar graphs for VAPA and VAPB (revised figure 6B). Degradation by autophagy is indicated both in the bar graphs and in the panels depicted in figure 6A. First, we show that in ATG5 KO MEFs, VAPA and -B accumulates in response to the addition of HBSS. This presumably reflects that the synthesis of VAPs is induced by starvation. Second, in WT cells there is no such increase in their levels in response to HBSS, and their levels are instead reduced. This correlates with a more efficient degradation in WT cells. Third, BafA1 stabilizes the VAPs in HBSS treated WT cells (HBSS+BafA1), indicating that the observed degradation is by autophagy. Finally, BafA1 does not similarly stabilize the VAPs in HBSS treated ATG5 KO MEFs. In conclusion, our data indicate a degradation of VAPs by autophagy in HBSS treated cells.

We have now added bar graphs (including standard deviations) in the revised figures 1D, 1F, 4I, 6B, 6F, 7B, 7D and 7G for those experiments that we consider most important to quantify. In

our initial MS and in our revised MS, several western blot experiments contain numbers under individual blots indicating the relative intensity of detected bands. These numbers indicate intensities of bands seen in the shown blot. When not showing bar graphs, we consider this to be the best way to quantify a western blot experiment, since the depicted gel and the corresponding numbers are directly compared. We consider the quantified single experiments as representative since they are all repeated several times with similar results.

Despite these facts, in some panels differences in protein level are visible and quite convincing. Not so in figure 6. Here, in some cases, a +10% difference in WB is considered relevant by the authors and highlighted as "supporting the model" (Figure 6A, VAPB, lanes 1 vs 2). In other cases, differences up to 60% are not commented or considered.

Answer: We thank the referee for commenting that some of our panels are quite convincing. We have now repeated the quantification of VAPA and -B after additional experiments (initially shown as numbers in Fig 6A) and replaced the numbers with bar graphs (revised Fig 6B). The new data supports our conclusion that the VAP proteins are degraded by autophagy in HBSS treated cells.

We also wrote in our text that VAP proteins are degraded by autophagy in FM ("Treatment of the wild type cells with Baf A1 caused accumulation of VAPA, VAPB and FAM134B (**Fig 6A**), suggesting involvement of autophagy also in their basal turnover in cells"). In the revised MS, we have added new bar graphs for VAPA and -B (Fig 6B in revised MS), and they show a significant accumulation in FM+BafA1 treated cells. However, we also see some accumulation of the VAPs in ATG5 KO MEFs, suggesting that the degradation of the VAPs in FM is not dependent on macroautophagy only. Therefore, we have now deleted this sentence from our text since our focus is on the stress-induced degradation of ER proteins.

The experiments performed in cells depleted of both VAPA and VAPB and the conclusions reported at page 16 would imply that in these cells CALCOCO1 does not associate with the ER membrane, does not traffic in the autophagosomes and is not delivered in LAMP1-organelles. This should be checked, shown and quantified.

Answer: As is shown in Fig 6C (Fig 6C and D in revised MS), efficient degradation of CALCOCO1 itself by autophagy is dependent on the VAP proteins as there is a clear accumulation of CALCOCO1 when both VAPA and VAPB are knocked down. As is seen in new Fig 6D, there is also a further accumulation of CALCOCO1 seen in BafA1-treated cells. From this data we conclude that the efficient trafficking of CALCOCO1 to autophagosomes or LAMP1 organelles is clearly affected by the loss of VAP proteins. However, there is also a degradation of CALCOCO1 that may be VAP-independent. This is not surprising as CALCOCO1 is likely involved in more than one selective autophagy pathway. In new Fig 6D we have also added RTN3 (ER tubules), FAM134B (ER sheets) and p62 and their behavior confirms the role of VAPs in the degradation of tubular ER (see modified text).

In Figure 6D, wild type HeLa cells with inducible VAPA expression are analyzed. Why is p62 not accumulating after 6 h treatment with BafA1 in non-induced cells? Lack of p62

accumulation is evident in the WB, and is also confirmed by the unchanged intensity value given for the polypeptide bands (average values for "more than 3 biological replicates") in cells exposed or not exposed to BafA1. Note that in Figure 1C the value of p62 was 2.5x higher in cells treated with BafA1 than in untreated cells. BafA1 treatment also does not affect the levels of other proteins tested in this assay. This seemingly contradicts results shown in other figures.

Answer: The lack of p62 accumulation was consistent in this cell line and a parallel cell clone as well. We do not have a good explanation for this. However, we realize that the original Fig 6D is confusing and do not add anything to our story. Therefore, we have deleted it in the revised MS.

Figure 6E is not convincing. One puncta of co-localization is shown. Moreover, by looking at the lower panel, I think that the arrow in LC3B and the arrow in EGFP-CALCOCO1 are showing two different puncta. To be more convincing, the authors should show the co-localization of these markers within LAMP1 compartments (as in Figure 1I) and quantify.

Answer: In the revised Fig (Fig 6G in revised MS), we have added more arrows and corrected the arrow that was misplaced in our initial figure (we thank the reviewer for seeing this). Co-localization with LAMP1 is shown in Fig 1K and quantified Fig EV2D and for VAPA and -B in Figs 8C and -D. The only protein that is not quantified in co-localization with LAMP1 is then LC3B that is known to co-localize with LAMP1 in Baf A1 treated cells.

- Figure 7

The conclusion/comment at page 17, end of the first paragraph "More specifically, CALCOCO1 KO impaired starvation-induced degradation of tubular ER proteins VAPA and VAPB but not ER sheets marker FAM134B or autophagy receptor p62 (Fig 7A)." is wrong. In fact, CALCOCO1 KO also impairs starvation-induced p62 degradation (the quantification of more than 3 biological replicates gives an unchanged value of 1.2 in HBSS with and without BafA1).

Answer: From studying p62 in two decades we have often observed that the combination of HBSS and Baf A1 gives variable accumulation of p62. This problem is never seen in FM. It is also not seen upon Torin 1 + Baf A1. The reason for the variability upon HBSS + Baf A1 is probably that much of p62 is quickly degraded before Baf A1 inhibits the degradation (HBSS and BafA1 are added together). There is also less translation in HBSS+BafA1 treated cells since autophagy is inhibited. We have repeated this experiment a number of times and consistently see a stabilization of p62 upon HBSS + Baf A1 in CALCOCO1 KO cells. Even in the displayed blot, we clearly see in the gel image that there is more p62 in HBSS+BafA1 treated cells than in HBSS treated cells. Since our initial panel is confusing and does not correlate with our repeated observations, we have therefore in our revised MS replaced the blot shown in our initial figure 7A with a new p62 blot where the difference is also quantified. For the revision, we also added a blot of NDP52 in our revised figure 7A, illustrating a pattern very similar to that of p62. The pattern of FAM134B is also very similar to those of p62 and NDP52, and our interpretation of the data is that HBSS induced degradation of all these three proteins is normal in CALCOCO1 KO cells.

In the revised MS we have deleted text and modified the sentence mentioned by the referee to reflect clearly what we mean.

The conclusion/comment at page 17, second paragraph "Compared to the non-induced cells, induced expression of EGFP-CALCOCO1 restored starvation and proteotoxic stress-induced degradation of tubular ER proteins RTN3, VAPA and VAPB (Fig 7B)." also seems wrong (or badly formulated). On expression of CALCOCO1 (as in its absence) proteotoxic stress (i.e., MG132) does not induce degradation of RTN3 (the RTN3 level actually increases, +1.4 times), VAPA (+1.3) or VAPB (+1.4). Also, starvation-induced degradation of RTN3 is very modestly affected as judged by the 20 and 10% level reduction, respectively, whereas VAPB increases upon starvation (1.3). Similar comments are valid for Figure 7D (where quantifications are missing).

Answer: We have now added bar graphs for the data in original Figs 7B (Fig 7D in the revised MS). We apologize to the reviewer for not explaining clearly how we interpret autophagy of ER proteins in response to starvation. This is now explained in our text. The production of ER proteins is strongly induced in response to proteotoxic stress or starvation. In the absence of ER-phagy, this causes expansion of the ER and accumulation of ER proteins. ER-phagy is induced to prevent this increase in the ER and the net effect is a level of ER proteins close to the level in FM. Hence, inhibition of ER-phagy is measured as an accumulation of ER proteins in HBSS- or MG132-treated cells. This explanatory text now starts the paragraph "CALCOCO1 is a soluble ER-phagy receptor". Induced expression of EGFP-CALCOCO1 in KO cells consistently caused starvation-induced degradation of tubular ER proteins RTN3, TEX264, VAPA and VAPB that was blocked by Baf A1 treatment (**Fig 7C and D**).

The authors should quantify the values for the biological replicates and should comment on data shown in Figure 7E, namely on the reduction of EGFP-CALCOCO1 in cells depleted of VAPA and/or VAPB and on the behavior of RTN3 in these experiments.

Answer: We have in the revised MS replaced the old figure 7E with two new figures (7H and 7I in revised MS) where VAPB and RTN3 levels are quantified. Actually, it was a Ctrl siRNA that caused an increase of the EGFP-CALCOCO1 in the original Fig 7E. We changed the Ctrl siRNA and that relieved the problem. This Ctrl siRNA is used in the new Fig 7I.

- Figure 8

In Figure 8A, B, the control of non-induced cells, as well as the quantification of the biological replicates are missing.

Answer: We have replaced the old figure 8B with a new figure 8B including non-induced cells. We did not similarly revise figure 8A to include data for non-induced cells, because our only aim here was to test if the degradation seen in induced (GFP-CALCOCO1 expressing) cells is inhibited by SAR405. We have now quantified RTN3 levels in Fig 8B.

Minor:

First sentence of the abstract: The ER plays ...

Answer: Corrected.

Phagophore is incorrectly written, page 3 and 4.

Answer: Corrected.

Page 5: "FAM134B also interacts with calnexin to mediate degradation of misfolded procollagen (PC) (Forrester et al., 2019)". This interaction has first been shown in Fregno et al EMBO J 2018 for degradation of misfolded ATZ.

Answer: Ref to Fregno et al is added in the revised MS.

Page 6: Please spell-out SKICH.

Answer: Done in the revised MS.

Page 13, line 5: "It's" in "Its".

Answer: Corrected.

Page 15, second paragraph, lines 10 and 12, "abolished" should be replaced with "reduced".

Answer: We have replaced "abolished" with "significantly reduced" and "strongly reduced".

Referee #3:

In this manuscript, Nthiga et al. investigate the function of CALCOCO1, a paralog of the autophagy receptors, TAX1BP1 and NDP52. Localized at the ER and cis-Golgi, CALCOCO1 is degraded by autophagy. The coiled-coil domain of CALCOCO1 enables it to form homo-oligomers. CALCOCO1 and TAX1BP1 bind to both the LIR docking site (LDS) and UIM docking site (UDS) of GABARAP family proteins via a domain the authors call the "UDS interacting regions (UIR)" since both proteins do not have typical UIM-like motifs. The interaction with GABARAP family proteins is required for autophagosome targeting and autophagic degradation of CALCOCO1. The authors further found that CALCOCO1 interacts with VAPA/B, and CALCOCO1 and serves as a soluble ER-phagy receptor for tubular ER rather than sheet ER.

This study can be divided into two parts. In the first part, the authors found that CALCOCO1 is degraded by autophagy and important for basal autophagy. This is a novel finding that is supported by the data. However, the second conclusion of CALCOCO1 also being able to

specifically target tubular ER is not convincing, with many pieces of data lacking appropriate controls. Overall, this reviewer thinks that this study is rather preliminary.

Major concern

1. There are two major logical flaws in this study.

1-1. As long as CALCOCO1 is important for basal (general) autophagy, how can the authors conclude that CALCOCO1 is also important for selective autophagy of the ER? For this to be true, the authors should show that the magnitude of the defect in ER-phagy is greater than that in general autophagy.

Answer: We have now added new experiments investigating whether CALCOCO1 has a role in starvation-induced bulk autophagy and Torin 1-induced autophagy (Fig 4G, H and I). We specifically tested the effect of CALCOCO1 on the turnover of LC3B-II in response to HBSS or Torin-1 and found that the absence of CALCOCO1 did not affect these processes. Our study indicate a basal autophagy role for CALCOCO1 in full medium, but not in Torin-1 or starvation induced autophagy. The effect of CALCOCO1 on ER-phagy during starvation therefore cannot be due to effects on general autophagy.

This also applies to the role of VAPs in ER-phagy. The authors did not perform such quantitative comparisons. The localization of CALCOCO1 on the ER does not necessarily indicate its role in ER-phagy. For example, VMP1, which also interacts with VAPs, is essential for general autophagy, not only for ER-phagy.

Answer: We show in our new Fig 7I (revised MS) that VAP proteins are needed for CALCOCO1 mediated degradation of RTN3 during starvation. Revised text: “To define whether VAPs are required for CALCOCO1-mediated tubular ER degradation, we investigated how depletion of VAPs in the induced reconstituted cells influences ER-phagy. Double depletion of VAPA and VAPB with siRNA impaired the EGFP-CALCOCO1-induced turnover of RTN3 under both normal and starvation conditions (**Fig 7H and I**). We interpreted this to mean that CALCOCO1 interacts with VAP proteins at the ER membrane to facilitate degradation of the tubular ER.”

The defect in ER-phagy observed in CALCOCO1 KO cells could be explained by the defect in general autophagy. For example, FAM134B accumulates in CALCOCO1 knockout cells even without ER-phagy stimulation (Fig. 7A).

Answer: The accumulation of FAM134B in CALCOCO1 KO cells is observed in full medium. Starvation induced degradation of FAM134B was similar in WT and KO cells. Our reasoning is that the accumulation of FAM134B under basal conditions could be due to long term effects caused by inefficient basal autophagy (FAM134B has functional LIR motif). Note that the band pattern of FAM134B in figure 7A is similar to those of the autophagy receptors NDP52 and p62.

Thus, the hypothetical role of CALCOCO1 and VAPs in ER-phagy should be more vigorously investigated especially in comparison with its role in general autophagy. The difference in these two pathways could be distinguished (for example by the use of RFP-GFP-LC3 and RFP-GFP-ER protein).

Answer: We have indeed considered this type of assay when planning experiments for this study. It is very challenging to find a quantitative reporter assay comparing effects on general autophagy and ER-phagy because they share a common pathway involving LC3B. We consider the double tag to be a better tool for monitoring degradation of single proteins, than to compare the efficiency of different autophagy pathways. As explained above, CALCOCO1 does not alter degradation of LC3B during starvation. It only alters degradation of selected ER proteins. It also does not alter degradation of p62 or NDP52 during starvation. In Fig 4G we show that during starvation, degradation of endogenous LC3B is not affected by the absence of CALCOCO1, but as seen in Fig 7C and D, the degradation of ER proteins VAPA, VAPB, TEX264 and RTN3 are affected. Hence, we feel certain that measuring levels of the endogenous proteins directly is a better experimental strategy than the tandem tag reporter assay (depending on overexpression).

1-2. The role of CALCOCO1 in tubular ER is also not clearly demonstrated. In many cases, the authors use VAPA and VAPB as tubular ER markers, but this is inappropriate. Because these proteins interact with CALCOCO1, it is natural that these proteins are degraded by autophagy together with CALCOCO1. In some cases, the authors use RTN3 as another marker of tubular ER. However, the degradation of RTN3 is not clear even in wild-type cells. The expression level of RTN3 (and also that of VAPs) actually increases during starvation in some cases (Fig. 6D, 7A, 7B). Thus, the authors' conclusion that CALCOCO1 mediates degradation of tubular ER is not valid. It seems that these results simply indicate that FAM134B is a more sensitive ER-phagy marker than RTN3 and VAPs, and demonstrate nothing about the selectivity of CALCOCO1. The authors should more directly determine whether CALCOCO1 is enriched in tubular ER and whether ER-phagy of tubular ER preferentially depends on CALCOCO1 in a more quantitative manner. The authors should use more established markers for sheet ER such as CLIMP-63.

Answer: In our revised MS, we have added bar graphs in Fig 7 (new Fig 7B and D) showing the effect of CALCOCO1 on HBSS- or MG132-induced degradation of ER proteins. These quantitative data confirm our previous conclusion that CALCOCO1 is needed for efficient degradation of tubular ER proteins, while the degradation of FAM134B is independent of CALCOCO1.

In our revised MS, we have also included two additional ER markers, i. e. the tubular ER protein TEX264 and the sheet ER marker CLIMP63. Testing degradation of TEX264, we confirmed an important role of CALCOCO1 in degradation of tubular ER proteins. However, our data with CLIMP63 did not confirm our initial conclusion that CALCOCO1 is not involved in the degradation of ER sheets. In figure 7A and -B, we failed to detect any HBSS induced degradation of CLIMP63, but our data in figure 7C and -D indicate that CALCOCO1 may play a role in HBSS-induced autophagy of CLIMP63, although CALCOCO1 expression had a

stronger effect on the degradation of tubular ER proteins like TEX264 and RTN3 (see Fig 7D). In summary, we cannot rule out that CALCOCO1 might play some role in the degradation of ER sheets, but the importance of CALCOCO1 is most evident for tubular ER proteins.

In the text in our revised MS we have described more clearly how we analyze ER-phagy in the experiment depicted in figure 7C, and our data clearly supports a role for CALCOCO1 in the degradation of tubular ER proteins. The experiment was done using CALCOCO1 KO cells reconstituted with inducible EGFP-CALCOCO1 with or without induction. First, we show that in the non-induced cells, VAPA and -B, RTN3, and TEX264 accumulate in response to either addition of HBSS or MG132 treatment (to induce proteotoxic stress). This presumably reflects that the synthesis of these tubular proteins is induced by starvation and proteotoxic stress (See Bernales et al. 2006. Autophagy counterbalances endoplasmic reticulum expansion during the unfolded protein response. PLOS Biol. 4, 2311-2324). In the induced cells however, there is no such increase in their levels in response to either HBSS or MG132 treatment. Instead, their levels are either reduced or comparatively equal to basal levels. This correlates with a more efficient degradation in the presence of induced expression of EGFP-CALCOCO1. Actually, BafA1 causes accumulation of these proteins in HBSS treated induced cells (HBSS+BafA1), indicating that the observed degradation is by autophagy. In our revised MS, we have also included bar graphs in a new figure 7D, supporting the data shown in figure 7C.

2. The interaction between CALCOCO1 and ATG8 family proteins is determined with only in vitro experiments. As phosphorylation and other factors are often important for the recognition of selective substrates, the interaction CALCOCO1 and ATG8 family proteins and the requirement of LIR and UIR should be tested in vivo (ideally at the endogenous level).

Answer: We co-precipitated endogenous, lipidated GABARAP with recombinant GST-CALCOCO1 (new Fig 2I in revised MS). In contrast, endogenous GABARAP could not be co-precipitated with the LIR/UIR mutated GST-CALCOCO1. We tried to immunoprecipitate endogenous CALCOCO1 with endogenous GABARAP from cell extracts using the available antibodies, but did not succeed.

3. Although the requirement of CALCOCO1 in basal autophagy is demonstrated, its mechanism is unknown. It may be independent of VAPs. Some mechanistic data are required. Does CALCOCO1 interact with FIP200 to initiate autophagy or affect the mTORC1 activity?

Answer: It is important, but beyond the scope of this manuscript to establish, on a mechanistic level, the roles displayed by CALCOCO1 in basal autophagy. We agree that it may be independent of VAPs. We have tested several of the core autophagy proteins (other than the

ATG8s) for an interaction with CALCOCO1, but we found no interaction with FIP200, ULK1 (see enclosed figure below) or other proteins that we have tested.

4. The requirement of the FFAT-like motif of CALCOCO1 for ER targeting should be determined in Fig. 5E using the FFAT mutant.

Answer: When expressing GFP-CALCOCO1 deleted for its FFAT-like motif, we observed no consistent difference in localization pattern between the WT construct and the FFAT motif deleted construct. The deletion of the FFAT-like motif did not affect the perinuclear and Golgi localization of GFP-CALCOCO1, and further studies are needed to establish the importance of the FFAT motif for CALCOCO1 ER localization.

5. It is important to determine whether CALCOCO1 is required for bulk autophagy during starvation, not only under basal conditions.

Answer: In our revised MS we have included new data in figures 4G and 4H/I indicating that CALCOCO1 has no noticeable effect on HBSS or Torin-1 induced bulk autophagy (measured as the turnover of LC3B) under the tested conditions. A loss of CALCOCO1 causes accumulation in full medium of ATG8s and autophagy receptors like p62 and NDP52, but all our data seems to indicate that the turnover of LC3B-II and degradation of p62 or NDP52 in response to HBSS or Torin-1 is normal in cells lacking CALCOCO1.

6. Many of the key experiments lack statistical analysis (e.g., Fig. 6 and Fig. 7).

Answer: In our revised MS we have added bar graphs including standard deviation for western blot experiments (new figures 1D, 1F, 4I, 6B, 7B, 7D) and we have also included new quantification of cell experiments (EV2D, 4B, 8D). We have included quantification for all those experiments that we consider to be key.

Minor concerns

1. Fig. 1E: Localization of CALCOCO1 on the ER should be demonstrated using ER markers. Ideally, the localization of endogenous CALCOCO1 should be determined.

Answer: None of the available antibodies for endogenous CALCOCO1 was specific in cells. However, using different tags on CALCOCO1, we observed fairly similar localization pattern in cells indicating that the tag did not significantly affect cellular localization. We have shown localization of CALCOCO1 with VAP proteins on the ER in Figures 5E, 6G and 8C in the revised MS.

2. The amino acid sequences of LIR and UIR should be shown together with surrounding residues.

Answer: In the revised text (end of paragraph “CALCOCO1 binds directly to ATG8 family proteins with preference for the GABARAP subfamily”), we have added the sequence of LIR (including surrounding residues) and the sequence of UIR.

3. Fig. 4B and 4C need rescue experiments. Fig. 4D needs a WT cell control.

Answer: Figure 4E in our revised MS show rescue experiment in CALCOCO1 KO cells reconstituted with EGFP-CALCOCO1. In addition, in our new figure 4H, analyses of HeLa WT, HeLa CALCOCO1 KO and HeLa CALCOCO1 KO cells rescued with EGFP-CALCOCO1 are shown. Elsewhere in the MS, we have either shown WT or CALCOCO1 KO cells or rescued cells before and after the induction of GFP-CALCOCO1 expression.

4. Fig. 5B should include negative controls (unrelated ER proteins).

Answer: A new control experiment (figure EV5C) has been added where we show an interaction of GST-CALCOCO1 with VAPA, but not with SEC13 or KDELR1.

5. Fig. 6A, It is unclear what the authors refer to "proteotoxic stress-induced degradation of ER proteins" in Fig. 6A. Please specify.

Answer: We then refer to the accumulation of ER proteins in response to the addition of MG132. This has been specified in the text of the revised MS (See also the reference to Bernales et al. 2006 given above).

6. The basal level of FAM134B in CALCOCO1 KO cells increased in Fig. 7A but decreased in Fig. 7B. The authors should provide some explanation.

Answer: We consistently observed that the basal level of FAM134B was elevated in rescued cells with induced EGFP-CALCOCO1 expression (figure 7C, 7F in the revised MS). This seemingly contradicts the increased FAM134B level seen in the selected CALCOCO1 KO clone analyzed (figure 7A), but it is possible that the lack of CALCOCO1 over time is responsible for the observed increase in FAM134B. It is beyond the scope of this manuscript to investigate further the relation between CALCOCO1 and FAM134B under basal conditions, and in particular, since stress induced degradation of FAM134B does not seem to depend on CALCOCO1.

7. Fig. 7E should include HBSS(-) controls.

Answer: The old figure 7E has been deleted, and replaced by new figures 7H and 7I containing more data including HBSS(-) controls (figure 7H).

We thank the reviewers for their constructive criticisms and very insightful comments, which have allowed us improve our manuscript by further probing several aspects of the study and its conclusions.

Thank you for submitting a revised version of your manuscript. It has now been seen by the original referees whose comments are shown below.

As you will see, while referee #1 finds that his/her criticisms have been sufficiently addressed and recommend the manuscript for publication, referee #2 and #3 still point out inconsistencies and overstatements, and also ask you to provide additional data to fully support the key findings. Furthermore, they request proper quantification and statistical analyses of the data.

We agree with the referees that these are important points that have to be addressed before pursuing publication of this manuscript.

Referee #1:

The authors have satisfactorily and fully answered the reviewers' requests.

Referee #2:

The manuscript has been improved. Inconsistent results shown in the previous version of the manuscript have disappeared (in some cases the authors modified their presentation to "make graphs more informative", in other cases they shifted from manual to automatized quantification to "analyze the data in what we consider to be a more accurate way", in other cases they "performed quantitative analyses more carefully", or panels have been replaced).

However, it is my opinion that the paper cannot be accepted as it is. There are mistakes in the figures, some inconsistency remains and should be explained, some "intriguing" result is not commented at all. There still are gels lacking MW markers, individual WB panels have been replaced but the loading controls remain those of the old panel (appropriate loading controls will be shown in the uncropped source images?).

I have added "NEW COMMENTS" after the authors' responses to my original points.

OLD COMMENT: The finding that deletion of >50% of the polypeptide sequence abolishes formation of CALCOCO1 homomeric complexes (Fig 1B) is not surprising/uninteresting (or certainly less important than the finding shown in Fig. EV1, where CC3 is identified as the "interacting region"). The authors should consider to swap 1B with (some of) the data shown in EV1. Is homomeric complex formation required for CALCOCO1 function in ER-phagy?

Answer: We tried to swap Fig 1B with Fig EV1A, but because of the larger size of Fig EV1A we found no satisfying way of displaying this. Therefore, the figures are not swapped in the revised MS. We made a cell line expressing a EGFP-CALCOCO1 Δ CC construct lacking the coiled-coil region (see attached image below with full-length mCherry-CALCOCO1 transiently transfected). Since this construct was highly mislocalized and accumulated in the nucleus when stably expressed in CALCOCO1 KO cells, we therefore chose to not use this cell line in our revised MS.

NEW COMMENT: This does not seem to answer our request. We are referring at the Δ CC3 protein (corresponding to myc- Δ 413-513), and not at the Δ CC protein (Δ 145-513) mentioned in the response. An easy solution would be to repeat the experiment shown in Fig. 1B (i.e., transient transfection of HEK293 cells followed by co-IP) by adding a lane showing that the Δ CC3 protein does not co-IP Myc-CALCOCO1. And then my question was if formation of the homomeric complex is required for the putative function of CALCOCO1 in ER-phagy (for instance, it is not required to associate with ATG8s, Fig. 2D).

OLD COMMENT: Figure 1G, H As expected, the number of CALCOCO1 puncta significantly increases on nutrient deprivation + BafA1 vs. MEM + BafA1 (panel 1G). The authors should explain why the CALCOCO1 co-localization with autophagy markers substantially decreases in HBSS + BafA1 compared to MEM + BafA1 (panel 1H, where error bars are missing).

Answer: In the revised MS, we have changed the y-axis of original Fig 1G (Fig 1I in revised MS) so that it shows the total number of puncta (instead of fold increase in puncta shown in our initial figure). This does not affect any of our conclusions, but the bar graphs become more informative this way. In particular, the use of fold increase was problematic since the number of CALCOCO1 puncta in FM is very low, making relative fold increases higher for CALCOCO1 as compared to LC3B and p62.

More importantly, we have also revised Fig 1H and Fig EV2D (Fig 1J in revised MS). We analyzed more images and analyzed them in what we consider to be a more accurate way. Instead of manually selecting CALCOCO1 positive puncta, we detected the puncta automatically, and this way we strongly increased the number of puncta analyzed for each cell. We thank the reviewer for the criticism of our initial analysis. We have now clearly improved our analysis of the CALCOCO1 puncta. We have also included error bars in Fig 1J. The reviewer asks about the substantial decrease in co-localization in HBSS + Baf compared to FM + Baf. In our new analysis (new Fig 1J), there was no difference in the co-localization with p62 and the difference in LC3B co-localization was rather small. Hence, our new data show that there is no substantial decrease in co-localization in HBSS-treated cells.

NEW COMMENT: Authors should specify "puncta per cell" on y-axis in Fig.1I-J. What do error-bars represent? And what about significance? The authors should specify in the legend that an automated system to quantify puncta has been used.

OLD COMMENT: Crucially, as shown in all other reports describing new ER-phagy receptors, the authors must show the association of CALCOCO1 (endogenous and recombinant) with endogenous, lipidated LC3s (and/or GABARAPs). These interactions should be abolished on deletion of the CALCOCO1's domain that mediates association with LC3/GABARAPs.

Answer: We show in a new Fig 2I that recombinant GST-CALCOCO1 interacts strongly with

endogenous, lipidated GABARAP from cell extracts. A construct deleted for the LIR and UIR motifs did not interact, verifying that the interaction depends on the identified motifs.

NEW COMMENT: Panel 2I shows that the ratio lipidated/non-lipidated GABARAP associated with GST (negative control) and with CALCOCO1 does not change. Based on current knowledge on ER-phagy receptors, these bind the lipidated forms of the ATG8 proteins. The fact that CALCOCO1 does not seem to distinguish GABARAP-I and -II (and does not show preferential binding to the latter) is not commented. This analysis should also be done to monitor the association of LC3 forms. Is "lack of preference for the lipidated forms of ATG8s" an intrinsic property of CALCOCO1? This would distinguish it from ER-phagy receptors.

OLD COMMENT: Figure 3E reveals a crucial difference in behavior if one compares ectopic CALCOCO1 (not stabilized by BafA1 in cells exposed to HBSS (lane 5) with the endogenous one (Figure 1C, lane 5, stabilized by BafA1 in cells exposed to HBSS). This difference may question the use of ectopic CALCOCO1 and the extent to which the recombinant protein recapitulates the behavior of the endogenous one.

Answer: We cannot exclude that there may be differences between ectopically, stably expressed EGFP-CALCOCO1 and endogenous CALCOCO1 that can affect our results. When investigating autophagy in HBSS treated cells (HBSS + Baf A1 versus HBSS alone), the accumulation of the endogenous protein seems to be more easily detected in western blots than the corresponding accumulation of ectopic EGFP-CALCOCO1. We do not consider it to be a major problem since we do also consistently observe an accumulation of the ectopically expressed protein. In the revised MS we have replaced the old blot of EGFP-CALCOCO1 (WT) with a new blot that more clearly show that the ectopic protein is stabilized by Baf A1. For the new blot, the same lysates used for the old blot were used.

NEW COMMENT: Having changed the relevant panel in Fig. 3E, the authors should note that the loading control (Actin?) is from another gel/WB. Since the experiment has been repeated many times and the results were consistent, the "n" should be given and the significance of the variations should be shown.

OLD COMMENT: Figure 4 (MW are missing in panel C and are wrong in panel D (EGFP-CALCOCO1 should be 125kD)). Figure 4A and first paragraph, page 13. This is unclear. Are the authors writing that in response to starvation there is INCREASED co-localization of ectopically expressed CALCOCO1 and WIPI2 and ATG13? This should be shown by comparing FM vs. HBSS (+/-BafA1 as done for other autophagy markers in 1H) and should be quantified.

Answer: We agree that the text on CALCOCO1 co-localization with WIPI and ATG13 was unclear. We have revised the text and added quantification in a new Fig 4B for the EGFP-CALCOCO1 co-localization with WIPI dots. There is a starvation-induced increase in both the number of WIPI and EGFP-CALCOCO1 puncta and a corresponding linear increase in the number of co-localized puncta. BafA1 did not affect the number of WIPI dots or the co-localization with EGFP-CALCOCO1, and therefore we did not include this data in the revised MS.

NEW COMMENT: The new Fig. 4B must be wrong! The authors show that there is (in average) 1

WIPI punctum per cell? Moreover, the authors are quantifying in FM and HBSS. Fig. 4A should therefore show both conditions.

OLD COMMENT: Figure 6 The WB shown in this figure do not support the conclusion that VAPA and VAPB levels are regulated by autophagy or basal autophagy. Notably, WB is per se a semi-quantitative approach. Moreover, all quantifications shown in the manuscript (Figure 1, 2, 4, 6, 7, ...) lack indication of the statistic relevance of the data.

Answer: We apologize that in our initial text the aim of the experiment depicted in Fig 6A did not come out clearly. Since we report a role for CALCOCO1 in starvation induced ER-phagy, we wanted to test if VAPA and -B are degraded by autophagy in HBSS treated cells. Our conclusion is that the VAP proteins are degraded by autophagy in response to HBSS treatment. In our revised MS, we have added bar graphs for VAPA and VAPB (revised figure 6B). Degradation by autophagy is indicated both in the bar graphs and in the panels depicted in figure 6A. First, we show that in ATG5 KO MEFS, VAPA and -B accumulates in response to the addition of HBSS. This presumably reflects that the synthesis of VAPs is induced by starvation. Second, in WT cells there is no such increase in their levels in response to HBSS, and their levels are instead reduced. This correlates with a more efficient degradation in WT cells. Third, BafA1 stabilizes the VAPs in HBSS treated WT cells (HBSS+BafA1), indicating that the observed degradation is by autophagy. Finally, BafA1 does not similarly stabilize the VAPs in HBSS treated ATG5 KO MEFs. In conclusion, our data indicate a degradation of VAPs by autophagy in HBSS treated cells.

We have now added bar graphs (including standard deviations) in the revised figures 1D, 1F, 4I, 6B, 6F, 7B, 7D and 7G for those experiments that we consider most important to quantify. In our initial MS and in our revised MS, several western blot experiments contain numbers under individual blots indicating the relative intensity of detected bands. These numbers indicate intensities of bands seen in the shown blot. When not showing bar graphs, we consider this to be the best way to quantify a western blot experiment, since the depicted gel and the corresponding numbers are directly compared. We consider the quantified single experiments as representative since they are all repeated several times with similar results.

NEW COMMENT: OK.

OLD COMMENT: The experiments performed in cells depleted of both VAPA and VAPB and the conclusions reported at page 16 would imply that in these cells CALCOCO1 does not associate with the ER membrane, does not traffic in the autophagosomes and is not delivered in LAMP1-organelles. This should be checked, shown and quantified.

Answer: As is shown in Fig 6C (Fig 6C and D in revised MS), efficient degradation of CALCOCO1 itself by autophagy is dependent on the VAP proteins as there is a clear accumulation of CALCOCO1 when both VAPA and VAPB are knocked down. As is seen in new Fig 6D, there is also a further accumulation of CALCOCO1 seen in BafA1-treated cells. From this data we conclude that the efficient trafficking of CALCOCO1 to autophagosomes or LAMP1 organelles is clearly affected by the loss of VAP proteins. However, there is also a degradation of CALCOCO1 that may be VAP-independent. This is not surprising as CALCOCO1 is likely involved in more than one selective autophagy pathway. In new Fig 6D we have also added RTN3 (ER tubules), FAM134B (ER sheets) and p62 and their behavior confirms the role of VAPs in the degradation of tubular ER (see modified

text).

NEW COMMENT: The authors should comment on differences between 6A (where FAM134B does accumulate in cells treated with BafA1) and new 6D, where it does not. For fig. 6C and D quantification is missing.

OLD COMMENT: Figure 6E is not convincing. One puncta of co-localization is shown. Moreover, by looking at the lower panel, I think that the arrow in LC3B and the arrow in EGFP-CALCOCO1 are showing two different puncta. To be more convincing, the authors should show the co-localization of these markers within LAMP1 compartments (as in Figure 1I) and quantify.

Answer: In the revised Fig (Fig 6G in revised MS), we have added more arrows and corrected the arrow that was misplaced in our initial figure (we thank the reviewer for seeing this). Co-localization with LAMP1 is shown in Fig 1K and quantified Fig EV2D and for VAPA and -B in Figs 8C and -D. The only protein that is not quantified in co-localization with LAMP1 is then LC3B that is known to co-localize with LAMP1 in Baf A1 treated cells.

NEW COMMENT: Fig. 6G does not exist, authors are referring to Fig. 6E. OK.

OLD COMMENT: Figure 7 The conclusion/comment at page 17, end of the first paragraph "More specifically, CALCOCO1 KO impaired starvation-induced degradation of tubular ER proteins VAPA and VAPB but not ER sheets marker FAM134B or autophagy receptor p62 (Fig 7A)." is wrong. In fact, CALCOCO1 KO also impairs starvation-induced p62 degradation (the quantification of more than 3 biological replicates gives an unchanged value of 1.2 in HBSS with and without BafA1).

Answer: From studying p62 in two decades we have often observed that the combination of HBSS and Baf A1 gives variable accumulation of p62. This problem is never seen in FM. It is also not seen upon Torin 1 + Baf A1. The reason for the variability upon HBSS + Baf A1 is probably that much of p62 is quickly degraded before Baf A1 inhibits the degradation (HBSS and BafA1 are added together). There is also less translation in HBSS+BafA1 treated cells since autophagy is inhibited. We have repeated this experiment a number of times and consistently see a stabilization of p62 upon HBSS + Baf A1 in CALCOCO1 KO cells. Even in the displayed blot, we clearly see in the gel image that there is more p62 in HBSS+BafA1 treated cells than in HBSS treated cells. Since our initial panel is confusing and does not correlate with our repeated observations, we have therefore in our revised MS replaced the blot shown in our initial figure 7A with a new p62 blot where the difference is also quantified. For the revision, we also added a blot of NDP52 in our revised figure 7A, illustrating a pattern very similar to that of p62. The pattern of FAM134B is also very similar to those of p62 and NDP52, and our interpretation of the data is that HBSS induced degradation of all these three proteins is normal in CALCOCO1 KO cells.

NEW COMMENT: Having changed some of the panels, the authors should note that the loading controls do not always correspond to the gel/WB shown in the figure.

OLD COMMENT: The conclusion/comment at page 17, second paragraph "Compared to the non-

induced cells, induced expression of EGFP-CALCOCO1 restored starvation and proteotoxic stress-induced degradation of tubular ER proteins RTN3, VAPA and VAPB (Fig 7B)." also seems wrong (or badly formulated). On expression of CALCOCO1 (as in its absence) proteotoxic stress (i.e., MG132) does not induce degradation of RTN3 (the RTN3 level actually increases, +1.4 times), VAPA (+1.3) or VAPB (+1.4). Also, starvation-induced degradation of RTN3 is very modestly affected as judged by the 20 and 10% level reduction, respectively, whereas VAPB increases upon starvation (1.3). Similar comments are valid for Figure 7D (where quantifications are missing).

Answer: We have now added bar graphs for the data in original Figs 7B (Fig 7D in the revised MS). We apologize to the reviewer for not explaining clearly how we interpret autophagy of ER proteins in response to starvation. This is now explained in our text. The production of ER proteins is strongly induced in response to proteotoxic stress or starvation. In the absence of ER-phagy, this causes expansion of the ER and accumulation of ER proteins. ER-phagy is induced to prevent this increase in the ER and the net effect is a level of ER proteins close to the level in FM. Hence, inhibition of ER-phagy is measured as an accumulation of ER proteins in HBSS- or MG132-treated cells. This explanatory text now starts the paragraph "CALCOCO1 is a soluble ER-phagy receptor". Induced expression of EGFP-CALCOCO1 in KO cells consistently caused starvation-induced degradation of tubular ER proteins RTN3, TEX264, VAPA and VAPB that was blocked by Baf A1 treatment (Fig 7C and D).

NEW COMMENT: The addition of "markers" for tubular ER that are themselves ER-phagy receptors activated upon nutrient deprivation adds confusion and raises new questions. The authors report that turnover of the ER-phagy receptors RTN3 and TEX264 (not of FAM134B) during starvation is regulated by CALCOCO1. This is not discussed. The author should put this in the context of the available literature on starvation-induced ER-phagy (e.g., the An et al and the Chino et al Mol Cell 2019 papers).

Minor:

MW markers are still absent in figs. 2, 3, EV1, EV4.

Page 17: As far as I know, nutrient deprivation does not increase the size of the ER. The authors are probably referring at ER stress.

Page 18 and 19, the receptor is TEX264 (not 26 or 164)

Referee #3:

The revised manuscript has been improved, but there are still several points that are not fully supported by experimental data or overinterpreted.

1-1:
The authors now show that CALCOCO1 is important for bulk (non-selective) autophagy under basal conditions, whereas the same protein is important for ER-phagy but not for bulk autophagy under starvation conditions. However, even the defect in bulk autophagy in CALCOCO1 KO cells under basal conditions is subtle: p62, NBR1, and NDP52 accumulate only slightly in CALCOCO1 KO cells (Fig. 4C, D, and E), which the authors demonstrate with only one immunoblot for each

experiment. Statistical analysis of several independent experiments is essential to suggest that CALCOCO1 has a role in basal autophagy. In addition, to clearly differentiate bulk autophagy and ER-phagy, this reviewer still recommends using specific reporters such as RFP-GFP-LC3 (for bulk autophagy) and RFP-GFP-ER protein (for ER-phagy). These reporters should be more sensitive and specific than detecting the amount of degradation of endogenous proteins.

1-2:

It is still unclear whether the roles of CALCOCO1 and VAPA/B are specific to tubular ER.

- The data in Fig. 6D is important, but again, the authors show only one blot without full quantification. As the differences are not large, statistical analysis of several independent experiments is essential. In addition, this experiment should include sheet ER markers such as CLIMP1.

- The authors use TEX264 as a marker for tubular ER, but the rationale behind this choice is unclear.

- The authors mention "In WT cells, starvation-induced degradation was seen for VAPA, FAM134B, p62, and NDP52" (Page 18). However, starvation-induced degradation of VAPA is not observed in Fig. 7A and B-in fact, it seems to increase after starvation.

- Although the authors state "The starvation-induced degradation of the tubular ER proteins" (Page 18), it is observed only for TEX264 in Fig. 7D (comparing the back and grey bars). The amounts of RTN3 and VAPA/B are unchanged.

- The amount of the sheet ER marker CLIMP63 is higher in TET-OFF cells than that in TET-ON cells, suggesting that CALCOCO1 also regulates the amount of sheet ER.

Having these data, it is difficult to conclude that the role of CALCOCO1 (and VAPA/B) is specific to tubular ER. If the authors still want to propose this model, the hypothesis should be validated using more specific markers such as tandem fluorescent protein-tagged tubular and sheet ER markers.

4: In the rebuttal letter, the authors admit that deletion of the FFAT-like motif does not affect the ER localization of CALCOCO1. The model in Fig. 8E showing VAPs recruiting CALCOCO1 to the ER is then inaccurate. To avoid misleading readers, the authors should present the data of the localization of the CALCOCO1 Δ LIR Δ 671-691 mutant and discuss the importance of VAPs in CALCOCO1 recruitment.

6: With regards to multiple comparison analysis, the kind of post-hoc test used after ANOVA should be described.

Re: EMBOJ-2019-103649R

Revision of "CALCOCO1 acts with VAMP-Associated Proteins to mediate ER-phagy"

Dear Dr. Argenzio,

Thank you for the reviews of our revised manuscript EMBOJ-2019-103649R entitled "CALCOCO1 acts with VAMP-Associated Proteins to mediate ER-phagy". We are grateful for the opportunity to submit a 2nd revised version of our manuscript and thank the reviewers for their constructive criticism and helpful comments that we have used to improve our paper further.

In the revised manuscript we have added new data in the form of new figure items and also revised original figure items. Hence, the revised MS contains 5 new main figure items (3F, 4D, 4F, 4H, 6E) and 7 new EV figure items (5C, 6A, 6B, 6D, 7A, 7B, 7C). We have revised 13 main figure items (1I, 1J, 2D, 2F, 2G, 2H, 2I, 3B, 3C, 3D, 5C, 5D, 6D) and 8 EV figure items (1B, 1C, 1D, 4A, 4B, 4C, 4D, 4E), and in two of these we have added new data (2I, 6D). We have also added appropriate quantifications and statistical analyses and missing molecular weight markers as requested by the reviewers. New text in the revised version is indicated in red.

Below we have answered all the comments made by the reviewers.

Referee #2:

The manuscript has been improved. Inconsistent results shown in the previous version of the manuscript have disappeared (in some cases the authors modified their presentation to "make graphs more informative", in other cases they shifted from manual to automatized quantification to "analyze the data in what we consider to be a more accurate way", in other cases they "performed quantitative analyses more carefully", or panels have been replaced).

However, it is my opinion that the paper cannot be accepted as it is. There are mistakes in the figures, some inconsistency remains and should be explained, some "intriguing" result is not commented at all. There still are gels lacking MW markers, individual WB panels have been replaced but the loading controls remain those of the old panel (appropriate loading controls will be shown in the uncropped source images?).

I have added "NEW COMMENTS" after the authors' responses to my original points.

OLD COMMENT: The finding that deletion of >50% of the polypeptide sequence abolishes formation of CALCOCO1 homomeric complexes (Fig 1B) is not surprising/uninteresting (or

certainly less important than the finding shown in Fig. EV1, where CC3 is identified as the "interacting region"). The authors should consider to swap 1B with (some of) the data shown in EV1. Is homomeric complex formation required for CALCOCO1 function in ER-phagy?

Answer: We tried to swap Fig 1B with Fig EV1A, but because of the larger size of Fig EV1A we found no satisfying way of displaying this. Therefore, the figures are not swapped in the revised MS. We made a cell line expressing a EGFP-CALCOCO1 Δ CC construct lacking the coiled-coil region (see attached image below with full-length mCherry-CALCOCO1 transiently transfected). Since this construct was highly mislocalized and accumulated in the nucleus when stably expressed in CALCOCO1 KO cells, we therefore chose to not use this cell line in our revised MS.

NEW COMMENT: This does not seem to answer our request. We are referring at the Δ CC3 protein (corresponding to myc- Δ 413-513), and not at the Δ CC protein (Δ 145-513) mentioned in the response. An easy solution would be to repeat the experiment shown in Fig. 1B (i.e., transient transfection of HEK293 cells followed by co-IP) by adding a lane showing that the Δ CC3 protein does not co-IP Myc-CALCOCO1. And then my question was if formation of the homomeric complex is required for the putative function of CALCOCO1 in ER-phagy (for instance, it is not required to associate with ATG8s, Fig. 2D).

NEW ANSWER: We thank the referee for clarifying the request. We actually tested the individual CC deletions in our initial co-IP experiments using transfected HEK293 cells. The complete experiment is now shown in the revised manuscript (Figure EV1A). When immunoprecipitated from transfected cells, a deletion of the entire CC region prevented binding, but individual CC deletion CALCOCO1 constructs, including the Δ CC3 construct, co-precipitated with WT EGFP-CALCOCO1 (new Figure EV1A). This suggested that all the CC regions contributed to the self-interaction of CALCOCO1. This was our reasoning when we processed out part of the co-IP figure showing individual CC deletions because we thought it did not add any new information. However, the co-IP experiment using *in vitro*-translated proteins strongly indicated that CC3 is more important for the self-interaction than other CC regions. Our reasoning was that the *in vitro* co-IP was a better way of investigating direct protein-protein interactions because it is not interfered with by indirect interactions from other protein complexes that are usually present in cell extracts. That is why we presented the *in vitro* co-IP figure in full. In conclusion, the previous figure 1B is now deleted and replaced by an extended figure inserted into figure EV1 as EV1A.

Is the self-interaction needed for the function of CALCOCO1 in ER-phagy? To test the importance of the self-interaction in cells, we used cells reconstituted with the Δ 145-513 construct (Δ CC) since this is the only construct that did not self-interact in any of our binding assays. EGFP-CALCOCO1 Δ 145-513 did not induce ER-phagy (new figure EV7C), suggesting that self-interaction is important.

OLD COMMENT: Figure 1G, H As expected, the number of CALCOCO1 puncta significantly increases on nutrient deprivation + BafA1 vs. MEM + BafA1 (panel 1G). The authors should explain why the CALCOCO1 co-localization with autophagy markers substantially decreases in HBSS + BafA1 compared to MEM + BafA1 (panel 1H, where error bars are missing).

Answer: In the revised MS, we have changed the y-axis of original Fig 1G (Fig 1I in revised MS) so that it shows the total number of puncta (instead of fold increase in puncta shown in our initial figure). This does not affect any of our conclusions, but the bar graphs become more informative this way. In particular, the use of fold increase was problematic since the number of CALCOCO1 puncta in FM is very low, making relative fold increases higher for CALCOCO1 as compared to LC3B and p62.

More importantly, we have also revised Fig 1H and Fig EV2D (Fig 1J in revised MS). We analyzed more images and analyzed them in what we consider to be a more accurate way. Instead of manually selecting CALCOCO1 positive puncta, we detected the puncta automatically, and this way we strongly increased the number of puncta analyzed for each cell. We thank the reviewer for the criticism of our initial analysis. We have now clearly improved our analysis of the CALCOCO1 puncta. We have also included error bars in Fig 1J. The reviewer asks about the substantial decrease in co-localization in HBSS + Baf compared to FM + Baf. In our new analysis (new Fig 1J), there was no difference in the co-localization with p62 and the difference in LC3B co-localization was rather small. Hence, our new data show that there is no substantial decrease in co-localization in HBSS-treated cells.

NEW COMMENT: Authors should specify "puncta per cell" on y-axis in Fig.1I-J. What do error-bars represent? And what about significance? The authors should specify in the legend that an automated system to quantify puncta has been used.

NEW ANSWER: We have done as suggested. The error bars represent mean \pm sd of puncta per cell. Significance is displayed as ***p < 0.001, **p < 0.005. This is now mentioned in the figure legend.

OLD COMMENT: Crucially, as shown in all other reports describing new ER-phagy receptors, the authors must show the association of CALCOCO1 (endogenous and recombinant) with endogenous, lipidated LC3s (and/or GABARAPs). These interactions should be abolished on deletion of the CALCOCO1's domain that mediates association with LC3/GABARAPs.

Answer: We show in a new Fig 2I that recombinant GST-CALCOCO1 interacts strongly with endogenous, lipidated GABARAP from cell extracts. A construct deleted for the LIR and UIR motifs did not interact, verifying that the interaction depends on the identified motifs.

NEW COMMENT: Panel 2I shows that the ratio lipidated/non-lipidated GABARAP associated with GST (negative control) and with CALCOCO1 does not change. Based on current knowledge on ER-phagy receptors, these bind the lipidated forms of the ATG8 proteins. The fact that CALCOCO1 does not seem to distinguish GABARAP-I and -II (and does not show preferential binding to the latter) is not commented. This analysis should also be done to monitor the association of LC3 forms. Is "lack of preference for the lipidated forms of ATG8s" an intrinsic property of CALCOCO1? This would distinguish it from ER-phagy receptors.

NEW ANSWER: We have repeated the experiment to monitor the association of CALCOCO1 with LC3B. The new data (Figure 2I), indicate that more LCB-II than LC3B-I binds to CALCOCO1. Due to lack of a good CALCOCO1 antibody for endogenous IP, we were unable to determine whether endogenous CALCOCO1 prefers binding to lipidated or unlipidated forms of ATG8s. Since we don't know the in vivo binding preference of endogenous CALCOCO1 therefore, we do not want to speculate too much on what forms of ATG8s are preferred by CALCOCO1. We have gone through the available literature and found no support for such a preference among ER-phagy receptors in general. Most papers we have seen show data where the ATG8 protein is immunoprecipitated and the receptor co-precipitated (e.g. Fig 1C from Chino et al., 2019 Molecular Cell 74, 909–921 shown below) and therefore it is impossible to

conclude about the binding preference.

Figure for Referees not shown

OLD COMMENT: Figure 3E reveals a crucial difference in behavior if one compares ectopic CALCOCO1 (not stabilized by BafA1 in cells exposed to HBSS (lane 5) with the endogenous one (Figure 1C, lane 5, stabilized by BafA1 in cells exposed to HBSS). This difference may question the use of ectopic CALCOCO1 and the extent to which the recombinant protein recapitulates the behavior of the endogenous one.

Answer: We cannot exclude that there may be differences between ectopically, stably expressed EGFP-CALCOCO1 and endogenous CALCOCO1 that can affect our results. When investigating autophagy in HBSS treated cells (HBSS + Baf A1 versus HBSS alone), the accumulation of the endogenous protein seems to be more easily detected in western blots than the corresponding accumulation of ectopic EGFP-CALCOCO1. We do not consider it to be a major problem since we do also consistently observe an accumulation of the ectopically expressed protein. In the revised MS we have replaced the old blot of EGFP-CALCOCO1 (WT) with a new blot that more clearly show that the ectopic protein is stabilized by Baf A1. For the new blot, the same lysates used for the old blot were used.

NEW COMMENT: Having changed the relevant panel in Fig. 3E, the authors should note that the loading control (Actin?) is from another gel/WB. Since the experiment has been repeated many times and the results were consistent, the "n" should be given and the significance of the variations should be shown.

NEW ANSWER: We have now informed in the figure legends of figures 3E, 6D, 7A and 7C that more than one loading control is used, but only one shown. For the first revision, we initially showed two different loading controls in these figure to indicate this, but this became too confusing and we chose to show only one loading control. For figure 3E, we have also included a new figure 3F showing quantification based on three experiments.

OLD COMMENT: Figure 4 (MW are missing in panel C and are wrong in panel D (EGFP-CALCOCO1 should be 125kD)). Figure 4A and first paragraph, page 13. This is unclear. Are the authors writing that in response to starvation there is INCREASED co-localization of ectopically expressed CALCOCO1 and WIPI2 and ATG13? This should be shown by comparing FM vs. HBSS (+/-BafA1 as done for other autophagy markers in 1H) and should be quantified.

Answer: We agree that the text on CALCOCO1 co-localization with WIPI and ATG13 was unclear. We have revised the text and added quantification in a new Fig 4B for the EGFP-CALCOCO1 co-localization with WIPI dots. There is a starvation-induced increase in both the number of WIPI and EGFP-CALCOCO1 puncta and a corresponding linear increase in the number of co-localized puncta. BafA1 did not affect the number of WIPI dots or the co-localization with EGFP-CALCOCO1, and therefore we did not include this data in the revised MS.

NEW COMMENT: The new Fig. 4B must be wrong! The authors show that there is (in average) 1 WIPI punctum per cell? Moreover, the authors are quantifying in FM and HBSS. Fig. 4A should therefore show both conditions.

NEW ANSWER: We have added a new figure EV5C showing the co-localization in FM and HBSS. We do not understand why the reviewer says the data in figure 4B must be wrong. From our experience, the observed average number of WIPI puncta per cell in full medium (FM) is always very low, and it correlates with what others have observed previously (e.g. Polson et al. *Autophagy*. 2010 May;6(4):506-22).

OLD COMMENT: Figure 6 The WB shown in this figure do not support the conclusion that VAPA and VAPB levels are regulated by autophagy or basal autophagy. Notably, WB is per se a semi-quantitative approach. Moreover, all quantifications shown in the manuscript (Figure 1, 2, 4, 6, 7, ...) lack indication of the statistic relevance of the data.

Answer: We apologize that in our initial text the aim of the experiment depicted in Fig 6A did not come out clearly. Since we report a role for CALCOCO1 in starvation induced ER-phagy, we wanted to test if VAPA and -B are degraded by autophagy in HBSS treated cells. Our conclusion is that the VAP proteins are degraded by autophagy in response to HBSS treatment. In our revised MS, we have added bar graphs for VAPA and VAPB (revised figure 6B). Degradation by autophagy is indicated both in the bar graphs and in the panels depicted in figure 6A. First, we show that in ATG5 KO MEFs, VAPA and -B accumulates in response to the addition of HBSS. This presumably reflects that the synthesis of VAPs is induced by starvation. Second, in WT cells there is no such increase in their levels in response to HBSS, and their levels are instead reduced. This correlates with a more efficient degradation in WT cells. Third, BafA1 stabilizes the VAPs in HBSS treated WT cells (HBSS+BafA1), indicating that the observed degradation is by autophagy. Finally, BafA1 does not similarly stabilize the VAPs in HBSS treated ATG5 KO MEFs. In conclusion, our data indicate a degradation of VAPs by autophagy in HBSS treated cells.

We have now added bar graphs (including standard deviations) in the revised figures 1D, 1F, 4I, 6B, 6F, 7B, 7D and 7G for those experiments that we consider most important to quantify. In our initial MS and in our revised MS, several western blot experiments contain numbers under individual blots indicating the relative intensity of detected bands. These numbers indicate intensities of bands seen in the shown blot. When not showing bar graphs, we consider this to be the best way to quantify a western blot experiment, since the depicted gel and the corresponding numbers are directly compared. We consider the quantified single experiments as representative since they are all repeated several times with similar results.

NEW COMMENT: OK.

OLD COMMENT: The experiments performed in cells depleted of both VAPA and VAPB and the conclusions reported at page 16 would imply that in these cells CALCOCO1 does not associate with the ER membrane, does not traffic in the autophagosomes and is not delivered in LAMP1-organelles. This should be checked, shown and quantified.

Answer: As is shown in Fig 6C (Fig 6C and D in revised MS), efficient degradation of CALCOCO1 itself by autophagy is dependent on the VAP proteins as there is a clear accumulation of CALCOCO1 when both VAPA and VAPB are knocked down. As is seen in new Fig 6D, there is also a further accumulation of CALCOCO1 seen in BafA1-treated cells. From this data we conclude that the efficient trafficking of CALCOCO1 to autophagosomes or LAMP1 organelles is clearly affected by the loss of VAP proteins. However, there is also a

degradation of CALCOCO1 that may be VAP-independent. This is not surprising as CALCOCO1 is likely involved in more than one selective autophagy pathway. In new Fig 6D we have also added RTN3 (ER tubules), FAM134B (ER sheets) and p62 and their behavior confirms the role of VAPs in the degradation of tubular ER (see modified text).

NEW COMMENT: The authors should comment on differences between 6A (where FAM134B does accumulate in cells treated with BafA1) and new 6D, where it does not. For fig. 6C and D quantification is missing.

NEW ANSWER: We agree that the increase in FAM134B with Baf A1 in figure 6D is smaller than in 6A, but we have looked at several blots and there is a consistent increase. We have now quantified figure 6D in a new figure 6E, and this data, based on several blots confirms the increase in FAM134B.

OLD COMMENT: Figure 6E is not convincing. One puncta of co-localization is shown. Moreover, by looking at the lower panel, I think that the arrow in LC3B and the arrow in EGFP-CALCOCO1 are showing two different puncta. To be more convincing, the authors should show the co-localization of these markers within LAMP1 compartments (as in Figure 1I) and quantify.

Answer: In the revised Fig (Fig 6G in revised MS), we have added more arrows and corrected the arrow that was misplaced in our initial figure (we thank the reviewer for seeing this). Co-localization with LAMP1 is shown in Fig 1K and quantified Fig EV2D and for VAPA and -B in Figs 8C and -D. The only protein that is not quantified in co-localization with LAMP1 is then LC3B that is known to co-localize with LAMP1 in Baf A1 treated cells.

NEW COMMENT: Fig. 6G does not exist, authors are referring to Fig. 6E. OK.

NEW ANSWER: The text has been modified accordingly

OLD COMMENT: Figure 7 The conclusion/comment at page 17, end of the first paragraph "More specifically, CALCOCO1 KO impaired starvation-induced degradation of tubular ER proteins VAPA and VAPB but not ER sheets marker FAM134B or autophagy receptor p62 (Fig 7A)." is wrong. In fact, CALCOCO1 KO also impairs starvation-induced p62 degradation (the quantification of more than 3 biological replicates gives an unchanged value of 1.2 in HBSS with and without BafA1).

Answer: From studying p62 in two decades we have often observed that the combination of HBSS and Baf A1 gives variable accumulation of p62. This problem is never seen in FM. It is also not seen upon Torin 1 + Baf A1. The reason for the variability upon HBSS + Baf A1 is probably that much of p62 is quickly degraded before Baf A1 inhibits the degradation (HBSS and BafA1 are added together). There is also less translation in HBSS+BafA1 treated cells since autophagy is inhibited. We have repeated this experiment a number of times and consistently see a stabilization of p62 upon HBSS + Baf A1 in CALCOCO1 KO cells. Even in the displayed blot, we clearly see in the gel image that there is more p62 in HBSS+BafA1 treated cells than in HBSS treated cells. Since our initial panel is confusing and does not correlate with our repeated observations, we have therefore in our revised MS replaced the blot shown in our initial figure 7A with a new p62 blot where the difference is also quantified. For the revision, we also added a blot of NDP52 in our revised figure 7A, illustrating a pattern very similar to that of p62. The pattern of FAM134B is also very similar to those of p62 and NDP52, and our interpretation of the data is that HBSS induced degradation of all these three proteins is normal in CALCOCO1

KO cells.

NEW COMMENT: Having changed some of the panels, the authors should note that the loading controls do not always correspond to the gel/WB shown in the figure.

NEW ANSWER: We have done this (see comment to figure 3E above).

OLD COMMENT: The conclusion/comment at page 17, second paragraph "Compared to the non-induced cells, induced expression of EGFP-CALCOCO1 restored starvation and proteotoxic stress-induced degradation of tubular ER proteins RTN3, VAPA and VAPB (Fig 7B)." also seems wrong (or badly formulated). On expression of CALCOCO1 (as in its absence) proteotoxic stress (i.e., MG132) does not induce degradation of RTN3 (the RTN3 level actually increases, +1.4 times), VAPA (+1.3) or VAPB (+1.4). Also, starvation-induced degradation of RTN3 is very modestly affected as judged by the 20 and 10% level reduction, respectively, whereas VAPB increases upon starvation (1.3). Similar comments are valid for Figure 7D (where quantifications are missing).

Answer: We have now added bar graphs for the data in original Figs 7B (Fig 7D in the revised MS). We apologize to the reviewer for not explaining clearly how we interpret autophagy of ER proteins in response to starvation. This is now explained in our text. The production of ER proteins is strongly induced in response to proteotoxic stress or starvation. In the absence of ER-phagy, this causes expansion of the ER and accumulation of ER proteins. ER-phagy is induced to prevent this increase in the ER and the net effect is a level of ER proteins close to the level in FM. Hence, inhibition of ER-phagy is measured as an accumulation of ER proteins in HBSS- or MG132-treated cells. This explanatory text now starts the paragraph "CALCOCO1 is a soluble ER-phagy receptor". Induced expression of EGFP-CALCOCO1 in KO cells consistently caused starvation-induced degradation of tubular ER proteins RTN3, TEX264, VAPA and VAPB that was blocked by Baf A1 treatment (Fig 7C and D).

NEW COMMENT: The addition of "markers" for tubular ER that are themselves ER-phagy receptors activated upon nutrient deprivation adds confusion and raises new questions. The authors report that turnover of the ER-phagy receptors RTN3 and TEX264 (not of FAM134B) during starvation is regulated by CALCOCO1. This is not discussed. The author should put this in the context of the available literature on starvation-induced ER-phagy (e.g., the An et al and the Chino et al Mol Cell 2019 papers).

NEW ANSWER: This is an interesting point, although in western blots we stain for the short form of RTN3 that has the same localization on ER, but is not an ER-phagy receptor. We used TEX264 as a marker for tubular ER-phagy because it is localized at 3-way junctions of the tubular ER. The degradation of the tubular ER therefore will inevitably include degradation of TEX264. Further studies however are needed to understand, mechanistically, whether CALCOCO1 co-operates with other ER-phagy receptors

Minor:

MW markers are still absent in figs. 2, 3, EV1, EV4.

Answer: We have now added MWs in figures 2D, 2F, 2G, 2H, 2I, 3B, 3C, 3D, 5C, 5D, EV1B, EV1C, EV1D, EV4A, EV4B, EV4C, EV4D, and EV4E, and then there should be MWs in all protein gels.

Page 17: As far as I know, nutrient deprivation does not increase the size of the ER. The authors are probably referring at ER stress.

Answer: This is now corrected in the text.

Page 18 and 19, the receptor is TEX264 (not 26 or 164)

Answer: This is now corrected in the text.

Referee #3:

The revised manuscript has been improved, but there are still several points that are not fully supported by experimental data or overinterpreted.

1-1:

The authors now show that CALCOCO1 is important for bulk (non-selective) autophagy under basal conditions, whereas the same protein is important for ER-phagy but not for bulk autophagy under starvation conditions. However, even the defect in bulk autophagy in CALCOCO1 KO cells under basal conditions is subtle: p62, NBR1, and NDP52 accumulate only slightly in CALCOCO1 KO cells (Fig. 4C, D, and E), which the authors demonstrate with only one immunoblot for each experiment. Statistical analysis of several independent experiments is essential to suggest that CALCOCO1 has a role in basal autophagy.

Answer: We have added three new figures including quantification of p62 and NDP52 in old fig 4C (see new figures 4C and D), old fig 4D (see new figures 4 E and F) and old fig 4E (see new figures 4G and H). Although the effect on LC3B-II and GABARAP-II was more consistent than the effect on the SLRs, the new data show that also the basal levels of the SLRs were affected by the lack of CALCOCO1.

In addition, to clearly differentiate bulk autophagy and ER-phagy, this reviewer still recommends using specific reporters such as RFP-GFP-LC3 (for bulk autophagy) and RFP-GFP-ER protein (for ER-phagy). These reporters should be more sensitive and specific than detecting the amount of degradation of endogenous proteins.

Answer: We have now done the double tag experiments, and our data with LC3B indicates an effect of CALCOCO1 in FM, but not so in response to starvation. With FAM134B, we observed no difference between cells expressing CALCOCO1 or not, and with VAPA CALCOCO1 induced degradation of VAPA in starved cells, but not in FM.

1-2:

It is still unclear whether the roles of CALCOCO1 and VAPA/B are specific to tubular ER.
- The data in Fig. 6D is important, but again, the authors show only one blot without full quantification. As the differences are not large, statistical analysis of several independent experiments is essential. In addition, this experiment should include sheet ER markers such as CLIMP1.

Answer: In figure 6D, we have added data for FAM134B and CLIMP63, and we have added quantitative data in figure 6E based on several independent experiments.
The new data shows that KD of VAPs does not affect the degradation of the sheet ER markers: FAM134B and CLIMP63.

- The authors use TEX264 as a marker for tubular ER, but the rationale behind this choice is unclear.

Answer: TEX164 is included since it is localised at the 3-way junctions of the tubular ER. The degradation of the tubular ER therefore will inevitably include degradation of TEX264. TEX164 is itself an ER-phagy receptor, and in the revised MS discussion we have discussed the possibility that CALCOCO1 may potentially cooperate with resident tubular ER-phagy receptors.

- The authors mention "In WT cells, starvation-induced degradation was seen for VAPA, FAM134B, p62, and NDP52" (Page 18). However, starvation-induced degradation of VAPA is not observed in Fig. 7A and B-in fact, it seems to increase after starvation.

Answer: We agree that our use of the text line "starvation induced degradation" can be misunderstood. When we wrote starvation induced degradation, we did not mean to say that there is a net degradation. In the absence of ER-phagy. There is an expansion of ER and accumulation of ER proteins under starvation. ER-phagy is induced to prevent this increase in the ER and the net effect is a level of ER proteins close to the level in FM. We have now revised the text on pages 18 and 19 to clarify this, so that the data in figures 7A-D are described correctly.

- Although the authors state "The starvation-induced degradation of the tubular ER proteins" (Page 18), it is observed only for TEX264 in Fig. 7D (comparing the black and grey bars). The amounts of RTN3 and VAPA/B are unchanged.

Answer: See our response to the previous question.

- The amount of the sheet ER marker CLIMP63 is higher in TET-OFF cells than that in TET-ON cells, suggesting that CALCOCO1 also regulates the amount of sheet ER. Having these data, it is difficult to conclude that the role of CALCOCO1 (and VAPA/B) is specific to tubular ER. If the authors still want to propose this model, the hypothesis should be validated using more specific markers such as tandem fluorescent protein-tagged tubular and sheet ER markers.

Answer: We have now performed double tag experiments that supports our conclusion, and all the data we have indicate that CALCOCO1 is important for the degradation of tubular ER-proteins. Since ER is a very complex organelle, it is not possible based on this single study to conclude that CALCOCO1 is only involved in degradation of tubular ER. But we find no evidence for that CALCOCO1 is needed for degradation of FAM134 or CLIMP63 under starvation, suggesting that it is not a receptor for ER sheets. We have also shown in our revised figure 6D and new figure 6E that knock down of VAPs has no effect on the level of the sheet markers FAM134B and CLIMP63. This is now mentioned in our text on page 22 where we suggest that the effect CALCOCO1 has on basal CLIMP63 level may be indirect since a loss of CALCOCO1 may affect ER homeostasis.

4: In the rebuttal letter, the authors admit that deletion of the FFAT-like motif does not affect the ER localization of CALCOCO1. The model in Fig. 8E showing VAPs recruiting CALCOCO1 to the ER is then inaccurate. To avoid misleading readers, the authors should present the data of the localization of the CALCOCO1 Δ LIR Δ 671-691 mutant and discuss the importance of VAPs in CALCOCO1 recruitment.

Answer: We favour the idea that the VAP interaction is important for attaching CALCOCO1 to the ER, but very likely other interactions are also important. To avoid misleading the readers, we have now included new data for the localization pattern of the FFAT-like motif mutated

CALCOCO1 construct (new figure EV6D), we have inserted new text in the results section on page 16, and we have added the following text in the discussion: “Deletion of the FFAT-like motif in CALCOCO1 did not prevent its localization on the ER, but this may be because the deletion mutant show some interaction with the VAPs and therefore can be recruited via these weak interactions. Our model (**Fig 8D**) shows the interaction of CALCOCO1 with the VAPs and the ATG8 family proteins. However, it is also possible that CALCOCO1 co-operates with other proteins. Further studies are needed to address this possibility.”

6: With regards to multiple comparison analysis, the kind of post-hoc test used after ANOVA should be described.

Answer: It`s one-way ANOVA, followed by Tukey multiple comparison text.

We thank the reviewers for their effort to help us to further improve our manuscript and hope that the manuscript now can be found acceptable for publication.

Thank you for submitting a revised version of your manuscript . It has now been seen by referee #2 and #3, whose comments are shown below.

As you will see, they both find that the remaining criticisms have been sufficiently addressed and recommend the manuscript for publication. However, there are a few editorial issues concerning text and figures that I need you to address before we can officially accept the manuscript .

Referee #2:

The ms has substantially been improved and can now be accepted for publication in the EmboJ.

Referee #3:

The authors have appropriately responded to this reviewer's concerns. The new data using the tandem fluorescent protein reporters are helpful. In these experiments, the authors should describe what "red-only dots (%)" in the Y-axis indicates (what is the denominator of the %?).

Accepted**6th May 2020**

I am pleased to inform you that your manuscript has been accepted for publication in The EMBO Journal.

Terje Johansen
EMBO J
2019-103649R